# IAPv4 ocean temperature and ocean heat content gridded dataset

Lijing Cheng[1,11*], Yuying Pan[1,11], Zhetao Tan[1,11], Huayi Zheng[1,11], Yujing Zhu[1,11], Wangxu Wei[1,11], Juan Du[1], Huifeng Yuan[2,11], Guancheng Li[3], Hanlin Ye[1], Viktor Gouretski[1], Yuanlong Li[4,11], Kevin E. Trenberth[5,6], John Abraham[7], Yuchun Jin[4,11], Franco Reseghetti[8], Xiaopei Lin[9], Bing Zhang[4,11], Gengxin Chen[10, 11], Michael E. Mann[12], Jiang Zhu[1,11]

[1] Institute of Atmospheric Physics, Chinese Academy of Sciences, Beijing, China, 100029.

[2] Computer Network Information Center, Chinese Academy of Sciences, Beijing, 100083.

[3] Eco-Environmental Monitoring and Research Center, Pearl River Valley and South China Sea Ecology and Environment Administration, Ministry of Ecology and Environment, PRC, Guangzhou 510611, China.

[4] Institute of Oceanography, Chinese Academy of Sciences, Qingdao, China.

[5] National Center for Atmospheric Research, PO Box 3000, Boulder, CO 80307, USA.

[6] University of Auckland, Auckland, New Zealand.

[7] University of St. Thomas, School of Engineering, 2115 Summit Ave., St Paul, MN 55105, USA.

[8] Istituto Nazionale di Geofisica e Vulcanologia, 40127, Bologna, Italy.

[9] Frontier Science Center for Deep Ocean Multispheres and Earth System and Physical Oceanography Laboratory, Ocean University of China, Qingdao, China.

[10] State Key Laboratory of Tropical Oceanography, South China Sea Institute of Oceanology, Chinese Academy of Sciences, Guangzhou, China

[11] University of Chinese Academy of Sciences, Beijing, China.

[12] Dept. of Earth and Environmental Science, University of Pennsylvania, Philadelphia PA, USA

*Correspondence to*: Lijing Cheng (chenglij@mail.iap.ac.cn)

**Abstract.** Ocean observational gridded products are vital for climate monitoring, ocean and climate research, model evaluation, and supporting climate mitigation and adaptation measures. This paper describes the 4th version of the Institute of Atmospheric Physics (IAPv4) ocean temperature and ocean heat content (OHC) objective analysis product. It accounts for recent developments in quality control (QC) procedures, climatology, bias correction, vertical and horizontal interpolation, and mapping and is available for the upper 6000 m (119 levels) since 1940 (more reliable after ~1957) for monthly and $1° \times 1°$ temporal and spatial resolutions. The IAPv4 is compared with the previous version, IAPv3, and to the other data products, sea surface temperatures (SSTs), and satellite observations. It has a slightly stronger long-term upper 2000 m OHC increase than IAPv3 for 1955-2023, mainly because of newly developed bias corrections. IAPv4 OHC 0-2000 m trend is also higher during 2005-2023 than IAPv3. The uppermost level of IAPv4 is consistent with independent SST datasets. The month-to-month OHC variability for IAPv4 is desirably less than IAPv3 and other OHC products investigated in this study, the trend of ocean warming rate (i.e., warming acceleration) is more consistent with the net energy imbalance at the top of the atmosphere than IAPv3, and the sea level budget can be closed within uncertainty. The gridded product is freely accessible at: http://dx.doi.org/10.12157/IOCAS.20240117.002 for temperature data (Cheng et al., 2024a) and http://dx.doi.org/10.12157/IOCAS.20240117.001 for ocean heat content data (Cheng et al., 2024b).

## 1. Introduction

Observational gridded products are essential for understanding the ocean, the atmosphere, and climate change; they support policy decisions and social-economy developments (Abraham et al., 2022; Abraham and Cheng, 2022; Cheng et al., 2022a). For instance, many of the climate indicators used in the Working Group I report of the 6th Intergovernmental Panel on Climate Change (IPCC-AR6-WG1) are based on gridded products (Gulev et al., 2021; IPCC, 2021), mainly because the raw oceanic data suffer from inhomogeneous data quality and irregular and incomplete data coverage (Abraham et al., 2013; Boyer et al., 2016; Cheng et al., 2022a; Meyssignac et al., 2019).

As more than 90% of the Earth's energy imbalance (EEI) in the past half-century has accumulated in the ocean, increasing ocean temperature ($T$) and ocean heat content (OHC) are essential climate variables for monitoring, understanding, and projecting climate change (e.g., Rhein et al., 2013; Hansen et al., 2011; Trenberth, 2022; Trenberth et al., 2009; von Schuckmann et al., 2020; Cheng et al., 2022). OHC also impacts air-sea and ice-sea interactions and thus exerts a considerable influence over the other components of the climate system. It provides critical feedback through energy, water, and carbon cycles (Cheng et al., 2022a; Trenberth, 2022; von Schuckmann et al., 2016). Substantial changes in ocean temperatures also profoundly impact ocean biogeochemical processes and ecosystems and are critical for ocean health and human society (Bindoff et al., 2019; Cheng et al., 2022a).

Many gridded T/OHC datasets have been produced by independent groups, and most of them are updated annually or more frequently (Cheng et al., 2022a; Good et al., 2013; Hosoda et al., 2008; Ishii et al., 2017; Levitus et al., 2012; Li et al., 2017; Meyssignac et al., 2019; Roemmich and Gilson, 2009). Most widely-used products are at 1° × 1° horizontal resolution and monthly temporal resolution from near-surface to at least 2000 m depth. Some products utilize all available *in situ* observations and span at least half a century, prominent examples being the data products compiled by the Institute of Atmospheric Physics (IAP) (Cheng and Zhu, 2016; Cheng et al., 2017) from 1940-present; Japan Meteorological Agency (JMA) (Ishii et al., 2017) from 1955-present; National Centers for Environmental Information (NCEI), National Oceanic and Atmospheric Administration (NOAA) from 1950-present (Levitus et al., 2012); and University of California since 1949 (Bagnell and DeVries, 2021). As Argo data has achieved near-global upper 2000 m open ocean coverage since ~2005, many Argo-based or Argo-only gridded products are available. Examples include gridded products from SCRIPPS after 2004 (Roemmich and Gilson, 2009); China Argo Real-time Data Center since 2005 (Li et al., 2017); and Copernicus since 2005 (von Schuckmann and Le Traon, 2011). These products usually span from ~2005 to the present for the upper ~2000 m. These data benefit from the high quality of Argo data but are not fully resolving polar regions, shallow waters, and regions with complex topography.

In 2016, the IAP group provided its first gridded product for the upper 700 m ocean (Cheng and Zhu, 2016) by merging all available observations since 1960. With a revised mapping method and a thorough evaluation process with synthetic observations, an update

(IAP version 3, IAPv3) became available in 2017 for the upper 2000 m ocean with data
since the 1950s (Cheng et al., 2017). The IAPv3 has supported scientific research, climate
assessment reports, and monitoring practices (Bindoff et al., 2019; Gulev et al., 2021;
WMO, 2022).
After the release of IAPv3, there has been progress with observation data quality
control and new/updated techniques for temperature data processing and reconstruction.
For example, Gouretski et al. (2022) found that old Nansen cast bottle data contained
systematic biases that impacted the T/OHC data before 1990. Revisions are also available
to the bias corrections for the Mechanical Bathythermographs (MBT) and eXpendable
Bathythermographs (XBT) data (Cheng et al., 2014; Gouretski and Cheng, 2020), mainly
impacting the data within 1940–2005. Tan et al. (2023) developed a new quality-control
system that advances the detection of outliers after accounting for the non-Gaussian
distribution of local temperatures in determining the local climatological range. The impact
of inhomogeneous vertical resolution of temperature profiles has been recognized
previously (Cheng and Zhu, 2014) and received more attention recently (Li et al., 2020)
with a new vertical interpolation approach (Barker and McDougall, 2020). Upgrading the
product with new developments is important to better support the ocean/climate research
and climate assessments.
This manuscript discusses the revisions to the IAP ocean objective analysis product
(IAPv4) since the publication of the IAPv3 (Cheng et al., 2017). The data and methods are
introduced in Section 2 and the results are presented in Section 3, with analyses of the
character of the IAPv4 on regional and global scales and at various time scales. The EEI
and sea level budgets based on the new data product are also investigated. A summary and
discussion are provided in Section 4, with some remaining issues and outlooks being
discussed.

**2. Data and Methods**
**2.1 Data source**
The majority of the *in situ* measurements used to create the data product come from
the World Ocean Database (WOD), downloaded in September 2023. Data from all
instrument types are used, including XBTs (Goni et al., 2019), Argo (Argo 2000),
Conductivity/Temperature/Depth profilers (CTDs), MBTs, bottles, moorings, gliders,
Animal Borne Ocean Sensors (McMahon et al., 2021) and others (Boyer et al., 2018) (Fig.

1). There is a total of 17,634,865 temperature profiles from January 1940 to September 2023 (Fig. 1a). MBT, XBT, Nansen Bottle and CTD data are the major instruments before 2000 (Fig. 1a, b). The spatial coverage of these data increased to >30% in 1960 and >70% in the late 1960s for $1° \times 1° \times 1$-year resolution. After 2005, there is a huge number of GLD and APB data, and as they are mainly distributed in the polar regions (APB) and coastal regions (GLD) (Fig. 1a), their spatial coverage is usually less than 5% for $1° \times 1° \times 1$ year resolution. By contrast, the Argo data cover most of the global open ocean since ~2005 (Fig. 1b).

Argo data are processed following the recommendations of the Argo community. Adjusted data are used where applicable. Both Delayed- and Real-Time Argo data have been incorporated in IAPv4. As Real-Time Argo data have only passed automated, simple QC tests in real-time, these data may still contain temperature, pressure, and salinity values affected by unknown errors. However, through a sensitivity study, Cheng (2024) indicated that including Real-Time Argo data does not bias the OHC calculation for the IAP analysis. Nevertheless, IAP data are updated frequently (every 1-3 months): each time the updated Argo data is used, the T/OHC fields are recalculated following the recommendation by the Argo group (Wong et al., 2020). The data from the Argo floats in the "grey list" have been removed from the calculation (https://data-argo.ifremer.fr/).

To complement the WOD with relatively less data in the Arctic and coastal regions of the Northwest Pacific, this presented product also uses data from other sources. The majority of these data are from the Chinese Academy of Sciences Ocean Science Data Center (Zhang et al., 2024), and some data are rescued from the old documents of marine surveys. All these data will be publicly available. There are a total of 85,990 additional temperature profiles, about 0.50% of the data, which is expected to improve the reconstruction in these data-sparse regions (compared with IAPv3 and other products).

The *in situ* data have been processed as described in a flow chart in Figure 2. In the following sections, the key techniques of data processing are introduced.

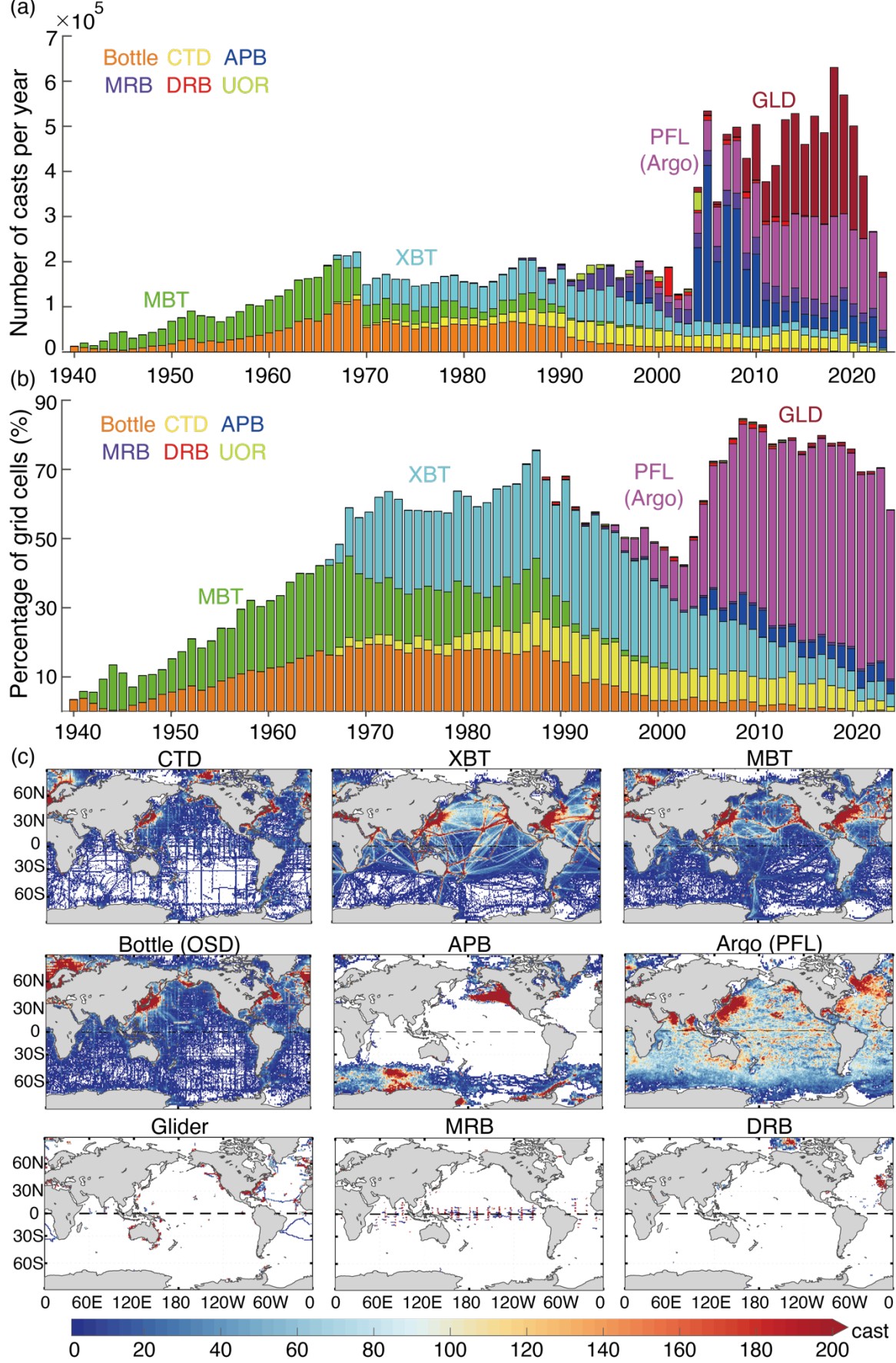

153

**Figure 1: (a) Yearly number of temperature casts for different instruments; (b) percentage coverage (%) of ocean data for each instrument, calculated as the ratio between the number of 1° × 1° × 1 year grid cells observed by each instrument and the total number of ocean grids; (c) number of subsurface temperature casts in 1-degree grid boxes from 1940 to 2023 collected by different instruments:** CTD (Conductivity/Temperature/Depth), XBT (eXpendable BathyThermographs), MBT (Mechanical BathyThermograph), Bottle, APB (Animal mounted Pinniped Borne), PFL (Profiling Floats, i.e. Argo), GLD (Glider), MRB (Moored Buoy), and DRB (Drifting Buoy).

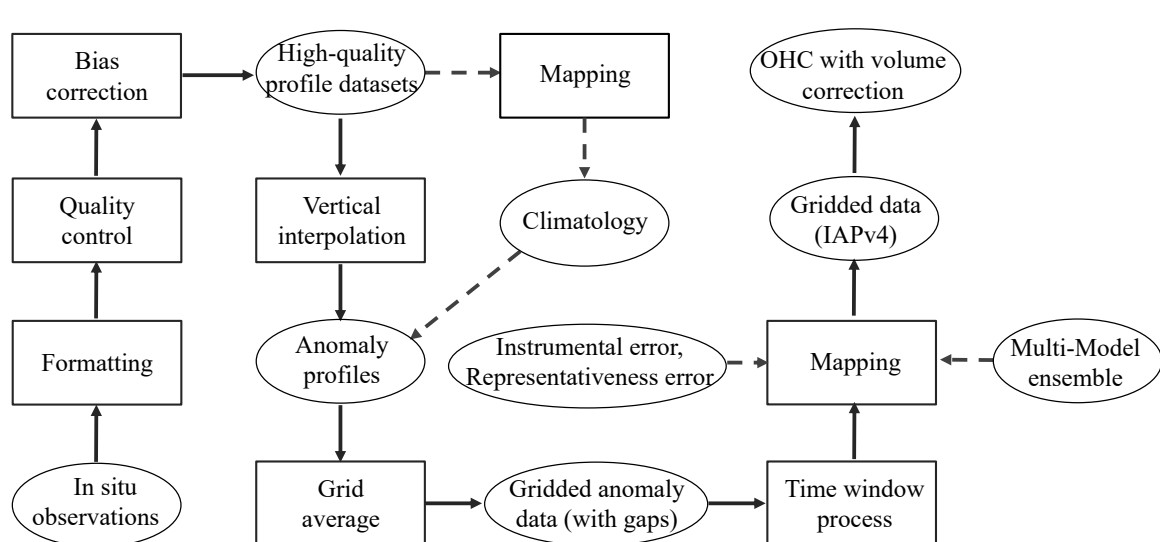

**Figure 2: Flow chart of the IAP data reconstruction processes from the raw *in situ* observations to gridded data (IAPv4) and OHC estimates.** The ellipses indicate the data (including data for error estimates), and the rectangle boxes show the techniques used to process the data.

**2.2 Data quality control**

The quality control (QC) procedure aims to identify spurious measurements (including outliers) and data with incorrect metadata through a set of quality checks and ensures the quality of the *in situ* dataset (Tan et al., 2022). There is growing evidence that QC is critical for accurate temperature/OHC reconstruction, as shown by Tan et al., (2023) where two different QC systems produced a difference of approximately 15 % (~7 %) in the OHC 0-2000 m trend from 1955 to 1990 (2005-2021). Unfortunately, the impact of QC on OHC

estimates has not been evaluated in previous community-assessments on T/OHC uncertainty (Boyer et al., 2016; Lyman et al., 2010). In this study, the QC procedure follows the CAS-Ocean Data Center (CODC) Quality Control system, named CODC-QC (Tan et al., 2023), where only the "good" data (flag=0) are used.

The CODC-QC system (Tan et al., 2023) has the following strengths, which make it particularly suitable for T/OHC reconstruction:

1) A new local climatological range is defined in this CODC-QC system to identify the outliers. Unlike many existing QC procedures, no assumption is made of a Gaussian distribution law in the new approach, as the oceanic variables (e.g., temperature and salinity) are typically skewed. Instead, the 0.5 % and 99.5 % quantiles are used as thresholds in CODC-QC to define the local climatological parameter ranges.

2) Local climatological ranges change with time to account for the long-term trends of ocean temperature accompanied by more frequent extreme events (e.g., Oliver et al., 2018; Sun et al., 2023). Previously, the use of the static local ranges tended to remove too many "extreme values" (at the tails of the temperature distributions) associated with climate change in recent years that were actually real, leading to a QC-procedure related bias in the gridded dataset and OHC estimate (Tan et al., 2023).

3) In addition, local climatological ranges for the vertical temperature gradient are constructed to account for the variability of 'vertical shape', increasing the ability of the scheme to identify spurious profiles.

4) The QC procedure is instrument-specific, accounting for characteristics inherent to particular instrumentation types. For example, XBT digital recording systems are allowed to continue to record beyond the rated terminal depth suggested by manufacturers (T7/DB probes below 760 m; T4/T6 below 460 m; T5 below 1830 m). Below the rated maximum depth, the XBT wire often breaks, leading to a characteristic change in recorded temperature values. The new QC procedure effectively identifies such profiles.

5) The thorough evaluation of the QC procedure performance and the application of the QC procedure to the manually QC-ed datasets (Thresher et al., 2008; Gouretski and Koltermann, 2004) demonstrated the effectiveness of the proposed scheme in removing spurious data and minimizing the percentage of mistakenly flagged good data.

Being applied to the entire temperature profile dataset the CODC-QC procedure identifies 6.22 % of all temperature measurements as outliers. The rejection rates (definition follows Tan et al., 2023) vary among instrumentation types (3.73 % for CTD,

1.97 % for Argo, 12.06 % for XBT, 4.93 % for MBT, 6.54 % for bottle, 5.92 % for APB,
4.54 % for DRB, 2.55 % for MRB). The overall percentage of outliers decreases over time
from ~5 % in the 1940s to ~2.5 % in the 2020s, reflecting the progressive improvement of
the instrumentation (Fig. 3). A rejection rate maximum (~12 %) during 2000~2010 is
linked to the XBT data, which are especially abundant in the 800–1100m layer and are
characterized by higher rejection rate below the maximum depth (Tan et al., 2023). The
generally higher rejection rate below 4000 meters is related to the gross errors (such as
measurements cooler than -2°C, big spikes, etc.) and the occurrence of the constant values
(recorded values don't change with depth). For example, the higher rejection rate within
2008-2009 below 4000 meters is because of the gross errors in the CTD data.

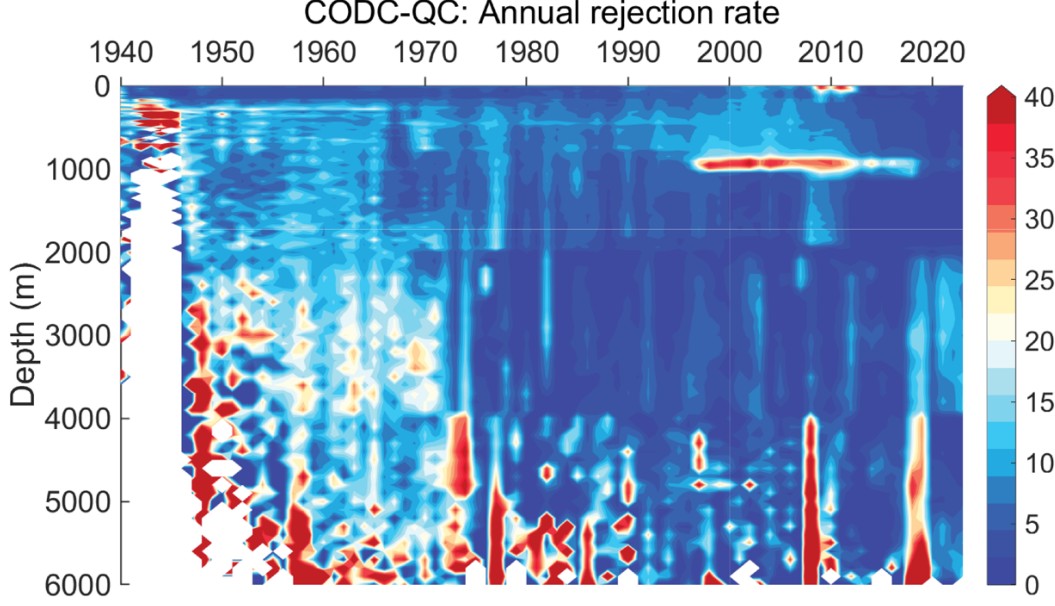


**Figure 3: The rejection rate (%) after CODC-QC as a function of calendar year and**
**depth.**

**2.3 Bias correction**
It is well known that data from several instrument types can exhibit biases both in
temperature and depth. Temperature profiles obtained using XBTs and MBTs provide an
example of biased data, especially because of uncertainties in the depth of measurement.
Gouretski and Koltermann (2007) demonstrated their significant impact on the magnitude
and variability of the global OHC estimates. That study triggered a series of publications
where different bias correction schemes have been suggested for XBT (Gouretski and
Reseghetti, 2010; Abraham et al., 2013; Cheng et al., 2016; Levitus et al., 2009; Wijffels et
al., 2008), MBT (Gouretski and Cheng 2020; Levitus et al., 2009) and other instruments
(Fig. 2). In the compilation of IAPv4, newly developed bias correction schemes are
applied.
The XBT temperature bias was found to be generally positive, as large as ~0.1 °C
before 1980 on the global 0–700 m average, diminishing to less than 0.05 °C after 1990
(Gouretski and Koltermann 2007; Wijffels et al., 2008). Here, we use an updated XBT bias
correction scheme (Cheng et al., 2014) to correct both depth and temperature biases in
XBT data, following the community recommendation (Cheng et al., 2016; Goni et al.,
2019). The depth and temperature biases depend on ocean temperature, probe type, and
time. An inter-comparison among several correction schemes rated the CH14 scheme the
most successful (Cheng et al., 2018). Using XBT and collocated CTD data, we updated the
CH14 scheme by re-calculating bias corrections between 1966-2016 and extending them
for the years 2017 to 2023.
Comparison with collocated reference CTD profiles recently revealed significant
biases in the old hydrographic profiles obtained by means of Nansen bottle casts
(Gouretski et al., 2022). Both depth and temperature measurements of bottle casts were
found to be biased, and the proposed correction scheme was also implemented in IAPv4.
The thermal bias is related to the time needed to bring the mercury thermometers in
equilibrium with the ambient temperature after the completion of the hydrographic cast.
The depth bias indicates an overestimation of the bottle depth due to the wire's deviation
from the vertical position and is mostly related to the hydrographic casts where the
thermometrical method of sample depth determination was not used. The correction
scheme includes a constant thermal bias of -0.02 °C and a depth- and time-variable depth
bias.
The MBT bias is as large as 0.28 °C before 1980 for the global average and reduces to
less than 0.18 °C after 1980 for the 0~200 m average. IAPv3 used Ishii and Kimoto, (2009)
(IK09) scheme to correct MBT bias, while a new scheme proposed by Gouretski and
Cheng, (2020) (GC20) is adopted in IAPv4. This shift is made because our assessment
indicates the under-correction of MBT bias by the IK09 scheme within the upper 120 m
and over-correction in the deeper layer, whereas GC20 corrects both depth and temperature
biases. GC20 also found the MBT bias to be country-dependent, explained in terms of
different instrumentation characteristics and working procedures. Therefore, the time-

varying bias corrections are applied separately for the MBT profiles obtained by ships from the United States, Soviet Union/Russia, Japan, Canada, and Great Britain. Data from all other countries are corrected using a globally averaged correction.

Finally, thermal biases were recently reported for the data obtained by different kinds of data loggers attached to marine mammals (APB). Gouretski et al. (2024) analysed temperature profiles obtained between 2004 and 2019 in the high and moderate latitudes of both hemispheres. Comparison with the collocated reference CTD and Argo float data revealed a systematic negative thermal offset (average value -0.027 °C) for mammal temperature profiles from SRDL (satellite-related data loggers). For the less accurate data from TDR (Temperature-Depth-Recorders), the comparison revealed a small positive temperature bias of 0.02 °C and the depth (pressure) bias indicating depth overestimation.

**2.4 Climatology**

For IAP and other data product generators, horizontal interpolation (mapping) is applied on a temperature anomaly field after removing a monthly climatology; thus, a pre-defined climatology field with an annual cycle is mandatory (Fig. 2). The accuracy of the climatology field is one of the key sources of uncertainty in reconstruction because the error in climatology will propagate into the anomaly field, impact the spatial dynamical consistency, and the accuracy of the reconstruction (Cheng and Zhu, 2015; Lyman and Johnson, 2014; Boyer et al., 2016).

In IAPv4, the adjusted mapping procedure (see below) has been applied to reconstruct the climatology field (Table 1). The merit of using IAP mapping for climatology is its ability to better represent the spatial anisotropy of temperature variability (non-Gaussian distribution). Unlike IAPv3, where the 1990–2005 reference period was used, IAPv4 uses data between 2006 and 2020 to construct 12 monthly climatologies, taking advantage of more reliable data combined with better and more homogeneous spatial and temporal coverage in the last two decades (Table 1). Following the recommendation in Cheng and Zhu, (2015), a relatively short period of 15-year is used because climatology constructed with longer period of data will result in different baselines at different locations (i.e., the baseline shifted to earlier years in the middle latitudes of the North Hemisphere and the baseline shifted to more recent years in the Southern Hemisphere) and this inconsistency will violate the spatial structure of the anomaly field (Cheng and Zhu, 2015). Recent

developments from other groups, such as Li et al., (2022), include the choice of a short-
period climatology.

IAPv4 used an 800 km influencing radii in climatology reconstruction, smaller than

the 20° for IAPv3, to more properly account for the rapid change of temperatures with
distance. There is a trade-off between data availability and the size of the influence radius.
Using radii smaller than 500 km does not ensure a global fractional coverage (defined as
the fraction of the total ocean area obtained by the mapping method) because of data
sparseness (Cheng, 2024). As our tests suggest, using 500~800 km results in very similar
reconstructions of climatology, therefore, 800 km is adopted.

**2.5 Vertical interpolation**

The vertical resolution of ocean temperature profiles changed dramatically over time

associated with instrument evolution and the increase of data storage capability. For
instance, the global mean vertical resolution at 500 m level changed from ~100 m in the
1960s to less than 10 m during the 2010s (Li et al., 2020). Vertical interpolation of the raw
profiles on standard levels is a critical process (Fig. 2): Cheng and Zhu (2014) indicated
that the use of linear or spline vertical interpolation methods can bias the temperature
reconstruction and OHC estimation (Barker and McDougall, 2020; Li et al., 2020; Li et al.,
2022). IAPv3 used the (Reiniger and Ross, 1968) (RR) method. Recently, Barker and
McDougall (2020) proposed a new approach using multiple Piecewise Cubic Hermite
Interpolating Polynomials (PCHIPs) to minimize the formation of unrealistic water masses
by the interpolation procedure.

Because the largest difference between interpolation methods is found mostly for the

low-resolution profiles (e.g., old Nansen casts), in practice, extremely low vertical
resolution profiles had to be removed to reduce the uncertainty in interpolation. In IAPv4,
this procedure is optimized compared to IAPv3, and only parts of profiles with a sufficient
vertical resolution are used. The thresholds for the vertical resolution are set by 50 m in the
upper 200m, 200m between 200 m and 1000 m, 500 m between 1000 m and 2000 m, and
600m between 2000 m and 6000 m. As no interpolation method can adequately interpolate
temperature for the vertical resolution beyond these thresholds, interpolation is not
performed in such cases to avoid errors (these extreme low-resolution data are not used in
further processing). Under this limitation for IAPv4, we still apply the RR method for
temperature profiles.
Finally, IAPv4 extends the set of standard vertical levels with a total of 119 levels
from 1 m to 6500 m (79 levels within the upper 2000 m) compared to 41 levels in IAPv3
between 1 m and 2000 m (Table 1). The increase in vertical resolution is critical for
accurately representing the mixed layer, as investigated below.

**2.6 Grid average and mapping**
The anomaly profiles are obtained by subtracting the monthly mean climatology from
the vertically interpolated profiles. These anomalies are then averaged (arithmetic mean)
into a 1° × 1° grid at each standard level (1° × 1° gridded average field) (Fig. 2). Due to the
general data sparsity, variable time windows (larger than one month) are used for monthly
reconstructions to ensure a truly global analysis (Supplementary Table 1). This process
takes advantage of the larger persistence of anomalies (generally smaller monthly and
inter-annual variability) in the deep ocean than in the upper ocean and thus is physically
grounded. Specifically, after 2005, data within a three-month window are merged to
provide a monthly reconstruction for each layer of the upper 1950 m. Before 2005, a time-
varying and depth-varying time window is used, and it is generally smaller in the upper
ocean and wider in the deeper ocean (Supplementary Table 1). Below 2000 m, a 5-year
(60-month) window is adopted. The use of a time window will reduce the monthly
variance compared to other datasets, which is likely too high compared with independent
Earth's Energy Imbalance data at the top of the atmosphere (Trenberth et al., 2016).
Mapping interpolates the gridded (e.g., box-averaged) observations horizontally into a
spatially complete map (Fig. 2) because not all 1° × 1° boxes are filled with data. (Fig. 2).
IAPv4 adopted a similar mapping approach (Ensemble Optimal Interpolation with dynamic
ensemble: EnOI-DE) as in IAPv3 introduced in Cheng and Zhu (2016) and Cheng et al.,
(2017) but with the following modifications:
1) the largest influence radius has changed from 20° in the upper 700 m (25° at 700–
2000 m) in IAPv3 to 2,000 km in the upper 700 m (2,500 km at 700–6000 m) in IAPv4, to
account for the reduced distance between two longitudes from tropics to the polar regions.
This change mainly helps to improve the reconstruction in the high-latitude regions;
2) The three iterative runs are taken to effectively bring in different scales of
variability with influencing radius changing from 2,000 km (2,500 km at 700–6000 m) to
800 km and 300 km, respectively, based on the tests presented in Cheng and Zhu (2016)
and Cheng et al., (2017);
3) For each month, IAPv3 used 40 model simulations (historical runs) from the
Coupled Model Intercomparison Project phase 5 (CMIP5) to provide a flow-dependent
ensemble, which is then constrained by observations to provide optimized spatial
covariance. IAP mapping uses model-based covariance because we argue that spatial
covariance can never be satisfactorily parametrized by some simple basic functions (such
as Gaussian) given its complexity. With model-based, flow-dependent, and dynamically-
consistent covariance, the IAP mapping provides a more realistic reconstruction than other
approaches based on Gaussian-based parameterized covariance, as evaluated by many
studies (Cheng et al., 2017; Cheng et al., 2020; Dangendorf et al., 2021; Nerem et al.,

2018).

4) The observation error variance (**R**), which represents the error of the observations,
is updated in IAPv4 as follows. **R** consists of both the instrumental error (**Re**) due to
inaccuracy and the representativeness error (**Rr**) due to the need to represent the spatial (at
1° by 1° and 1 m standard grid depths) and temporal (1 month) averages from a limited
numbered of observations (Cheng and Zhu, 2016):
$$\mathbf{R} = \mathbf{Re} + \mathbf{Rr} = (\sum_1^M \boldsymbol{Ei})/M + \sigma^2/M,$$
where M observations exist for a given grid cell. $\boldsymbol{Ei}$ is the instrument's precision for
each individual observation, assuming random error (the basic assumption is that after bias
correction, the systematic errors can be eliminated). **Re** in each grid cell is set to the mean
of the typical precision of the different instruments contributing data in the cell, which is
set according to IQuOD (International Quality-Controlled Ocean Database) specification
(Cowley et al., 2021). $\sigma^2$ represents the variance of the various temperature measurements
against the monthly mean value. The data from 2005 to 2022 are used to calculate $\sigma^2$ in
each grid because of greater data abundance and quality compared to earlier times.
As the representativeness error (**Rr**) is expected to be flow-dependent (i.e., the error is
expected to be higher in areas with a large gradient of the flow speed and regions of higher
variability), more observations are required to represent the mean value. Figure 4 shows a
larger variance ($\sigma^2$) in the boundary-current regions and near the Antarctic Circumpolar
Current (ACC) in the upper ocean (e.g., 10 m, 200 m, 500 m). At 200 m, it shows a larger
$\sigma^2$ in the Western Pacific Ocean, corresponding to the large thermocline variations at this
layer. Below 1000 m, larger $\sigma^2$ along the ACC frontal regions and in the North Atlantic
Ocean occur because of a stronger mixing and convection in these regions.
The uncertainty in the derived gridded reconstruction is also based on the EnOI
framework formulated by Cheng and Zhu, (2016). The uncertainty accounts for
instrumental, sampling and mapping errors. Other error sources, including the choice of
climatology, vertical interpolation, bias corrections, and QC, are not considered in this
uncertainty estimate. Therefore, a more thorough uncertainty quantification method is
needed, and this is under development in a separate study.

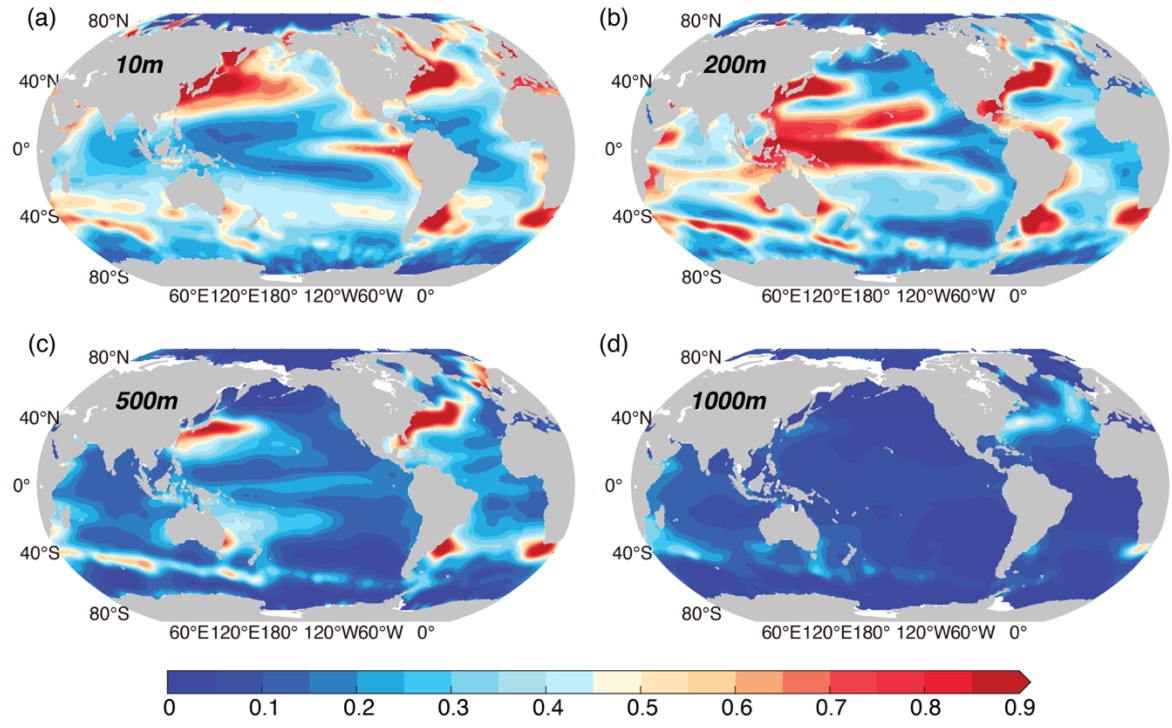


**Figure 4: Variance (σ²) of ocean temperature at several representative layers.** (a)
10 m, (b) 200 m, (c) 500 m and (d) 1000 m. The unit is °C².

**Table 1.** General information on IAPv4 and IAPv3 data products.

| | IAPv3 | IAPv4 |
|---|---|---|
| **Horizonal resolution** | Global (1° × 1°) | Global (1° × 1°) |
| **Vertical levels** | 41 levels from 1 m to 2000 m (1, 5, 10, 20, 30, 40, 50, 60, 70, 80, 90, 100, 120, 140, 160, 180, 200, 250, 300, 350, 400, 450, 500, 550, 600, 650, 700, 750, 800, 850, 900, 1000, 1100, 1200, 1300, 1400, 1500, 1600, 1700, 1800, 2000) | 119 levels from 1 m to 6000 m (1, 5, 10, 15, 20, 25, 30, 35, 40, 45, 50, 55, 60, 65, 70, 75, 80, 85, 90, 95, 100, 110, 120, 130, 140, 150, 160, 170, 180, 190, 200, 220, 240, 260, 280, 300, 320, 340, 360, 380, 400, 425, 450, 475, 500, 525, 550, 575, 600, 625, 650, 675, 700, 750, 800, 850, 900, 950, 1000, 1050, 1100, 1150, 1200, |

| | | |
|---|---|---|
| | | 1250, 1300, 1350, 1400, 1450, 1500, 1550, 1600, 1650, 1700, 1750, 1800, 1850, 1900, 1950, 2000, 2100, 2200, 2300, 2400, 2500, 2600, 2700, 2800, 2900, 3000, 3100, 3200, 3300, 3400, 3500, 3600, 3700, 3800, 3900, 4000, 4100, 4200, 4300, 4400, 4500, 4600, 4700, 4800, 4900, 5000, 5100, 5200, 5300, 5400, 5500, 5600, 5700, 5800, 5900, 6000) |
| **Time period and resolution** | 1940–2022 (reliable data after 1955), monthly | 1940–present (reliable data after 1955), monthly |
| **Quality-control** | WOD (Garcia et al., 2018) | CODC-QC (Tan et al., 2023) |
| **Vertical interpolation** | RR (Reiniger and Ross, 1968) interpolation | RR (Reiniger and Ross, 1968) interpolation |
| **Climatology** | IAP climatology: simple gridded average and then spatial interpolation with distance-weighted average | Improved IAP reconstruction with EnOI approach |
| **XBT bias correction** | CH14 (updated in 2018) | CH14 (revised and updated in 2023) |
| **MBT bias correction** | IK09 (Ishii and Kimoto, 2009) | GC20 (Gouretski and Cheng, 2020) |
| **APB bias correction** | None | GCR24 (Gouretski et al., 2024) |
| **Bottle bias correction** | None | GCT22 (Gouretski et al., 2022) |
| **Mapping** | EnOI-DE with influencing radius of 20, 8, 3 degrees, iteratively. | EnOI-DE with influencing radius of 2000, 800, 300 km, iteratively. Representative error updated with 2005-2022 observations. The radius of influence does not cross the land. |
| **Uncertainty** | Given by EnOI framework accounting for instrumental error and horizonal sampling/mapping error | Given by EnOI framework accounting for instrumental error and horizonal sampling/mapping error |
| **DOI** | / | http://dx.doi.org/10.12157/IOCAS.20240117.002 for temperature data (Cheng et al., 2024a) and http://dx.doi.org/10.12157/IOCAS.20240117.001 for ocean heat content data (Cheng et al., 2024b) |


**2.7 OHC calculation and volume correction**

Based on the gridded temperature reconstruction (Table 1), OHC in each grid is calculated as OHC $(x, y, z) = c_p \iiint_{V(x,y,z)} \rho T dV(x,y,z)$. following TEOS-10 standards, where $c_p$ is a constant of $\sim$ 3991.9 J (kg K)$^{-1}$ according to the new TEOS-10 standard formulation as conservative temperature and absolute salinity are used, $\rho$ is potential density in kg m$^{-3}$, and $T$ is conservative temperature measured in degrees Celsius (here it is anomaly relative to the 2006–2020 baseline) (Cheng et al., 2022a).

As OHC is an integrated metric over a specific ocean volume, properly identifying ocean volume is critical, especially in shallow waters. Previous studies found a 10–20 % difference in the OHC trend in recent decades between different land-ocean masks (von Schuckmann and Le Traon, 2011; Meyssignac et al., 2019; Savita et al., 2022). Specifically, in marginal sea areas with complex topography, $1° \times 1° \times \Delta z$ grid boxes (where $\Delta z$ is the depth range of the grid box) near coasts and islands typically cover both ocean and land areas but are assigned to represent land or ocean only. Thus, the gridded ocean temperature datasets are subjected to errors from inaccurate land-sea attribution. Here, we offer a volume correction (VC) for these grid boxes to improve the OHC estimate, as follows.

For each $1° \times 1° \times \Delta z$ grid box, we introduce a VC factor (denoted as $F_{VC}$) to correct the OHC values: OHC$_{VC}$ (x, y, z) = OHC (x, y, z) $\times$ F$_{VC}$ (x, y, z). First, we assume the seawater volume distribution in 1 arc-minute topographic data of ETOPO1 as "truth". No correction is needed if a box is assigned to ocean according to ETOPO1 data, thus, F$_{VC}$=1. If a fraction of a $1° \times 1° \times \Delta z$ grid box is land according to ETOPO1 and IAP data includes T/OHC values, the F$_{VC}$ is represented by the fraction of the ocean volume in this box (illustrated in Fig. 5), and the volume for OHC calculations can be corrected with F$_{VC}$(i). In a grid box, if there is no IAP data (i.e., it is land according to the IAP mask), but this box contains some ocean volume according to ETOPO1 data, we define F$_{VC}$ (a) again as the fraction of the ocean volume in this box, and then this F$_{VC}$(a) is added to the adjacent grid boxes where there are values in IAP data. If all the adjacent grid boxes contain no data, the volume is equally redistributed to the diagonal boxes (Fig. 5). The volume is discarded if there is no data in all adjacent and diagonal boxes.

With this approach, the VC factor in each grid box is a sum of two components: a

local adjustment $F_{VC}(i)$ and a redistribution from the adjacent grids:

$F_{VC}(a))$: $F_{VC} = F_{VC}(i) + F_{VC}(a)$,

To avoid misidentification of sea ice, we performed VC only on the global grid points

within 60 °S to 60 °N. Eventually, we obtained a three-dimensional FVC that fits the IAP
grids ($119 \times 360 \times 180$; depth coverage to 6000 m) and used it to compute OHC. The VC
applied to ~15% of all the $1° \times 1° \times \Delta z$ grid boxes of IAPv4 ocean grid boxes (with $F_{VC} \neq$
1) for the entire 0-6000 m ocean and ~10% grid boxes of the upper 2000 m. Since the open
ocean accounts for the vast majority of the global ocean volume, the influence of the VC
method on the global OHC trend is small. For example, the upper 2000 m OHC trend with
VC is ~0.15% (~0.45%) smaller than without VC from 1958-2023 (2005-2023) for IAPv4.
However, it can significantly affect regional OHC estimates, especially in regions with
complex topography. For example, the Maritime Continent region's 0-2000 m OHC trend
is reduced by 6.9% (4.2%) after applying VC from 1958-2023 (2005-2023) (Jin et al.,

2024).

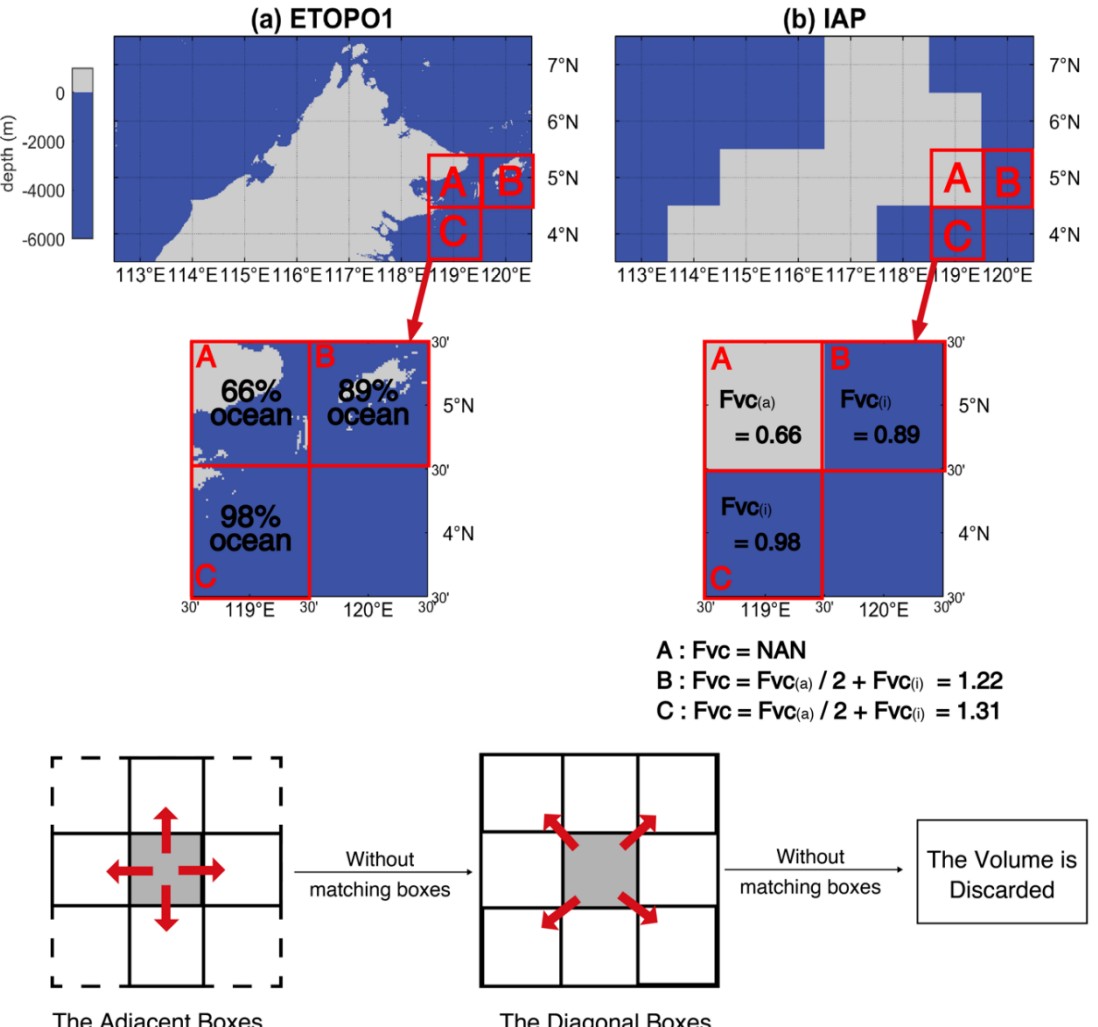


**Figure 5:** An example explaining the Volume Correction algorithm. (a) Bathymetry derived from ETOPO1. (b) Bathymetry in IAPv4 analysis.

**2.8 Independent datasets for comparison and evaluation**

Four Sea Surface Temperature (SST) datasets are used to evaluate the upper-most layer (1 m) of IAPv4, including Extended Reconstructed SST version 5 (ERSST5) (Huang et al., 2017); Japan Meteorological Agency Centennial Observation-Based Estimates of SSTs version 1 (COBE1) (Ishii et al., 2005), and its version 2: COBE2 (Hirahara et al., 2014); Hadley Centre Sea Ice and Sea Surface Temperature dataset (HadISST) (Rayner et al., 2003). The anomalies relative to a 2006-2020 average were computed by removing the monthly climatology. Measurements of SST are made *in situ* by means of thermometers or retrieved remotely from infrared and passive microwave radiometers on satellites (Kennedy, 2014; O'Carroll et al., 2019). Satellite SST observations began in the early

1980s. *In situ* SST observations go back to the 19[th] century and involve many different measurement methods, including wooden and later insulated metal buckets to collect water samples, engine room inlet measurements, and sensors on moored and drifting buoys (Kennedy, 2014). The subsurface temperatures are collected as "profiles" which contain multiple measurements at discrete vertical levels. Because of the differences in observation systems, SSTs are fundamentally different in their temporal and spatial coverage and temporal extent compared to subsurface observations on which OHC estimates rely. SST measurements also have different uncertainty sources and error structures; thus, the two systems are typically treated as independent data sources and have been used for cross-validation (Gouretski et al., 2012).

An independent in situ observation dataset in the Labrador Sea is used to evaluate IAPv4. This dataset, provided by the Bedford Institute of Oceanography (BIO) (Yashayaev, 2007; Yashayaev and Loder, 2017), includes independently validated and bias-corrected data from multi-section hydrological surveys (i.e., AR7W) in the Labrador Sea, spanning from 1896 to 2020 (this study used 1960-2020 data). These data have not been incorporated into the WOD.

The capability of the new product to close the sea level budget and the Earth's energy budget also provides tools for validation. A superior dataset should be capable of closing the sea level and the Earth's energy budgets. The total sea level change has been monitored via satellite altimetry since 1993 (from the University of Colorado https://sealevel.colorado.edu/). The ocean mass change is derived from JPL RL06.1Mv3 Mascon Solution GRACE and GRACE-FO data since 2002 (Watkins et al., 2015). For long-term total sea level change since the 1950s, we use a tide-gauge-based reconstruction (Frederikse et al., 2020). During the same period, the estimates of the Greenland ice sheet, Antarctic ice sheet, land water storage, and glacier ice melt contributions from Frederikse et al., (2020) are used to derive ocean mass change. To derive steric sea level, IAP salinity data is used (Cheng et al. 2020). The temperature and salinity data are converted to steric sea level based on the Thermodynamic Equation Of Seawater – 2010 (TEOS-10) standard (McDougall and Barker, 2011).

For the energy budget, the ice, land, and atmosphere heat content changes are from (von Schuckmann et al., 2023) from 1960 to the present. Because of the less reliable data before the 1990s for land, sea ice and ice sheets, the other set of land–atmosphere–ice data from 2005–19 is used as in Trenberth, (2022) to investigate the recent changes. The net

radiation change at the top of the atmosphere is based on Clouds and Earth's Radiant
Energy Systems (CERES) Energy Balanced and Filled (EBAF) data from Loeb et al.,
(2021) and Loeb et al., (2018) and Deep-C data from the University of Reading (Liu and
Allan, 2022; Liu et al., 2017).
Several gridded ocean T/OHC gridded products are used here for inter-comparison,
including the IAPv3 (Cheng et al., 2017), the EN4 ocean objective analysis product from
the UK Met Office Hadley Centre (Good et al., 2013); the ocean objective analysis product
(Ishii et al., 2017) (termed "ISH" hereafter) from JMA, an Argo-only gridded product from
SCRIPPS (Roemmich and Gilson, 2009) (termed "RG" hereafter), and an OHC product
based on random forest regressions (termed "RFROM" hereafter) using in situ training
data from Argo and other sources on a 7-day × 1/4° × 1/4° grid with latitude, longitude,
time, SSH, and SST as predictors (Lyman and Johnson, 2023). Several datasets available
in IPCC-AR6 (Gulev et al., 2023) are used for comparison, including: the PMEL product
from Lyman and Johnson, (2014); Machine learning based reconstruction of OHC by
Bagnell and DeVries, (2021); BOA product based on refined Barnes successive corrections
by the China Argo Real-time Data Center (Li et al., 2017); International Pacific Research
Center (IPRC) (2005-2020), von Schuckmann and Le Traon 2011 (KvS11); Green function
based OHC estimate derived from SST (Zanna et al., 2019).

## 2.9 Trend calculation and uncertainty estimates

The trends in this study have been estimated by a LOWESS approach (Cheng et al.,
2022b), i.e., we apply a locally weighted scatterplot smoothing (LOWESS) to the time
series (25-year window, equal to an effective 15-years smoothing), and then the OHC
difference between the first and the end year is used to calculate the trend. This approach
provides an effective method to quantify the local trend by minimizing the impact of year-
to-year variability and start/end points.
Throughout this paper, the 90 % confidence interval is shown. The uncertainty of
trend also follows the approach in Cheng et al., (2022a) based on a Monte Carlo
simulation. First, a surrogate OHC series is formed by simulating a new residual series
(after removing the LOWESS smoothed time series) based on the AR(1) process and
adding it to the LOWESS line. Then a LOWESS trendline is estimated for each surrogate.
This process is repeated 1000 times, and 1000 trendlines are available. The 90 %
confidence interval for the trendline is calculated based on ± 1.65 times the standard
deviation of all 1000 trendlines of the surrogates. Secondly, the uncertainty in the rate of
the OHC is estimated by the 1000 LOWESS trendlines: 1) calculating the rate based on the
difference between the first and last annual mean value of the LOWESS trendline in a
specific period; 2) calculating ± 1.65 times the standard deviation of the 1000 rate values.

**3. Results**
**3.1 Climatological annual cycle**
The annual cycle of the OHC above 2000 m of IAPv4 is compared with IAPv3, ISH,
EN4, RG and RFROM (Fig. 6 and Fig. 7) for 2006–2020. There is a consistent annual
cycle among different datasets for the global and hemispheric oceans. Globally, the ocean
releases heat from boreal spring to autumn and accumulates heat from boreal autumn to
spring, which is dominated by the southern hemisphere due to its larger ocean surface area
(Fig. 6). The two hemispheres show opposite annual variations in OHC, associated with
the annual change of solar radiation and different distribution of land and sea. For the
global OHC above 2000 m, IAPv4 shows a positive peak in April and a dip in August,
with the magnitude of OHC variation of 60.4 ZJ for IAPv4 (66.9 ZJ for IAPv3), consistent
with other datasets: 53.2 ZJ for ISH, 58.1 ZJ for EN4, 69.2 ZJ for RG and 56.6 ZJ for
RFROM (where 1 ZJ = $10^{21}$ J).
There are some unphysical variations in the OHC annual variations for IAPv3 (blue
lines). For example, the global OHC shows large spikes in January and December, and a
big shift from September to October, by contrast, the other three data products show much
smoother changes (Fig. 6a). The IAPv3 Arctic OHC (north of 69.5 °N) shows different
phase change compared with the other datasets together with a big shift from September to
December, and the magnitude of variability is much larger in IAPv3 than other datasets
(Fig. 6d). The improvement in IAPv4 is mainly because of the methodology
improvements: IAPv3 used 1990–2005 data to construct climatology which suffered from
errors related to sparse data coverage, use of "degree distance" instead of "km distance",
and other error sources. Therefore, the IAPv4 analysis presents a physically tenable OHC
seasonal variation.

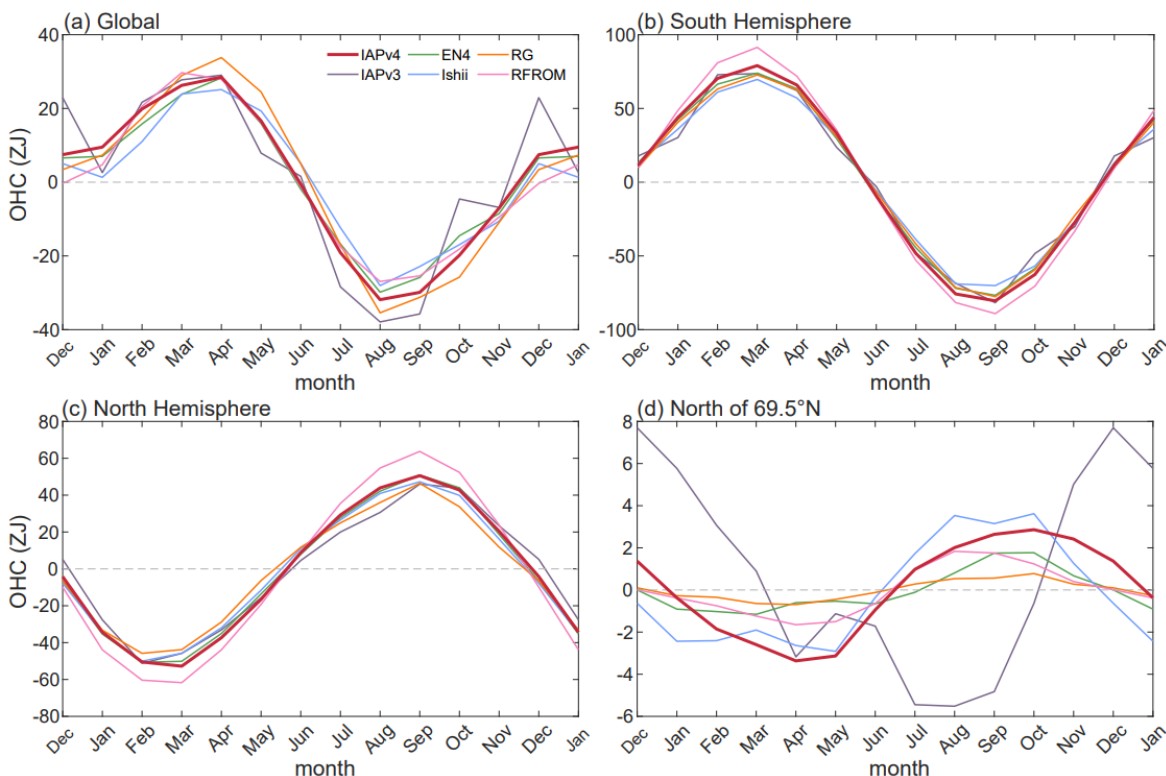

**Figure 6: Annual cycle of OHC of upper 2000 m for (a) the global oceans, (b) the Southern Hemisphere, (c) the Northern Hemisphere and (d) the oceans north of 69.5°N.** Five different data products are presented, including IAPv4 (red), IAPv3 (black), ISH (purple), EN4 (green), RG (orange), and RFROM (pink).

     IAPv4 OHC data shows significant improvements in the Arctic region, reflected in both the spatial distribution and seasonal variation of OHC. In IAPv3, the maximum upper 2000 m OHC occurs in December, and the minimum OHC occurs in August. However, for IAPv4, the maximum amounts to 2.9 ZJ in October and decreases to a minimum of −3.4 ZJ in April. This estimate of the Arctic annual cycle is consistent with a constrained Arctic OHC estimate with atmospheric data by enforcing energy budget closure (Mayer et al., 2019). Furthermore, the spread of the OHC annual cycle in the Arctic region across different datasets is reduced from 5.2 ZJ to 2.5 ZJ, indicating a smaller uncertainty. The spatial OHC anomaly distribution in the Arctic region of the IAPv4 is more spatially homogeneous than IAPv3, and IAPv3 appears as rays emerging from the pole which are not physical (Fig. 7). IAPv4 displays a consistent seasonal variation north of 69.5 °N mainly because of the changes of the influencing radius from "degrees" to "kilometers".

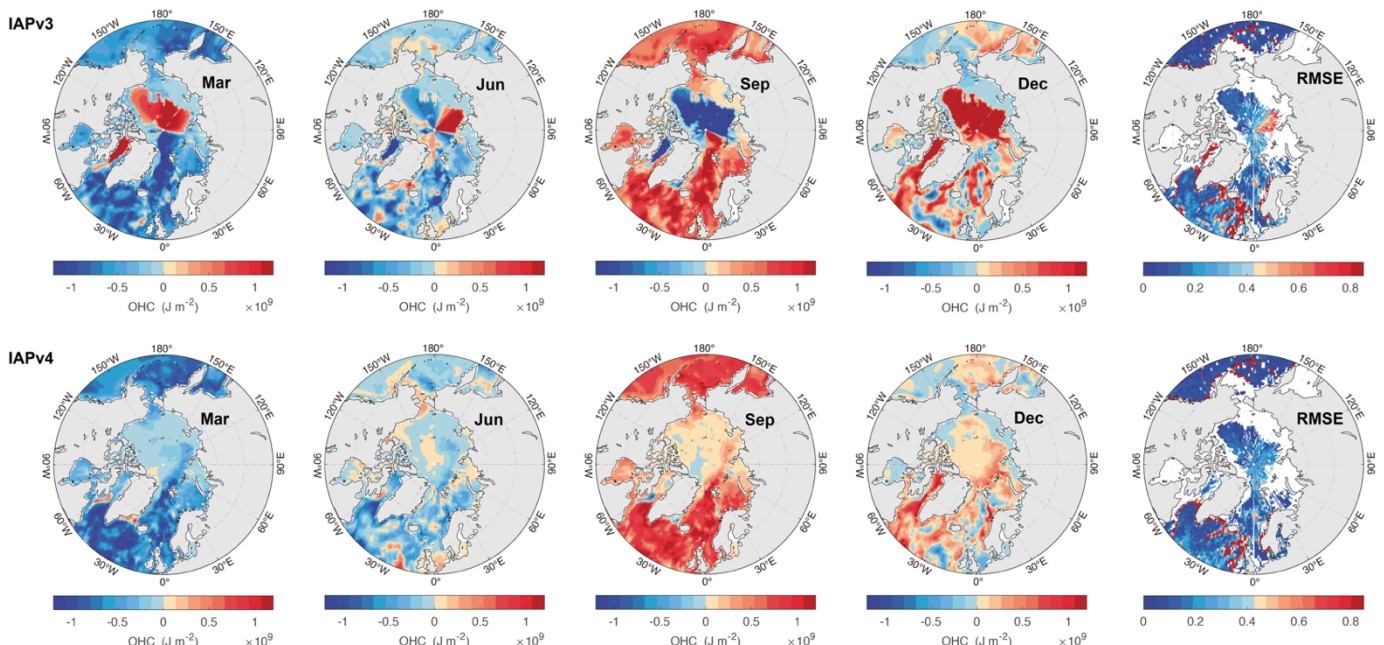

**Figure 7: Seasonal distribution of monthly mean upper 2000 m OHC anomalies and root mean square error (RMSE) of OHC 0-2000 m between gridded data and in situ observations. For OHC anomalies, four months are shown: March, June, September, and December. The OHC anomalies are relative to the 2006 – 2020 annual mean.** The upper and lower panels are for IAPv3 and IAPv4 products, respectively. The panels in the last column are for annual RMSE for IAPv3 (upper) and IAPv4 (lower), respectively.

## 3.2 Mixed layer depth

Mixed layer depth (MLD) provides a crucial parameter of upper ocean dynamics relevant for upper-deeper ocean and air-sea interactions. Spatial distributions of the MLD in March and August are shown in Fig. 8 for IAPv4, based on criteria of $\Delta T = 0.2$ °C temperature for the 10 m depth temperature. As expected, the seasonal variations of the MLD are generally opposite in the northern and southern hemispheres. The MLD shows a much stronger seasonal variation in the subtropics and midlatitudes (for example, 20°~70° in both hemispheres) than in other regions (including the tropics, for example, 20°S~20°N), which is manifested as shallower MLD (~20 m) in summer due to strong surface heating that increases stratification, and deeper MLD in winter (>70 m) because of surface cooling and increased surface wind creating stronger mixing.

In the north hemisphere, the maximum MLD occurs during the wintertime in the subpolar North Atlantic deep water formation regions (40 °N ~ 65 °N), with values over

500 m in the Iceland Basin. In comparison, in the midlatitudes, the maximum of MLD is
generally less than 125 m in the wintertime. The MLD minimum in the north hemisphere is
in the summertime, and the values are mostly within 20 m depth. In the Southern
Hemisphere, the MLD maximum values (deeper than 300 m) occur between 45 °S and
60 °S of the Southern Ocean (north of the Antarctic Circumpolar Current) in the boreal
summer where the year-round intense westerly winds are located. The minimum MLD in
this region in the boreal winter is less than 70 m. The seasonal variation of the MLD is
well established by previous studies (Chu and Fan, 2023; de Boyer Montégut et al., 2004;
Holte et al., 2017), and this evaluation confirms that IAPv4 temperature data is capable of
reasonably representing the MLD. However, as pointed out by de Boyer Montégut (2004),
the MLD estimated from the average temperature profiles might lead to an underestimation
of MLD by ~25% compared to the MLD computed from individual profiles based on the
same 0.2 °C criterion method. This potential issue needs further investigation.

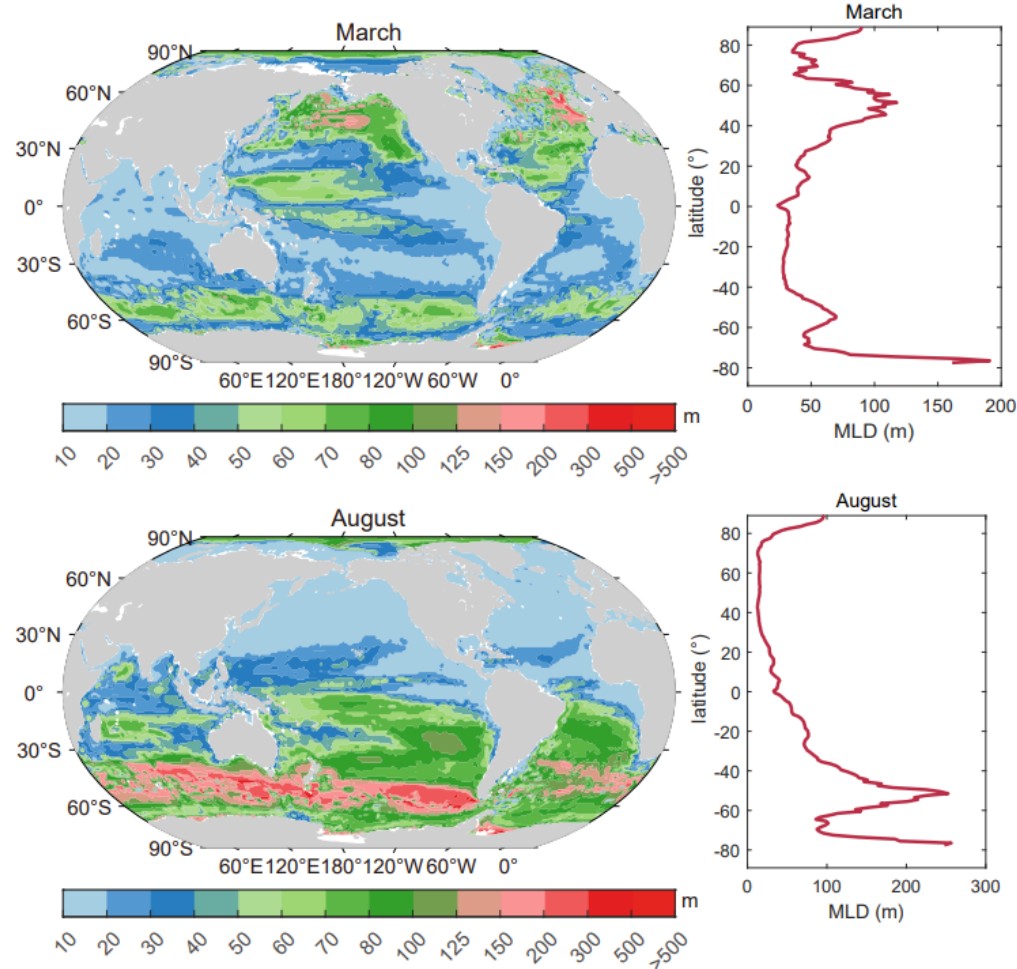


**Figure 8: Spatial pattern of the climatological mean MLD (left panels) and zonal**
**mean MLD (right panels) in March (top) and August (bottom) estimated from the**
**IAPv4.** Here, the MLD is calculated using the temperature difference criterion of $\Delta T =$
0.2 ℃ between the surface and 10-meter depth.


**3.3 Sea surface temperature**
IAPv4 and IAPv3 temperature time series at 1 m depth (Fig. 9) are compared with
four independent SST data products (ERSST5, HadISST, COBE1, and COBE2). All data
products including IAPv4 show robust sea surface warming in the global ocean and four
main basins since 1955 (Fig. 9). Since the HadISST and COBE2 data did not include the
year 2023, we compare the long-term SST trend during 1955–2022 using these products
(Fig. 9f). The global-mean IAPv4 SST rate between 1955 and 2022 is $1.01 \pm 0.15$ °C
century$^{-1}$ (90 % CI), which is within the range of the SST products (ranging from 0.78 to
1.05 °C century$^{-1}$). The 1955–2022 trend of IAPv4 SST is slightly weaker than IAPv3 for
the global ocean ($1.11 \pm 0.16$ °C century$^{-1}$) and all the ocean basins. The largest difference
between IAPv4 and other SST products comes mainly from the Pacific and the Southern
Ocean before 1980, associated with sparser in situ observations for both SST and
subsurface temperature data.
The spatial distribution of long-term SST trends over the 1955–2022 period provides
insights into the data consistencies and differences. First, IAPv4 shows a pattern of SST
consistent with other datasets (Fig. 10). More rapid warming is found in the poleward
western boundary currents regions, such as the East Australian Current and the Gulf
Stream. The warmer ocean in the upwelling areas, such as the Tropical Eastern Pacific and
Gulf of Guinea, are identified by all data products. The surface warming in the South
Indian for IAPv4 data is weaker than for IAPv3, ERSST5, and COBE2 but is more
consistent with HadISST and COBE1. The surface cooling to the south of 60 °S can also
be found in all the datasets but with some discrepancies in magnitude and locations related
to data sparsity. The tropical Pacific SST trends are mostly insignificant in the eastern and
south-eastern Pacific Ocean because of the strong inter-annual and decadal fluctuations
(Figure not shown).

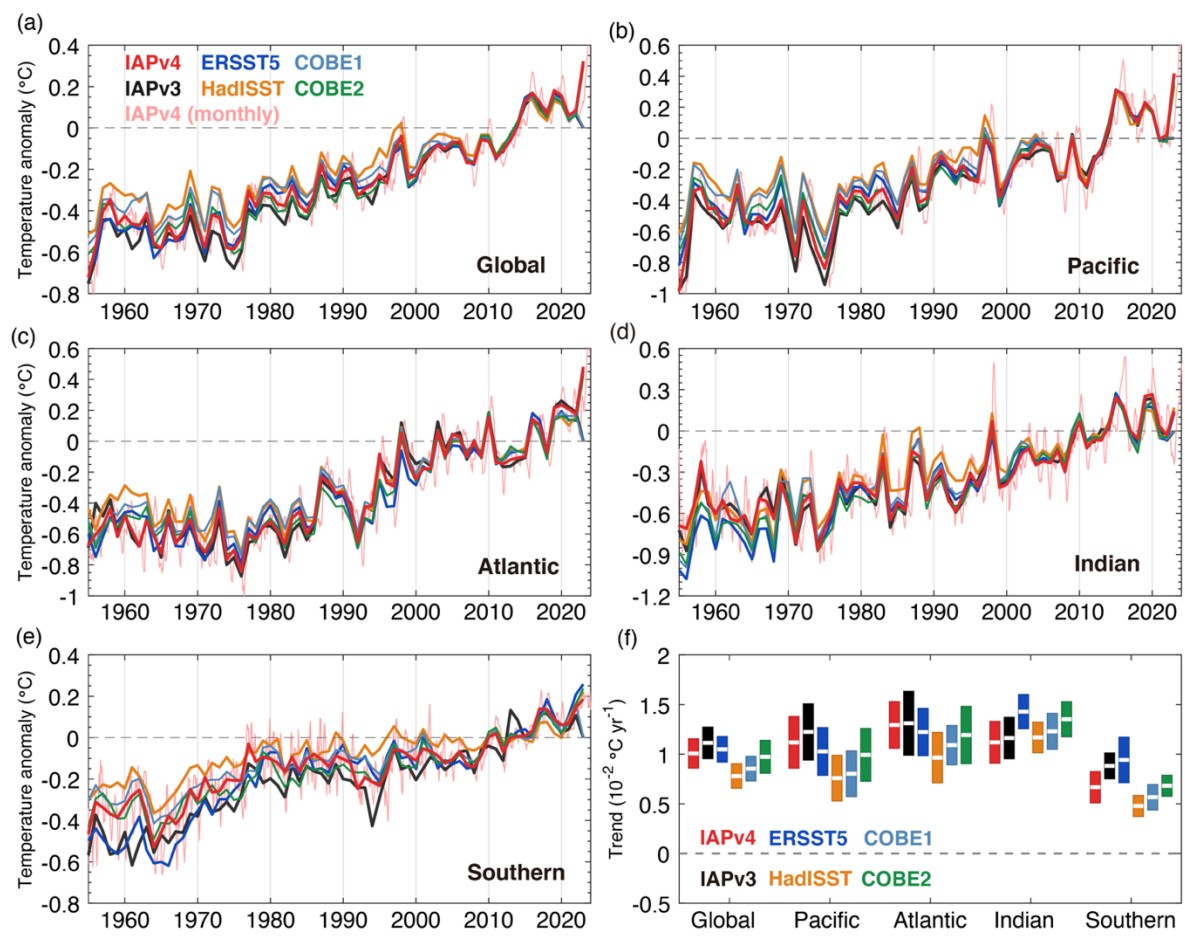

**Figure 9: Global and basin time series of SST change for IAPv4, compared with ERSST/HadISST/COBE1/COBE2 and IAPv3 from 1955 to present**. (a) Global, (b) Pacific, (c) Atlantic, (d) Indian and (e) Southern oceans (South of 30 °S) (units: °C). (f) shows the warming rate from 1955 to 2022 The pink thin line is the monthly time series of IAPv4 SST and other time series are annual time series of different datasets. The vertical scales are different for different panels. All anomaly time series are relative to a 2006–2020 baseline.

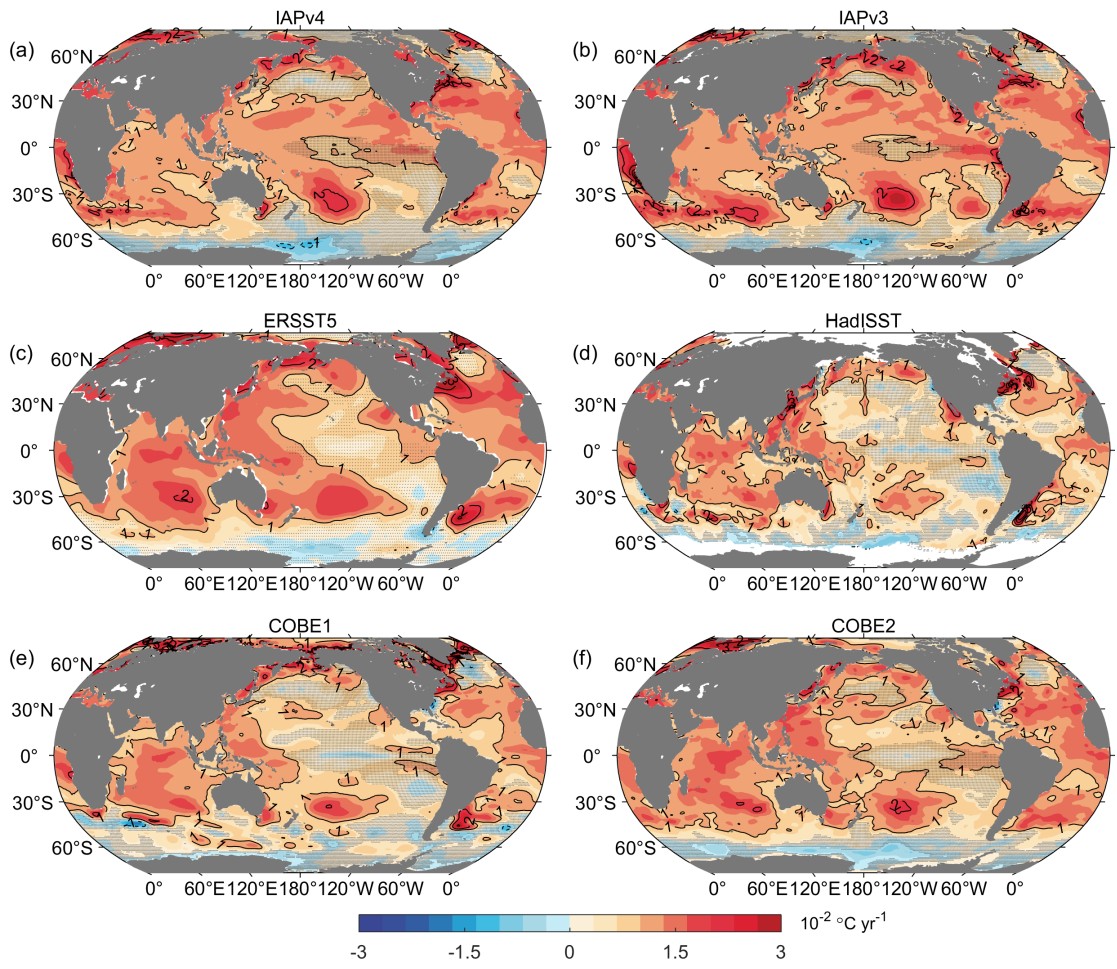

**Figure 10: Spatial maps of the SST long-term trends during the 1955–2022 period.** (a) IAPv4, (b) IAPv3, (c) ERSST5, (d) HadISST, (e) COBE1 and (f) COBE2 (units: $10^{-2}$ °C $yr^{-1}$). The contour line interval is $0.5 \times 10^{-2}$ °C $yr^{-1}$. The stippling indicates the regions with signals that are not statistically significant (90 % CI).

## 3.4 Global OHC time series

Global OHC time series for 0–700 m, 700–2000 m, 0–2000 m, and 2000-–6000 m layers of IAPv4 (Fig. 11) for 1955–2023 versus IAPv3 show a robust ocean warming, with a linear warming rate of $4.4 \pm 0.2$ ZJ $yr^{-1}$ (0–700 m), $2.0 \pm 0.1$ ZJ $yr^{-1}$ (700–2000 m), and $6.4 \pm 0.3$ ZJ $yr^{-1}$ (0–2000 m). The long-term warming revealed by IAPv4 is greater than IAPv3 ($4.1 \pm 0.2$ ZJ $yr^{-1}$ for 0–700 m, $1.9 \pm 0.1$ ZJ $yr^{-1}$ for 700–2000 m and $6.0 \pm 0.3$ ZJ $yr^{-1}$ for 0–2000 m). Before ~1980, bottle bias correction reduces the time-varying systematic warm bias in Nansen bottle data and leads to a stronger warming rate from 1955–1990. The updated MBT and XBT corrections are mainly responsible for the

difference between 1980 and 2000 (Cheng et al., 2014; Gouretski and Cheng, 2020). Data QC impacts the intra-seasonal and inter-annual variation of the OHC time series (Tan et al., 2023). Also, because of the application of Bottle/XBT/MBT corrections, IAPv4 shows a stronger upper 2000 m ocean warming trend than most of the other available products assessed in Fig. 12.

From 2005–2023, the new IAPv4 product shows stronger warming than IAPv3. The mean upper 2000 m warming rate is $10.7 \pm 1.0$ ZJ $yr^{-1}$ for IAPv4 and $9.6 \pm 1.1$ ZJ $yr^{-1}$ for IAPv3 (Fig. 11), mainly because of the replacement of the WOD-QC system by the new CODC-QC system in IAPv4. Tan et al., (2023) indicated that the WOD-QC system had removed more extreme higher temperature values in the regions of warm eddies and marine heat waves than CODC-QC. The IAPv3 700–2000 m OHC shows a much bigger drop in 2018 than IAPv4 (Fig. 11b), while the IAPv4 indicates an approximately linear 700–2000 m warming since 2005, resulting in stronger 700–2000 m warming in IAPv4 ($3.6 \pm 0.5$ ZJ $yr^{-1}$) than in IAPv3 ($2.9 \pm 0.5$ ZJ $yr^{-1}$). Compared with other available products shown in Fig. 12, IAPv4 shows a similar OHC 0–2000 m trend to RFROM from 2005–2023, but with stronger warming trends than the two Argo-based products (BOA and SCRIPPS). From 1993–2023, IAPv4 showed a stronger OHC 0–2000 m trend than NCEI, Ishii, OPEN, and Zanna data and a slightly weaker trend than PMEL and RFROM (Fig. 12).

Since the 1990s, the World Ocean Circulation Experiment (WOCE) provided a global network of abyssal ocean observations, sustained by repeated hydrological lines and a deep-Argo program (Katsumata et al., 2022; Roemmich et al., 2019; Sloyan et al., 2019). These high-quality data provide an opportunity to estimate deep OHC changes below 2000 m in this study. IAPv4 provides a new OHC estimate below 2000 m by collecting 5 years of data centered on each month. The result (Fig. 11d) indicates a robust abyssal (2000–6000 m) ocean warming trend since ~1993 of $2.0 \pm 0.3$ ZJ $yr^{-1}$. This is higher (within the uncertainty range) than the previous estimate of $1.17 \pm 0.5$ ZJ $yr^{-1}$ in Purkey and Johnson (2010) but consistent with the recent assessment showing the acceleration of deep ocean warming in the Southwest Pacific Ocean (Johnson et al., 2019).

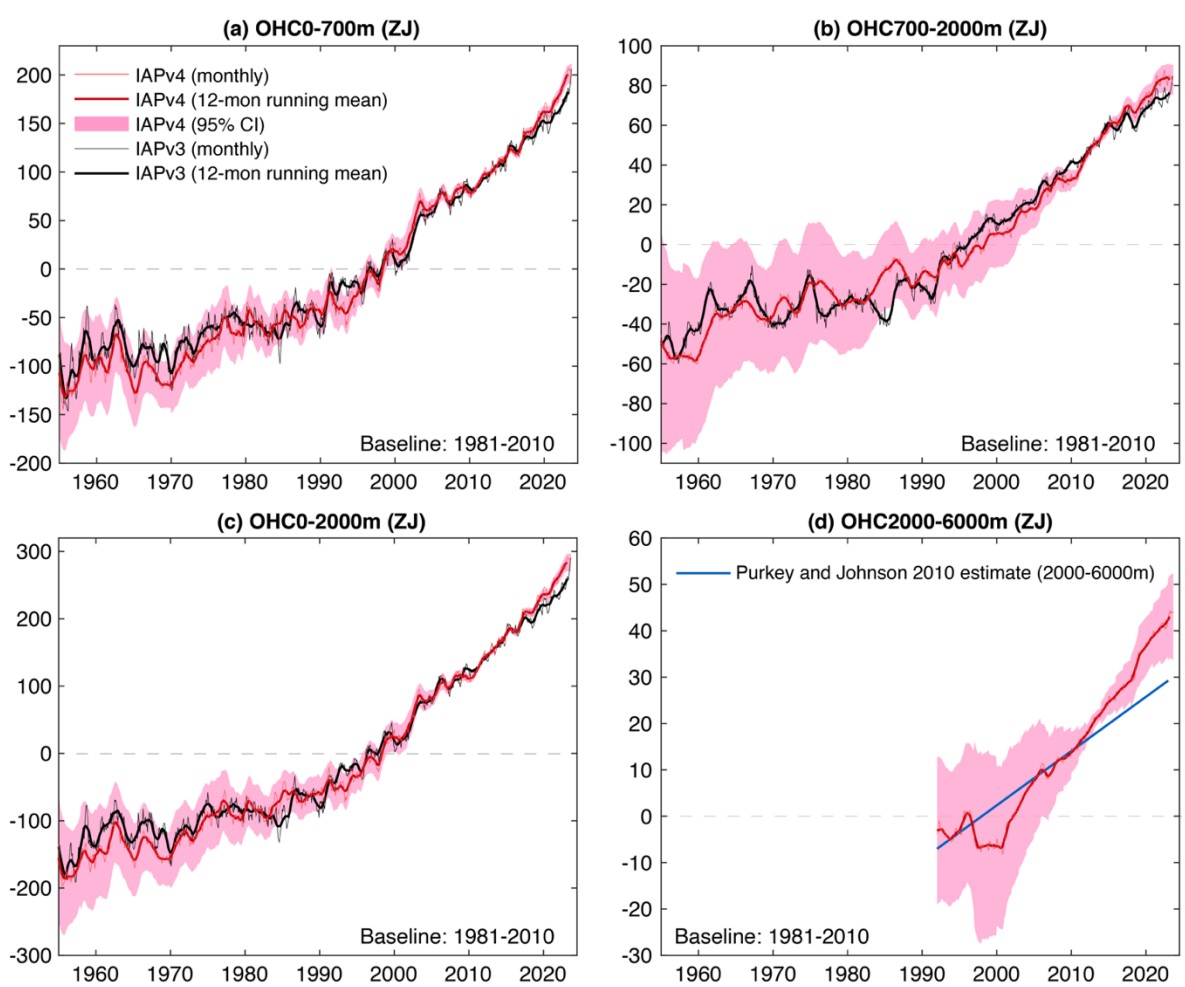

**Figure 11: Global OHC time series for 0–700 m (a), 700–2000 m (b), 0–2000 m (c) and
2000–6000 m (d).** All-time series are relative to a 1981–2010 baseline. The shading
indicates the 90 % confidence interval. The vertical scales are different for different panels.
The unit is ZJ.

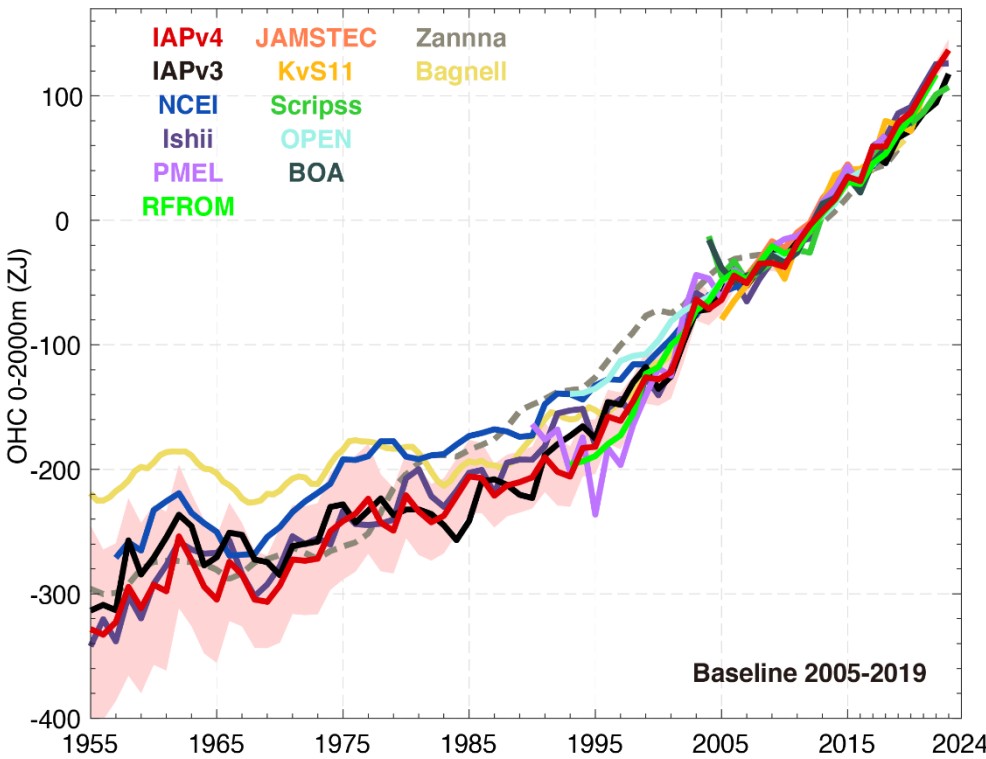

**Figure 12: A comparison of annual mean OHC 0-2000 m time series from different data products.** Solid and dashed lines represent direct and indirect estimates, respectively, and shading indicates the IAPv4 90% confidence interval (pink shading). OHC anomalies are relative to a 2005–2019 baseline. The plot is updated from Cheng et al. (2022a).

Another feature of IAPv4 is the suppression of month-to-month noise compared to many available data products. Trenberth et al. (2016) noted that the month-to-month variation (quantified by the standard deviation of the monthly dOHC/dt time series) in all *in situ*-based OHC records is much larger than implied by the CERES records, suggesting that the OHC variation on this time scale is most likely spurious. Therefore, the magnitude of the month-to-month variation in the OHC record can be used as a benchmark of the data quality. The standard deviation of the CERES record is 0.67 $Wm^{-2}$ from 2005 to 2023 (Loeb et al., 2018). While IAPv4, IAPv3, ISH, EN4, BOA, NCEI, and SIO data show a standard deviation of dOHC/dt time series of 3.52, 3.52, 7.49, 8.79, 10.05, 11.29, 10.00 $Wm^{-2}$, respectively for the upper 2000 m (Table 2). Note that differentiation to get the rate of change amplifies noise, and applying a 12-month running smoother significantly knocks down the noise so that the IAPv4 standard deviation becomes 0.75 $Wm^{-2}$, the smallest among the datasets investigated in this study (Table 2) and is the most physically plausible time series from this noise-level perspective. In addition, Lyman and Johnson's (2013) data

suggest a yearly variance ratio of 1.3 between annual RFROM and CERES data from 2008
to 2021. Using the yearly mean OHCT indicates a ratio of 1.4 at the same period between
IAPv4 and CERES, which is similar to that of RFROM.

**Table 2. Characteristics of Month-to-month variation of OHCT compared with**
**CERES.** Comparisons of different ocean gridded products: the monthly standard deviation
(std dev) of the monthly rates of change of OHC (W m$^{-2}$); the corresponding standard
deviation of the 12-month running mean (13-points are used, with start-point and endpoint
weighted by 0.5), and the linear trend with 90% confidence limits (Wm$^{-2}$) (global surface
area). The values are for 2005–2022. The OHC trend for CERES is calculated as the mean
of net TOA radiation flux within 2005–2022 multiplied by 0.9, assuming 90% of the EEI
stored in the ocean.

| Source | Std dev | Std dev (12 month) | OHC Trend (2005–2022) |
|---|---|---|---|
| IAPv4 | 3.52 | 0.75 | $0.66 \pm 0.04$ |
| IAPv3 | 3.52 | 0.79 | $0.56 \pm 0.03$ |
| ISH | 7.49 | 1.35 | $0.63 \pm 0.05$ |
| EN4 | 8.79 | 1.03 | $0.67 \pm 0.04$ |
| BOA | 10.05 | 1.16 | $0.60 \pm 0.07$ |
| NECI | 11.29 | 1.11 | $0.61 \pm 0.07$ |
| SIO | 10.00 | 1.24 | $0.56 \pm 0.08$ |
| CERES | 0.67 | 0.33 | 0.77 |


**3.5 Regional OHC trends**
For 1960–2023 (Fig. 13), the IAPv4 trends are slightly weaker than IAPv3 in the
Pacific Ocean but slightly higher in the Atlantic Ocean (Fig. 13), with more than 95 % of
the ocean area showing a warming trend. The polar regions also show remarkable
differences compared to IAPv3 (Section 3.1), mainly because of the change of covariance,
which improves the spatial reconstruction in the polar regions. The IAPv4 shows stronger
warming near the boundary currents regions, mainly because of the improved QC that does
not flag high-temperature anomalies. Nevertheless, the pattern of trends is very similar in
the two versions of data, indicating the robustness of the ocean warming pattern. The
Atlantic Ocean (within 50 °S–50 °N) and the Southern Ocean store more heat than the
other basins, probably associated with the deep convection and subduction processes
effectively transporting heat into the deep layers (Cheng et al., 2022a). The cold spots
mainly include the Northwest Pacific and subpolar North Atlantic Ocean. In particular, the
so-called "warming hole" in the subpolar North Atlantic Ocean can extend to at least 800
m and is responsible for decreased OHC in this region. Some studies have linked this
fingerprint to the slowdown of AMOC (Rahmstorf et al, 2015; Caesar et al., 2018).

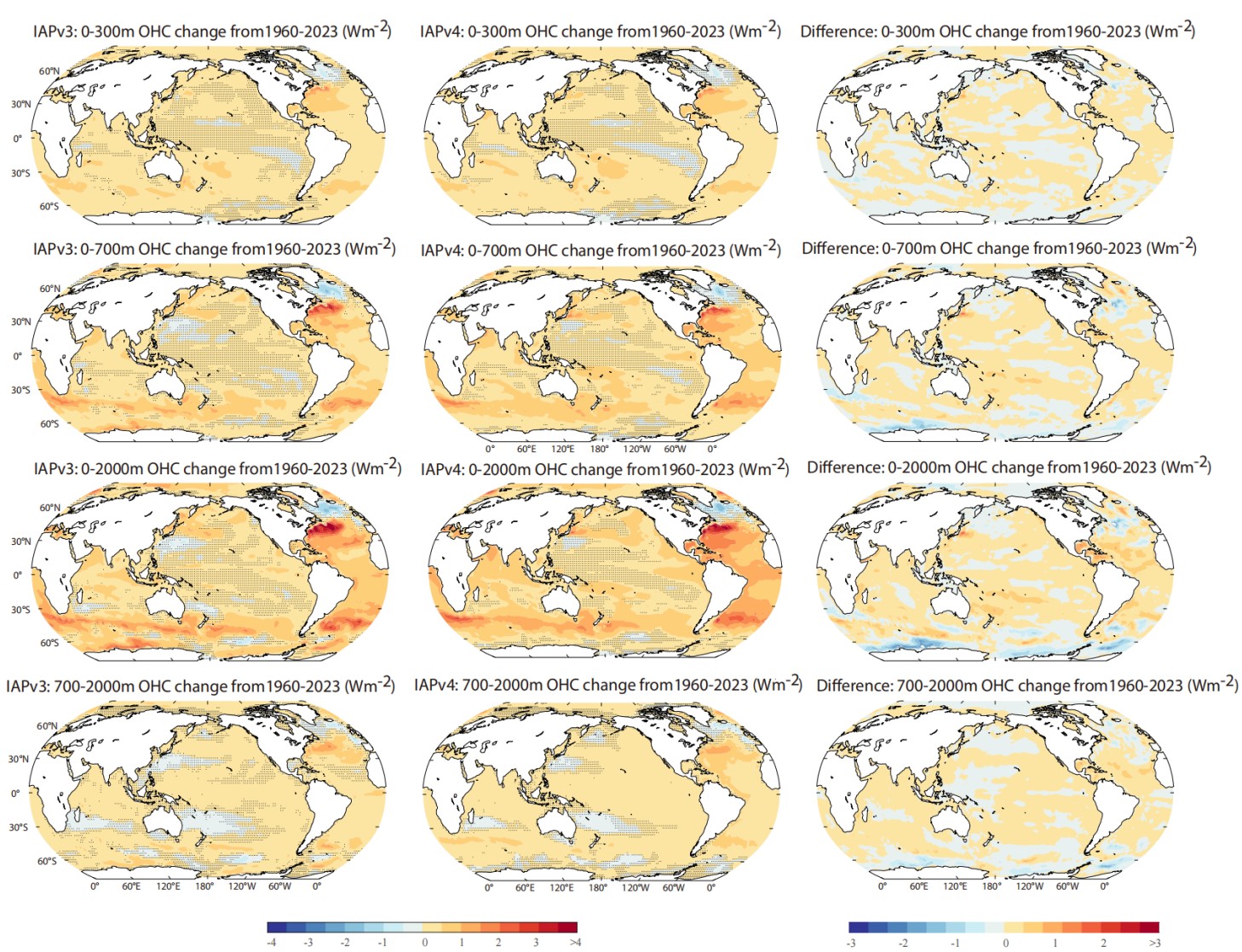

**Figure 13: Spatial pattern of the OHC trends for 0–300 m, 0–700 m and 0–2000 m, 700–2000 m from 1960 to 2023**. The left panels show IAPv3, the middle panels are IAPv4; the right panels are the difference between IAPv4 and IAPv3.

For 1991–2023 (Fig. 14), the IAPv4 and IAPv3 pattern is also consistent. A trend
pattern mimicking a negative Pacific Decadal Variability (PDV) phase appears in the Pacific
for the 0–300 m, 0–700 m, and 0–2000 m OHCs. There is a contrast between the warming
trend of the tropical western Pacific and the cooling trend of the tropical eastern Pacific.
Some studies have linked this pattern to the natural climate mode (PDV) (England et al.,
2014), but some suggest it is a forced change driven by greenhouse gas increases (Fasullo
and Nerem, 2018; Mann, 2021). Below 700 m, the 1960–2023 and 1991–2023 trend
patterns are similar because deep ocean warming mainly occurs after 1990. Broad warming
in most regions, but subtropical oceans in the West Pacific and South Indian oceans show a
cooling, which is likely related to the subtropical gyre intensification in the North but a
spin-down in the North Pacific Ocean (Zhang et al., 2014).

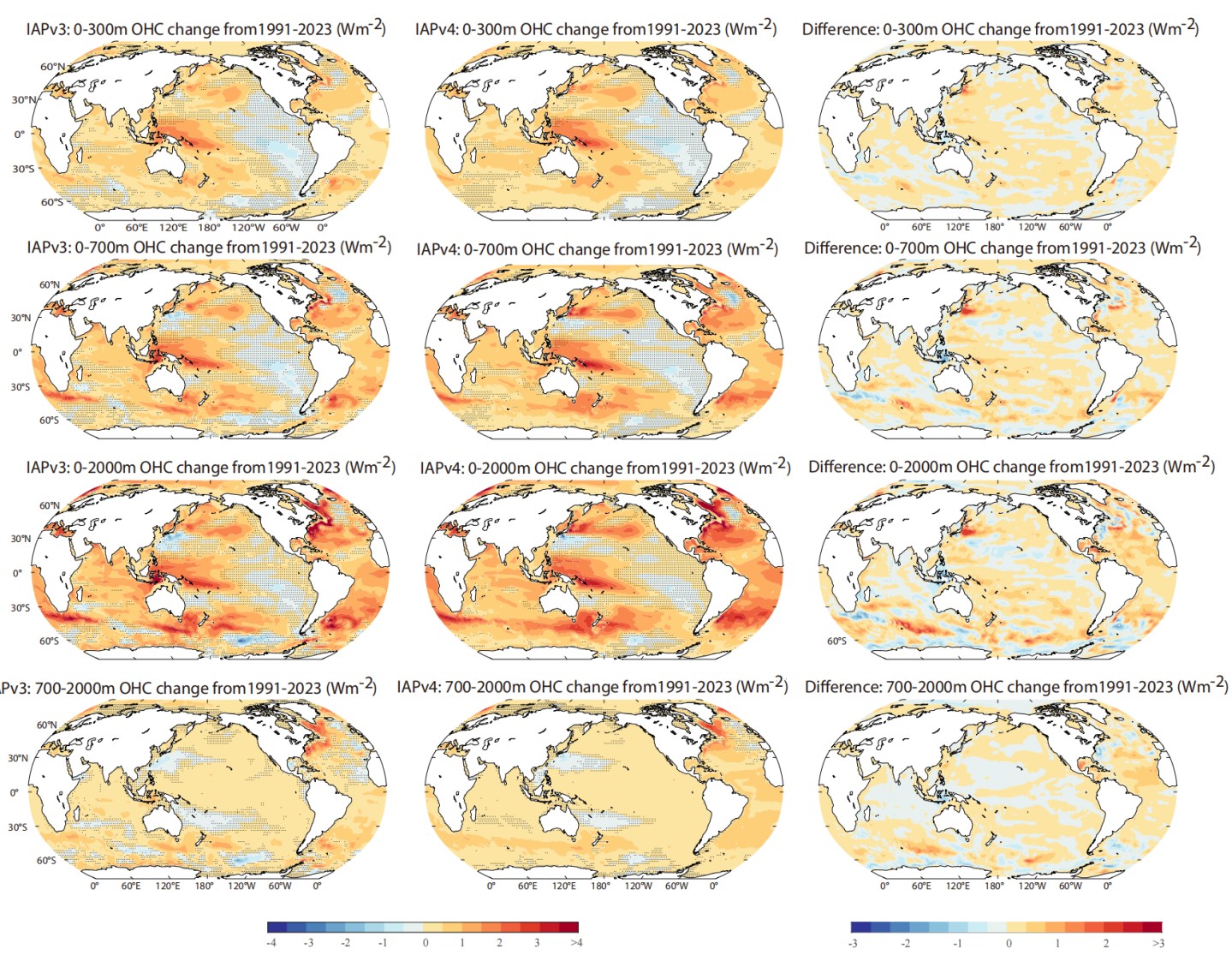

**Figure 14: Spatial pattern of the OHC trends for 0–300 m, 0–700 m, 0–2000 m and**
**700–2000 m from 1991 to 2023.** The left panels show IAPv3, the middle panels are
IAPv4; the right panels are the difference between IAPv4 and IAPv3.

Furthermore, the reconstruction of IAPv4 is compared with completely independent

observations in the central Labrador Sea (see Data and Methods section for details;
Yashayaev, 2007; Yashayaev and Loder, 2017) for the 200-2000 m mean temperature time
series (Fig. 15). The direct observations show a substantial decadal variation in the central
Labrador Sea, with negative anomalies 1970-2003 and 2015-2020, and positive anomalies
1963-1972 and 2004-2014. Reconstructed based on data from WOD, IAPv4 can well
represent this decadal variability. The largest difference occurs in 1989, where direct
observations show nearly zero anomaly while IAPv4 shows a big negative anomaly; this
difference is likely caused by using a time window in IAPv4, which has a smoothing effect
on the time series.

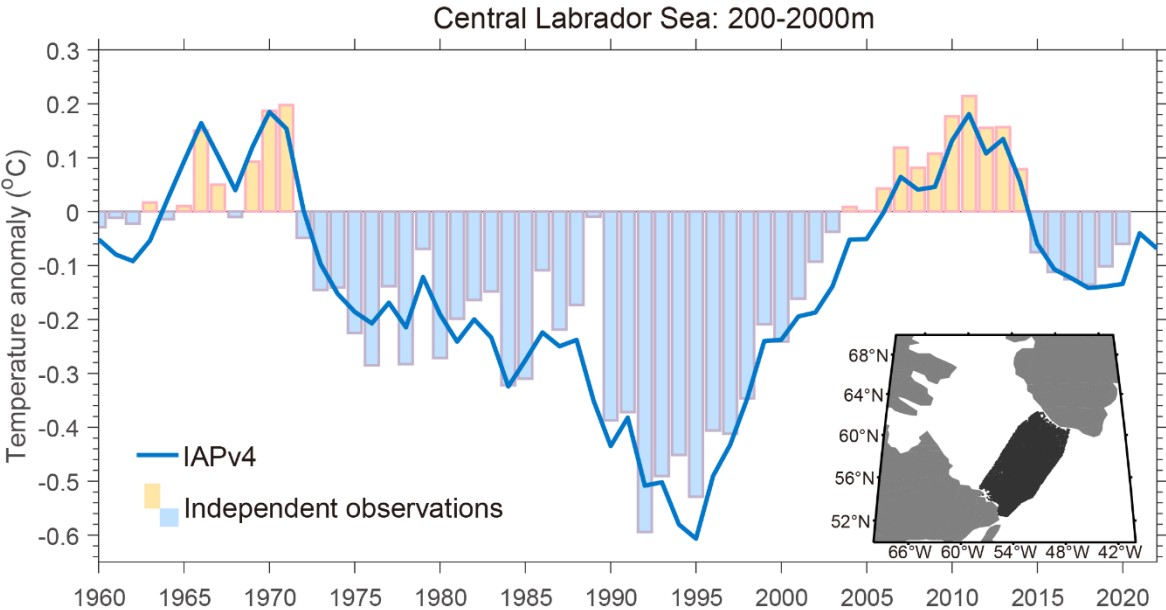


**784    Figure 15: Comparison of IAPv4 data with independent observations in the central**

**785    Labrador Sea (304 -310 °E, 55-61 °N) from 1960 to 2020.** The 200-2000 m averaged

temperature anomaly time series is shown, and the baseline is 1960-2020. The inner box
shows the locations of the independent observations in black dots (showing a total of
49,849 profiles).

**790    3.6 Ocean meridional heat transport**

The ocean meridional heat transport (MHT) is fundamental to maintaining the earth's

energy balance. Thus, its change and stability are key to the climate system and its
variability. The direct observations of ocean MHT are only possible in several cross-basin
sections such as RAPID. The ocean MHT can be derived from the OHC and air-sea heat
flux data (Trenberth and Fasullo, 2017; Trenberth et al., 2019) as follows: we integrate the
OHCT, air-sea heat flux and heat gain/loss by sea ice changes from the North Pole
southward in the Atlantic Ocean, and solve the energy budget equation, the residual at each
latitude is the MHT, i.e.,
$$MHT(\varphi) = \int_{\varphi}^{90} \left[ Fs + \frac{dOHC}{dt} + Q_{ice} \right] a\, d\varphi$$

Where $a$ is the Earth's radius, $\varphi$ is latitude, $Fs$ is net surface heat flux, and $Q_{ice}$ is
the heat inferred from the changes of sea ice mass. Consistent with Trenberth et al. (2019),
this study uses the sea ice volume data from the Pan-Arctic Ice Ocean Modeling and
Assimilation System (PIOMAS; Schweiger et al. 2011), and assumes a constant latent heat
of fusion of $3.34 \times 10^5$ J kg$^{-1}$ and a density of ice of 900 kg m$^{-3}$. Both $Fs$ and OHCT are
important for the MHT derivation: the integrated air-sea heat flux dominates the magnitude
of the MHT, while the OHCT dominates the variability of the MHT (Liu et al., 2020).
The comparison between OHC-derived MHT and RAPID data allows one to check
the consistency among various observations. Here, we calculate the Atlantic MHT from
April 2004 to December 2022 using IAPv4 OHC and air-sea net heat flux data ($F_S$) derived
by TOA net energy flux and atmospheric heat divergence (Fig. 16). $F_S$ is an average of
three available products including MAYER2021 (Mayer et al., 2021) TF2018 (Trenberth et
al., 2019) and the DEEP-C Version 5.0 from Reading University (Liu and Allan, 2022; Liu
et al., 2020). The data are adjusted following Trenberth et al. (2019) approach to ensure
zero MHT on the Antarctica coast. The inferred time series of MHT at 26.5 °N from other
OHC data sets (IAPv3, Ishii, and EN4) are also shown in Fig. 16, compared with the
RAPID observations (Johns et al., 2023).
The Inferred long-term mean (April 2004－December 2022) MHT from the updated
IAPv4 OHCT (solid red line with the mean transport of 1.18 PW) is identical to the
RAPID observation of 1.18 ± 0.19 PW. Different OHC datasets cause different inter-
annual variability in the MHT. It is shown that, from 2008 to 2020, the RAPID MHT
agrees best with the IAPv4 estimates with a correlation of 0.52. By comparison, the
correlation coefficients between RAPID and IAPv3, EN4, and Ishii are 0.33, 0.51, and
0.49, respectively. Over the entire period of 2005~2022, the IAPv4 lies mostly within the
RAPID uncertainty envelope.

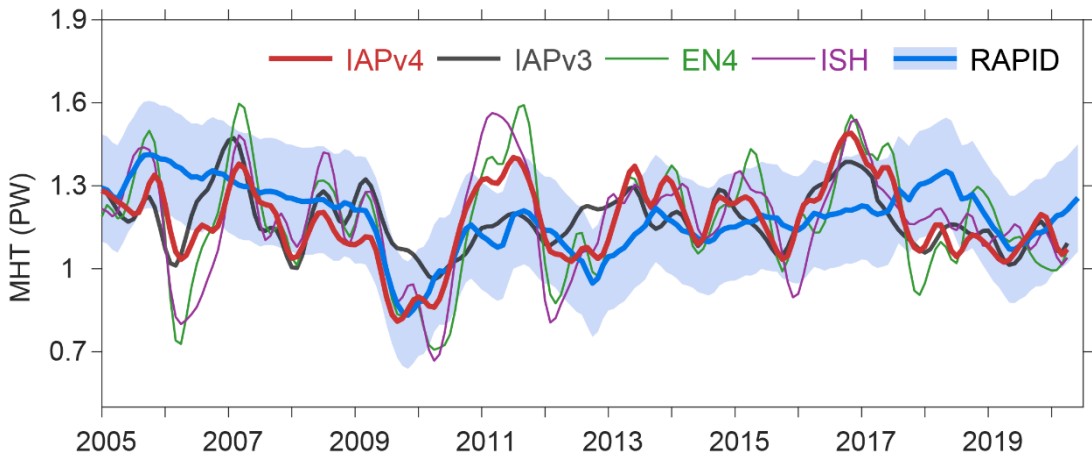


**Figure 16: Derived meridional heat transport at 26.5 °N.** The 12-month running mean northward MHT across 26.5 °N of different data sets compared with results from the RAPID array in PW. The error bars for RAPID in grey are 1.64 σ.


**3.7 Inter-annual variability**

The year-to-year variation of OHC is strongly influenced by ENSO from global to regional scales (Cheng et al., 2019; Roemmich and Gilson, 2011). To demonstrate the change of OHC associated ENSO, Figure 17 shows a Hovmöller diagram of the zonal upper 2000 m OHC and its change (time derivative of OHC: d(OHC)/dt) in the tropical Pacific Ocean from 1985 to 2023, compared with the Oceanic Niño Index (ONI). It is evident that both OHC and OHCT are closely correlated with ENSO.

Before the onset of El Niño events, there is an accumulation of heat (d(OHC)/dt > 0) in the southern and equatorial tropical Pacific ocean region (20 °S–- 5 °N). The positive tropical Pacific dOHC/dt leads ONI by ~15 months (with peak correlation >0.5), making it a precursor of El Niño (Cane and Zebiak 1985; McPhaden, 2012; Lian et al., 2023). In contrast, heat is redistributed (d(OHC)/dt < 0) from the tropical Pacific (20 °S – 5 °N) to the North Pacific (5 °N – 25 °N) during and after El Niño (Cheng et al., 2019), with a maximum correlation >0.8 at 5 months after the El Niño peak. The magnitude of the prominent change can reach up to 50 Wm$^{-2}$ during the 1997–1998 and 2015–2016 extreme El Niño events. For the other moderate El Nino events, the regional Pacific OHC change varies around 10–20 Wm$^{-2}$ (Mayer et al., 2018). This typical heat recharge-discharge paradigm is crucial in ENSO evolution (Jin, 1997). Correspondingly, the zonal OHC anomalies in the Pacific Ocean show a warming state (OHC > 0) between ~20 °N and ~5 °S before the peak of El Niño events (with peak correlation >0.7 at 5 months before El

Niño peak), followed by a period of cooling (OHC < 0) after the peak of El Niño (with peak correlation >0.7 at 12 months after El Niño peak). These variations are all physically meaningful and indicate that IAPv4 represents regional inter-annual variability, especially associated with ENSO.

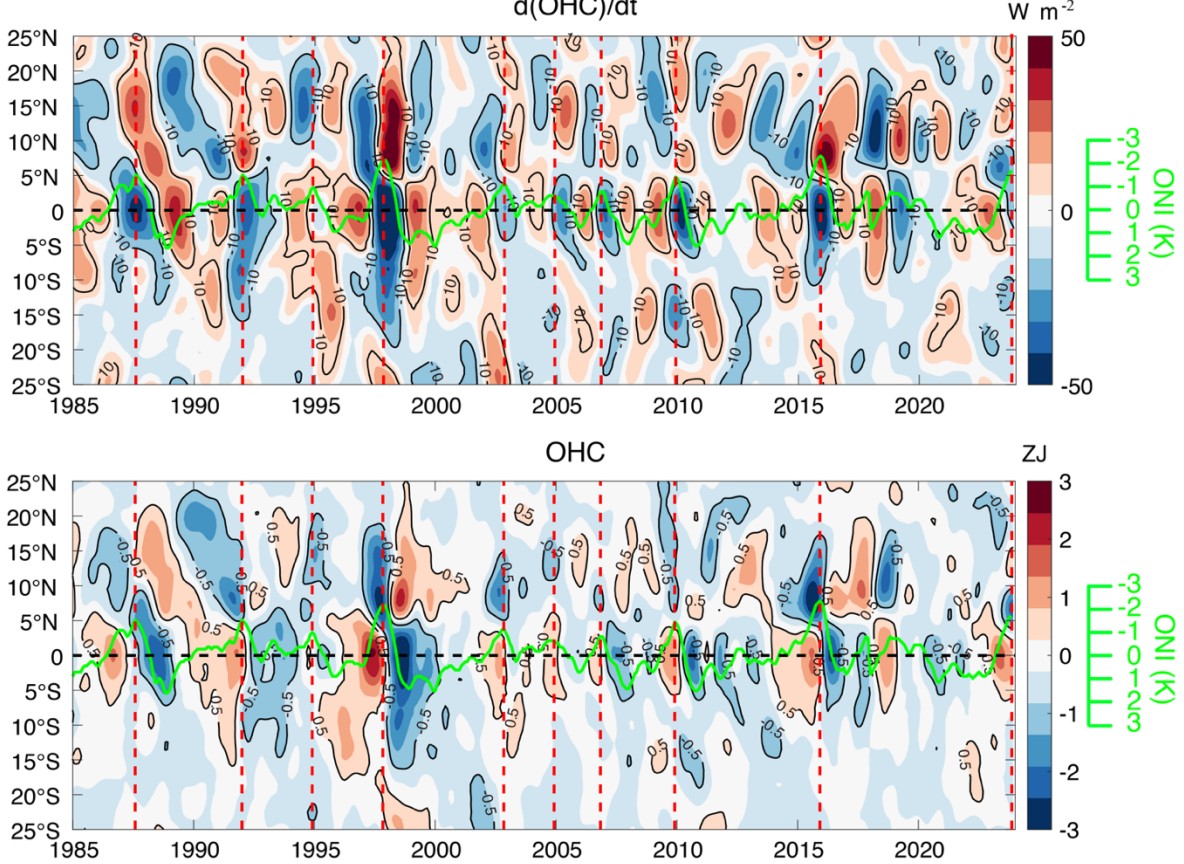

**Figure 17:** Hovmöller diagrams illustrating the zonal mean (top) upper 2000 m d(OHC)/dt (Wm$^{-2}$) and (bottom) OHC (ZJ) in each 1 ° latitude band within 25 °S ~ 25 °N in the tropical Pacific basin using IAPv4 data. The ONI is shown in green. Vertical dashed lines denote the peak time of each Niño event.

### 3.8 Ocean and Earth Energy Budget

The EEI provides a critical quantifier of the Earth's energy flow and climate change. It is also policy-relevant because it clearly shows the need to stabilize the climate system. With new T/OHC data, we re-assess the Earth's energy inventory since 1960. The land, atmosphere, and ice contributions are from the estimates obtained by von Schuckmann et al. (2023) for 1960-2023 and by Trenberth (2022) for 2015-2019.

It is evident that the earth has been accumulating heat since 1960. The Earth's heat inventory is $524.0 \pm 95.6$ ZJ from 1960 to 2023 and $260.3 \pm 25.3$ ZJ from 2005–2023 based on our data. The upper 700 m ocean, 700–2000 m, 2000 m-bottom, land, ice, and atmosphere contribute to 59.3%, 24.1%, 7.4%, 5.2%, 2.9%, and 1.1% of the total EEI, respectively, since 1960. The relative contribution has changed with time; for instance, since 1993, the contributions are 53.7% (0–700 m ocean), 24.8% (700–2000 m ocean), 12.8% (2000 m–bottom ocean), 4.1% (land), 3.2% (ice), and 1.4% (atmosphere). The land and ice contribution has increased in the recent two decades because of accelerated land and sea ice melting (Comiso et al., 2017; Hugonnet et al., 2021; Minière et al., 2024). From 2005–2019, more reliable land–atmosphere–ice datasets in Trenberth (2022) suggest a non–ocean contribution of 13.4 ZJ. Combined with the results for OHC with IAPv4, the accumulated EEI is 182.5 ZJ with the ocean heat uptake of $169.1 \pm 19.7$ for 2005–19, consistent with the value of $186.4 \pm 23.1$ ZJ using the non–ocean contribution data by von Schuckmann et al. (2023).

The derived energy inventory has been compared with satellite–based observations at the top of the atmosphere (TOA). Two comparisons are made: (1). integrate the TOA EEI to compare with the energy inventory (Fig. 18); (2) take the time derivative of the annual OHC to compare it with the TOA net radiation flux (Fig. 19). Here we always assume 90% of EEI is stored in the ocean and leads to an increase of OHC (Trenberth et al. 2009; Hansen et al., 2011; von Schuckmann et al., 2020).

The first approach avoids calculating the time derivative of OHC, which exacerbates noise in the time series. The net CERES change has been adjusted to 0.71 $Wm^{-2}$ within 2005–2015, here we adjust the trend of the integrated CERES data to the IAPv4 OHC trend to make it consistent and then compare the variability difference (Fig. 18). The RMSE between DeepC and IAPv4 is 17.9 ZJ and 15.5 ZJ between CERES and IAPv4. The comparison also indicates that the heat inventory shows a stronger heat increase from 2000 to 2005 but too slow heat accumulation during 2005–2010 compared with DeepC and CERES (Fig. 18). This might be due to the data gaps before the Argo network was fully established. DeepC and CERES show stronger heat accumulation since ~2015 than the heat inventory, probably associated with the accelerated abyssal ocean warming found by the Deep-Argo program (Johnson et al., 2019). Furthermore, IAPv4 OHC shows a slightly higher (but consistent within the uncertainty range) Earth's heat uptake compared to von

Schuckmann et al. (2023) results by 76.2 ZJ from 1960 to 2020, mainly because the
correction of Nansen bottle biases and the updates of XBT and MBT biases in IAPv4 data.
The second approach to compare OHC with satellite–based EEI is to calculate the
time derivative of OHC. To suppress the month–to–month noises, we estimate annual
OHC based on one–year data centered on June (Fig. 19a) and December (Fig. 19b)
separately, and then dOHC/dt is calculated with a forward derivative approach based on
the annual time series. The annual mean of EEI time series is also used here for
comparison (Fig. 19). The IAPv4 and CERES estimates show inter–annual variability with
a correlation of 0.44 (the correlation is statistically significant at 90% confidence level,
where autocorrelation reduction is taken into account). The higher correlation of IAPv4
versus CERES than IAPv3 increases confidence for the new data (correlation of only ~0.15
for IAPv3). The trend of dOHC/dt is 0.36 Wm$^{-2}$ dec$^{-1}$ from 2005 to 2023, within the
uncertainty range of the CERES record ($0.50 \pm 0.47$ Wm$^{-2}$ dec$^{-1}$ in Loeb et al., 2021).
However, it should be noted that the calculation of dOHC/dt is sensitive to the choices of
methods, data products, and time periods because of the noises and variability in the OHC
time series. A careful analysis of the trend of dOHC/dt (and EEI) is a research priority.

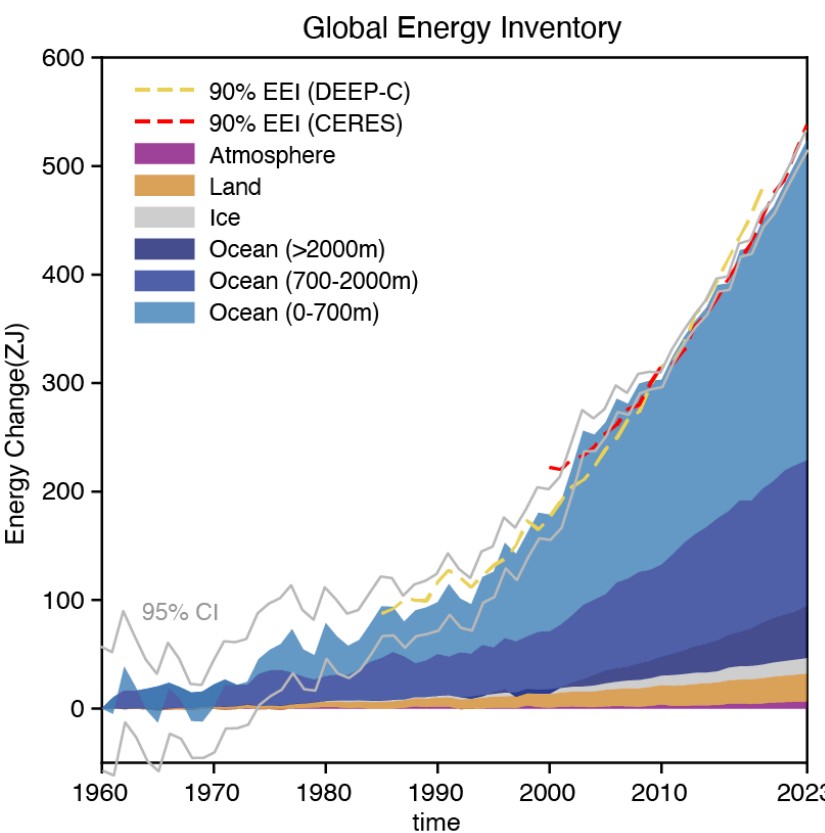


**Figure 18: The global energy budget from 1960 to 2023.** The atmosphere, land, and ice
heat inventory is from von Schuckmann et al., (2023). Integrated EEI from DEEP–C
(1985–2018) (Liu and Allan, 2022) and CERES (2001–2023) (Loeb et al., 2021) dataset
are presented by dashed lines for comparison, with the trend adjusted to the IAP estimate
to account for the arbitrary choice of integration constant. 95% Confidence Interval is
presented assuming the independency of different budget components.

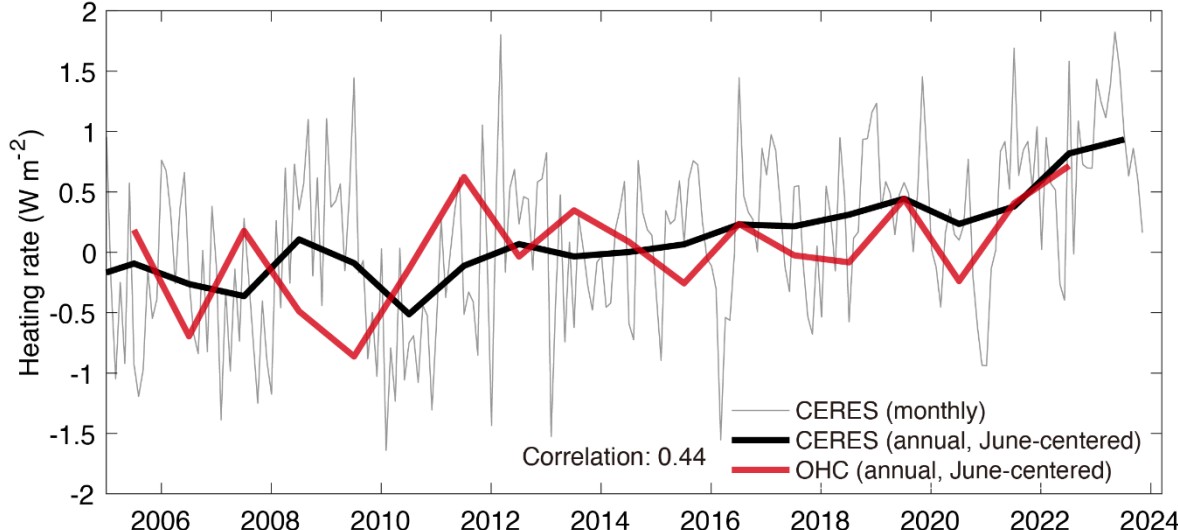


**Figure 19: Annual ocean heating rate compared with CERES data.** Both annual OHC
and CERES EEI data are centred in June. The long–term mean is removed for all-time
series.

**3.9 Steric sea level and sea level budget**
The updated IAPv4 data is used to assess the sea level budget for 1960-2023 in
combination with other data, including IAP salinity data, glaciers, Greenland, Antarctic ice
sheets mass loss from Frederikse et al. (2020) and altimetry sea level record (see Methods
section for details). From 1960 to 2023, the observed GMSL rise is 2.07 $\pm$ 0.55 mm yr$^{-1}$
(Frederikse et al., 2020), which is derived by combining tide-gauge observations with
estimates of local vertical land motion from permanent Global Positioning System stations
and the difference between tide- gauge and satellite altimetry observations (Frederikse et
al., 2018). During the same period, the sum of contributors (Glaciers, Greenland and
Antarctic ice sheets, land water storage, and steric sea level) yields a mean sea level rise of
1.87 $\pm$ 0.42 mm yr$^{-1}$. Thus, the sea level budget can be closed within a 90% confidence
interval. This updated estimate indicates that the steric sea level, Antarctic ice sheet,

Greenland ice sheet, glaciers, and land water storage contribute to the total sea level with 47.3%, 8.6%, 18.0%, 29.1%, and -3.1%, respectively for 1960-2023.

To isolate the contribution of the IAPv4 to the sea level budget, we replace the steric sea level estimate in Frederikse et al., (2020) with IAPv4 and re-assess the sea level budget for 1960-2018, 1993-2018 and 2005-2018, and the other components are identical to Frederikse et al. (2020). Two metrics are used to quantify the performance of sea level budget closure: the mean residual error and the temporal root mean square error (RMSD) between the observed GMSL and the sum of contributions. We find that the residual sea level budget based on IAPv4 is $0.20 \pm 0.53$, $0.11 \pm 0.34$, $0.47 \pm 0.56$ mm yr$^{-1}$ for 1960-2018, 1993-2018 and 2005-2018, respectively. These mean residual errors are all smaller than presented in Frederikse et al., (2020), which showed a residual error of $0.29 \pm 0.57$, $0.20 \pm 0.34$ and $0.54 \pm 0.58$ mm yr$^{-1}$ for 1960-2018, 1993-2018 and 2005-2018, respectively. The RMSD using IAPv4 (or using steric sea level in Frederikse et al., 2020) is 5.59 (5.35), 4.89 (5.33) and 4.21 (4.51) mm for the above-mentioned three periods, respectively. Therefore, both metrics show that IAPv4 data improves the sea level budget in three typical periods.

A similar test is done with the IPCC-AR6 sea level budget estimate (Gulev et al., 2021): the thermosteric sea level estimate in IPCC-AR6 is replaced by IAPv4, and the sea level budget is re-assessed for 1993-2018. IAPv4 suggests a larger thermosteric sea level rise of $1.43 \pm 0.16$ for 1993-2018 than IPCC ($1.31 \pm 0.36$ mm yr$^{-1}$) from 1993-2018. Replacing the thermosteric sea level estimate by IAPv4 reduces the mean residual error from $0.40 \pm 0.57$ to $0.28 \pm 0.48$ mm yr$^{-1}$. This suggests that the stronger warming since the 1993 revealed by IAPv4 than assessed in IPCC-AR6 (Gulve et al., 2021), seems more realistic.

After 2002, the GRACE satellite supported the direct observation of barystatic sea level, which is the sum of the sea level change due to the land water storage, Antarctica ice sheet, Greenland ice sheet, and glaciers. The sea level budget can be obtained by comparing altimetry-based GMSL with the barystatic sea level observed by GRACE and the steric sea level. It is evident that the sea level budget can be closed between 2002 and 2015 with ±5 mm residual errors (Fig. 20b). However, after ~2015, the sum of steric and barystatic sea level is smaller than the total sea level rise for all ocean temperature products. Previous studies have attributed this misclosure to salinity data biases (Barnoud et al., 2021), altimetry data errors (Barnoud et al., 2023), and GRACE data errors (Wang et

al., 2021). The steric sea level inferred from IAPv4 showed a lower residual (~5 mm)
between 2005–2023 than ISH and EN4 data (10~20 mm), indicating that the temperature
data might be partly responsible for lack of closure of sea level budget since ~2015. This
suggests again that the stronger warming in recent years, as indicated by IAPv4, is more
realistic. As discussed in Section 3.4, the QC is mainly responsible for the increased
warming of IAPv4 compared with IAPv3 since ~2015 (Fig. 11).

Many traditional QC procedures use a static climatological range check to filter out

outliers, which does not account for the increase of extreme events with climate change
and removes too many extreme (positive) values during the recent period. Thus, we
strongly recommend that data product generation groups revisit the QC procedure.
Furthermore, as the stronger long-term OHC trends since ~1960 in IAPv4 than in IAPv3
are mainly attributed to the bias corrections for Nansen Bottle, MBT, and XBT data, it is
also recommended that international groups to revisit the biases in ocean observations.

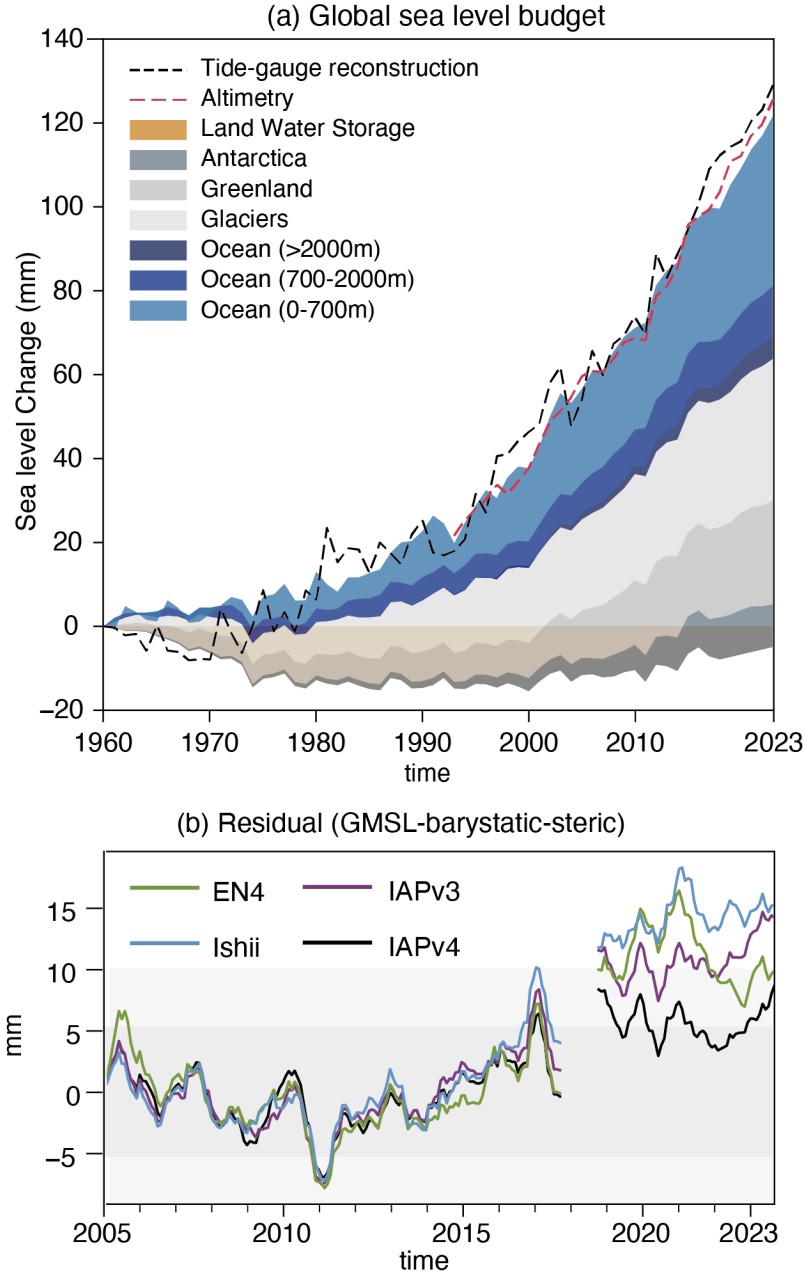


**Figure 20: (a) The sea level budget from 1960 to 2023.** Observed global mean sea level

for 1960–2023 and the individual contributions from land water storage, Antarctica,

Greenland and Glaciers (Frederikse et al., 2020). The budget is relative to a 1960 baseline.

Here, the land water storage and Glaciers data are through 2018, and a linear extrapolation

is made for 2019–2023. Antarctica ice sheet and Greenland ice sheet changes are estimated

by GRACE after 2018. Tide gauge after 2018 are updated by altimetry. Altimetry sea level

is shown in red dashed line for comparison. (b) Sea level budget residual time series since

2005. The residual of GMSL minus barystatic and steric sea level. The seasonal cycle is

reduced based on 2005–2015 climatology. 6–month running means are shown here to

reduce the noise.


**4. Data availability**

IAPv4 global ocean temperature product is available at
http://dx.doi.org/10.12157/IOCAS.20240117.002 (Cheng et al., 2024a) and
http://www.ocean.iap.ac.cn/.
IAPv4 global ocean heat content product is available at
http://dx.doi.org/10.12157/IOCAS.20240117.001 (Cheng et al., 2024b) and
http://www.ocean.iap.ac.cn/.
The code used in this paper includes data quality control, and the resultant dataset is
available at http://www.ocean.iap.ac.cn/.

The data used in this study (but not generated by this work) are listed below. IAP data are
available at http://www.ocean.iap.ac.cn/. The NCEI/NOAA data are available at
(https://www.ncei.noaa.gov/products/climate-data-records/global-ocean-heat-content). ISH
data from (https://climate.mri-jma.go.jp/pub/ocean/ts/v7.2/). The EN4 data
(https://www.metoffice.gov.uk/hadobs/en4/index.html) For SST: ERSSTv5
(https://www1.ncdc.noaa.gov/pub/data/cmb/ersst/v5/netcdf/); COBE2
(https://psl.noaa.gov/data/gridded/data.cobe2.html); and HadSST3
(https://www.metoffice.gov.uk/hadobs/hadsst3/data/download.html). For sea level data:
AVISO+ GMSL (https://www.aviso.altimetry.fr/en/data/products/ocean-indicators-
products/mean-sea-level.html#c15723), JPL GRACE (https://grace.jpl.nasa.gov/data/get-
data/jpl_global_mascons/), the data in Frederikse et al., (2020) from
(https://zenodo.org/records/3862995). The data in von Schuckmann et al., (2023)
(https://www.wdc-climate.de/ui/entry?acronym=GCOS_EHI_1960-2020). Argo data were
collected and made freely available by the International Argo Program and the national
programs that contribute to it (https://argo.ucsd.edu, https://www.ocean-ops.org). DEEP-C
data from https://doi.org/10.17864/1947.000347; CERES data (https://ceres-
tool.larc.nasa.gov/ord-tool/jsp/EBAFTOA41Selection.jsp); GIOMAS ice volume data
from (https://psc.apl.washington.edu/zhang/Global_seaice/data.html). SCRIPPS data from
(http://sio-argo.ucsd.edu/RG_Climatology.html); BOA data from
(https://argo.ucsd.edu/data/argo-data-products/).

## 5. Summary and Discussion

This paper introduces a new version of the ocean temperature and heat content gridded products and describes the data source and data processing techniques in detail. The key technical advances include the new QC, new or updated XBT/MBT/Bottle/APB bias corrections, new ocean temperature climatology, improved mapping approach, and grid-cell ocean volume corrections. These data and technical advances allow a better estimate of long-term ocean temperature and heat content changes since the mid-1950s from the sea surface down to 2000 m. We show that the new data product could better close the sea level and energy budgets than IAPv3. For rates of change, compared with CERES, the IAPv4 also shows a better correlation from 2005 to 2023 than IAPv3.

Despite several marked improvements, issues needing further investigation remain. Although inter-annual and decadal-scale changes of satellite-based EEI and observational OHC are generally consistent, a mismatch remains between EEI and OHC for their month-to-month variation, as the monthly variation of OHC is still much larger than implied by EEI. There are several possibilities, in our opinion: first, there is substantial heat storage and release for land and ice monthly, which needs to be accurately quantified; second, the accuracy of OHC estimate on a monthly basis still needs to be improved for month-to-month variation because of the limited data coverage; third, the EEI observed by CERES also suffers from sampling biases on a monthly basis (Loeb et al., 2009). Thus, a better understanding of the monthly variation of OHC and EEI is still a research priority. Besides, the failure to close the 2015-2023 sea level budget indicates that the underlying data still has bias problems, which need to be explored and resolved.

Second, the application of CODC-QC in IAPv4 leads to a stronger ocean warming rate in the past decade than WOD-QC used in IAPv3 because WOD-QC removes more positive temperature anomalies than CODC-QC. This could imply that the rate of increase in OHC is still slightly underestimated and deserves an in-depth investigation. Several fundamental questions must be answered: first, are there still real temperature extremes being removed by CODC-QC, such as in small warm/cold eddies? Are the extremes well sampled by the current observation system? If not, what is the impact? Moreover, it is clear that the high latitudes where sea ice occurs are not well sampled and need more attention.

Third, during the development of the data product, we discovered that much metadata relating to the profiles in the World Ocean Database is missing and that much existing metadata is incorrect, also giving rise to duplicate profiles, putting a strain on the overall

quality of a database of oceanic observations. More than ever, long-term concerted efforts are needed to eliminate duplicate profiles and identify and correct missing metadata using statistical methods, expert control, or machine learning techniques. For example, the International Quality-Controlled Database (IQuOD) group is coordinating some activities related to data processing techniques, uncertainty quantification, and improving the overall quality of ocean data (Cowley et al., 2021).

Fourth, the deep ocean changes below 2000 m are estimated based on the currently available data, including data from hydrological sections and Deep-Argo. IAP mapping technique is applied. Because of the lack of independent observations with global ocean coverage, evaluating the deep ocean change estimate is still dicey. Thus, the below-2000 m estimate should be used with caution, as also indicated in previous estimates (Purkey and Johnson, 2010; Desbruyères et al., 2017; Good et al., 2013). A community-agreed evaluation approach for the deep ocean changes is critically needed. Besides, other mapping techniques deserving further investigation include interpolation on isopycnal surfaces (Palmer and Haines, 2009).

Furthermore, the quantification of uncertainty for *in situ* measurements, gridded T/OHC values, and the global OHC estimates need to be improved. IAPv4 only accounts for the instrumental error and sampling/mapping error. In the future, comprehensive quantification of other uncertainty sources will be made, including the choice of climatology, vertical interpolation, XBT/MBT/APB/Bottle corrections, etc. It is also necessary to analyze the correlation between these error sources. This also helps to understand regions with larger uncertainty for OHC estimates, which supports the design of the global ocean observing system.

**Author contributions.** L.C. has worked on this study's conceptualization, coordination, methodologies, and writing the manuscript. Z.T. worked on *in situ* observation collections, metadata format, and the automated quality control procedure (CODC-QC) development. Y.P. has worked on calculating and comparing the OHC annual cycle, the mixed layer depth, and the MHT among different data sets. V.G. worked on bias correction schemes for MBT, APB, and bottle data and on developing the automated quality control procedure. H.Y. worked on the analysis of inter-annual variability. J.D. has worked on OHC trend calculation and analysis. G.L. worked on SST calculation and its analysis. H. Z. worked on global energy and sea level budget calculations and analyses. Y.L. and Y.J. worked on the

volume correction. All authors have contributed to formal analysis, data validation, and editing of the original draft.

**Acknowledgement and Funding.** The IAP/CAS analysis is supported by the National Natural Science Foundation of China (Grant no. 42122046, 42076202, 42206208, 42261134536), Strategic Priority Research Program of the Chinese Academy of Sciences (Grant no. XDB42040402), the new Cornerstone Science Foundation through the XPLORER PRIZE, DAMO Academy Young Fellow, Youth Innovation Promotion Association, Chinese Academy of Sciences, National Key Scientific and Technological Infrastructure project "Earth System Science Numerical Simulator Facility" (EarthLab), the Young Talent Support Project of Guangzhou Association for Science and Technology. The calculations in this study were carried out on the ORISE Supercomputer. Some data were collected onboard of R/V Shiyan 6 implementing the Open Research Cruise NORC2022-10+NORC2022-303 supported by NSFC Shiptime Sharing Projects 42149910. NCAR is sponsored by the US National Science Foundation. We acknowledge the World Climate Research Programme's Working Group on Coupled Modelling, which is responsible for CMIP, and we thank the climate modeling groups for producing and making their model output available through the Earth System Grid Federation. The Argo Program is part of the Global Ocean Observing System.

**Competing interests.** The contact author has declared that none of the authors has any competing interests.

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
