# Peer review of "IAPv4 ocean temperature and ocean heat content gridded 1"

_Earth System Science Data, 2024_

## Author Comment (AC2)

The authors thank reviewer#1 for your constructive comments and evaluation on our study, here the point-to-point replies are provided in blue, the comments are in **black**, the modified texts for the manuscript are shown in orange.

**Referee #1's comments and replies:**

This manuscript presents the latest edition of the IAP OHC data set, including a comprehensive description of methodological advancements and evaluation. The paper is informative for developers of similar data sets as well as users and will be a useful reference for the community. It is generally well written, but I have a number of comments and suggestions for clarification and improvement. In addition, the section on the sea level budget appears half-baked to me. That section requires more substantial revision and elaboration, or it could possibly be removed given the paper is already quite long.

Re: Thanks for the evaluation. We appreciate your helpful comments and suggestions. We have tried to address all of them. Please find the replies and the revisions introduced below. We are grateful that these comments helped improve our paper's quality.

Regarding your specific concern about the sea level budget section, we have decided to maintain it as it provides useful metrics for evaluating the dataset. However, as we agree that the material needs major work, we have rewritten it to illustrate better the impact of IAPv4 on sea level budget closure.

**Specific comments:**

L44: The authors postulate consistency of IAP OHC data with EEI. Given the only moderate correlation and the only modest visual agreement in Fig. 17 the authors should specify based on which metric they conclude "consistency".

Re: This sentence has been revised to be more specific for the consistency: "the trend of ocean warming rate (i.e., warming acceleration) is more consistent with the net energy imbalance at the top of the atmosphere than IAPv3", so it mainly refers to the trend of EEI (trend of ocean warming rate)

L328-331: I understand that even post 2005 monthly data are actually based on 3-month windows. This is important to note more explicitly. It also has implications for the variance of the time series as presented in table 2 – in fact this method will reduce the monthly variance compared to other data sets which might represent truly monthly data. Regarding time- and depth varying windows: Could this be illustrated with a time-depth Hovmoeller diagram displaying the employed window length? Also, was the impact of time-varying window length on signals assessed with synthetic data?

Re: This is a good point, instead of the Hovmoeller diagram, we included this information as Supplementary Table 1 to list the time- and depth-varying windows. A table is used instead of a figure to increase the transparency.

This method will reduce the monthly variance in temperature and OHC time series, but it is an improvement compared with other datasets (we respectfully disagree with the statement that

other data sets might represent truly monthly data). As noted in Trenberth et al. (2016) and other studies, the month-to-month in all OHC datasets explored in that study is likely spurious compared with CERES data. Boyer et al. (2016) and others (e.g. Meyssignac et al. 2019) also noted that the current observation system is still too sparse to monitor the physical month-to-month variability: too high noise level. Thus, it is desired to reduce the month-to-month variability. Besides, the increase of windows with depth is also physically meaningful because of the reduced temporal variability with depth. Furthermore, a careful investigation of inter-annual variability (ENSO) of OHC by Cheng et al. (2019) also indicates that the choices give reliable estimates of inter-annual variability compared with other data (e.g., Roemmich and Gilson 2011). A sentence added in section 2.6 "The use of a time window will reduce the monthly variance compared to other datasets, which is likely too high compared with independent Earth's Energy Imbalance data at the top of the atmosphere (Trenberth et al. 2016)."

References:

Trenberth, K. E., Fasullo, J. T., Von Schuckmann, K., and Cheng, L.: Insights into Earth's Energy Imbalance from Multiple Sources. J. Climate, 29, 7495-7505, https://doi.org/10.1175/jcli-d-16-0339.1, 2016.

Roemmich, D., and Gilson, J.: The global ocean imprint of ENSO. Geophys. Res. Lett., 38, https://doi.org/10.1029/2011GL047992, 2011.

Cheng, L., Trenberth, K. E., Fasullo, J. T., Mayer, M., Balmaseda, M., and Zhu, J.: Evolution of Ocean Heat Content Related to ENSO. J. Climate, 32, 3529-3556, https://doi.org/10.1175/jcli-d-18-0607.1, 2019.

Boyer, T., Domingues, C. M., Good, S. A., Johnson, G. C., Lyman, J. M., Ishii, M., Gouretski, V., Willis, J. K., Antonov, J., Wijffels, S., Church, J. A., Cowley, R., and Bindoff, N. L.: Sensitivity of Global Upper Ocean Heat Content Estimates to Mapping Methods, XBT Bias Corrections, and Baseline Climatologies, J. Climate, 29, 4817–4842, https://doi.org/10.1175/JCLI-D-15- 0801.1, 2016.

Meyssignac, B., Boyer, T., Zhao, Z., Hakuba, M. Z., Landerer, F. W., Stammer, D., Köhl, A., Kato, S., L'Ecuyer, T., Ablain, M., Abraham, J. P., Blazquez, A., Cazenave, A., Church, J. A., Cowley, R., Cheng, L., Domingues, C. M., Giglio, D., Gouretski, V., Ishii, M., Johnson, G. C., Killick, R. E., Legler, D., Llovel, W., Lyman, J., Palmer, M. D., Piotrowicz, S., Purkey, S. G., Roemmich, D., Roca, R., Savita, A., Schuckmann, K. von, Speich, S., Stephens, G., Wang, G., Wijffels, S. E., and Zilberman, N.: Measuring Global Ocean Heat Content to Es- timate the Earth Energy Imbalance, Front. Mar. Sci., 6, 432, https://doi.org/10.3389/fmars.2019.00432, 2019.

L343: Does that mean the influence radius changes with depth? Is this a physically based choice or is this pragmatic owing to data availability?

Re: The influencing radius changes with depth for the first iteration (2,000 km for the upper 700 m and 25,000 km at 700–6000 m), and no change for the second and third iteration (800 km and 300 km). This is based on a test provided by Cheng&Zhu (2016) paper (their Fig.3), which subsamples the recent decadal data based on past data locations to test the different choices of the influencing radius. The results show that a ~20-degree influencing radius can minimize the

reconstruction error for the upper 700m. The 700-2000m radius is further determined by Cheng et al. (2017) with the same approach.

Although it is more of a statistical result, it is physically meaningful because the spatial decorrelation distance is longer in the deeper ocean than in the upper ocean, and the decadal to multi-decadal variability of ocean temperature is generally associated with large spatial patterns.

L345-346: "real forcings" is perhaps a bit overconfident. Better say something like "reconstructed".

Re: You are right. We removed the "real forcings", so the sentence reads as follows: "For each month, IAPv3 used 40 model simulations (historical runs) from the Coupled Model Intercomparison Project phase 5 (CMIP5) to provide a flow-dependent ensemble……"

L360: Please explain E_i and M. If "E" is instrumental error, is it meant to represent a bias (which could simply be subtracted) or random error?

Re: Good point. We have included an explanation and basic assumptions for this error: "*Ei* is the instrument's precision for each individual observation, assuming random error (the basic assumption is that after bias correction, the systematic errors can be eliminated).".

Table 1: Instead of saying doi: "YES" I suggest to simply state the doi.

Re: Done, doi provided here.

L416-423: As a reader I would like to see a number of how strong the effect of the VC on OHC trends is (especially as a number from earlier works is provided).

Re: Good point. The impact of VC on global and basin OHC is tested here. Some texts have been added in the revised manuscript: "Since the open ocean accounts for the vast majority of the global ocean volume, the influence of the VC method on the global OHC trend is small. For example, the upper 2000 m OHC trend with VC is ~0.15% (~0.45%) smaller than without VC from 1958-2023 (2005-2023) for IAPv4. However, it can significantly affect regional OHC estimates, especially in regions with complex topography. For example, the Maritime Continent region's 0-2000 m OHC trend is reduced by 6.9% (4.2%) after applying VC from 1958-2023 (2005-2023) (Jin et al. 2024)."

Moreover, a previous study (Jin et al. 2024) indicated that the VC could be very important for some data products, for instance, EN4, thus this correction is recommended, I quote from Jin et al. 2014: "*The VC fixes the overestimated volume of EN4 and thereby reduces its OHC variability (Figure 1b; standard deviation adjusted from 2.50 to 1.73 ZJ), achieving a better agreement with those of IAP and Ishii (1.50 and 1.47 ZJ; Table S1 in Supporting Information S1). In this sense, the VC also reduces the uncertainty in OHC variability.*"

Reference:
Jin, Y., Li, Y., Cheng, L., Duan, J., Li, R., & Wang, F. (2024). Ocean heat content increase of the Maritime Continent since the 1990s. Geophysical Research Letters, 51, e2023GL107526. https://doi.org/10. 1029/2023GL107526

Fig. 6: Here and in other instances I suggest to bring in the Lyman and Johnson (2023; LJ2023) data (https://doi.org/10.1175/JTECH-D-22-0058.1) as they state generally improved quality over IAPv3 data. Good agreement with the LJ2023 data (which are derived in a different fashion than the IAP data) would strengthen the confidence in state of the art OHC data sets.

Re: Thanks for the suggestion; we have added Lyman and Johnson 2023 data into Fig.6 and also included some brief discussions about LJ2023 data. Lines 518-527 are rewritten

"The annual cycle of the OHC above 2000 m of IAPv4 is compared with IAPv3, ISH, EN4, RG and RFROM (Fig. 6 and Fig. 7) for 2006–2020. There is a consistent annual cycle among different datasets for the global and hemispheric oceans. Globally, the ocean releases heat from boreal spring to autumn and accumulates heat from boreal autumn to spring, which is dominated by the southern hemisphere due to its larger ocean surface area (Fig. 6). The two hemispheres show opposite annual variations in OHC, associated with the annual change of solar radiation and different distribution of land and sea. For the global OHC above 2000 m, IAPv4 shows a positive peak in April and a dip in August, with the magnitude of OHC variation of 60.4 ZJ for IAPv4 (66.9 ZJ for IAPv3), consistent with other datasets: 53.2 ZJ for ISH, 58.1 ZJ for EN4, 69.2 ZJ for RG and 56.6 ZJ for RFROM (where 1 ZJ = $10^{21}$ J)."

[Figure]

**Figure 6: Annual cycle of OHC of upper 2000 m for (a) the global oceans, (b) the Southern Hemisphere, (c) the Northern Hemisphere and (d) the oceans north of 69.5°N**. Six different data products are presented, including IAPv4 (red), IAPv3 (blue), ISH (purple), EN4 (green), RG (orange), and RFROM (pink).

Fig. 7: IAPv4 clearly looks more plausible than v3, but it is still only qualitative. Is there a way to really validate the data, e.g., with data from ice-tethered profilers?

Re: This is difficult because we merged all available data into IAPv4, so the ice-tethered profilers are not independent. Nevertheless, we have done two additional checks:

1) compare the IAPv4 data with some independent data found in the central Labrador Sea. A paragraph is added: "Furthermore, the reconstruction of IAPv4 is compared with completely independent observations in the central Labrador Sea (see Data and Methods section for details; Yashayaev, 2007; Yashayaev and Loder, 2017) for the 200-2000 m mean temperature time series (Fig. 15). The direct observations show a substantial decadal variation in the central Labrador Sea, with negative anomalies 1970-2003 and 2015-2020, and positive anomalies 1963-1972 and 2004-2014. Reconstructed based on data from WOD, IAPv4 can well represent this decadal variability. The largest difference occurs in 1989, where direct observations show nearly zero anomaly while IAPv4 shows a big negative anomaly; this difference is likely caused by using a time window in IAPv4, which has a smoothing effect on the time series."

[Figure]

**Figure 15: Comparison of IAPv4 data with independent observations in the central Labrador Sea (304-310 °E, 55-61 °N) from 1960 to 2020.** The 200-2000 m averaged temperature anomaly time series is shown, and the baseline is 1960-2020. The inner box shows the locations of the independent observations in black dots (showing a total of 49849 profiles).

2) Provide the gridded averaged observations without any interpolation and calculate the RMSE between IAPv4 and gridded averages, IAPv3 and gridded averages. This comparison is more quantitative. See the updated Fig.7. It is evident that IAPv4 shows smaller RMSE than IAPv3 in the polar regions.

[Figure]

**Figure 7: Seasonal distribution of monthly mean upper 2000 m OHC anomalies and root mean square error (RMSE) of OHC 0-2000 m between gridded data and in situ observations. For OHC anomalies, four months are shown: March, June, September, and December. The OHC anomalies are relative to the 2006 – 2020 annual mean.** The upper and lower panels are for IAPv3 and IAPv4 products, respectively. The panels in the last column are for annual RMSE for IAPv3 (upper) and IAPv4 (lower), respectively.

L575-584: Given that cooling SST trends in the eastern equatorial and south-eastern Pacific have received quite some attention recently, I suggest to discuss more potential causes of the non-existing cooling trend in that region in the IAP data.

Re: Thanks for the suggestion. We have explored a bit more the SST trends in the two regions you mentioned (Fig. X). From the time series, it appears that the tropical SST changes show substantial variability from inter-annual (mainly ENSO) to decadal (mainly PDV) scales. The cooling trends from the 1990s to the present are mostly associated with the PDV phase change. Thus, the long-term trends (e.g. from the 1950s to the present) are subtle: indeed, according to our calculation, the tropical SST trends within 1955-2020 are mostly not significant (based on our LOWESS-based approach in Cheng et al. 2023). Based on these results, we tried not to delve into the explanation of the trends in Fig.10, instead, we added a discussion suggesting that the trends are mostly not significant because of the strong inter-annual to decadal variability.

[Figure]

Fig.X2. SST time series (a, b) and trends (c) in two regions denoted in the boxes in (d). Two regions: Equatorial Eastern Pacific (EEP) and Southeast Pacific (SEP) are shown.

Fig. 11: This is a good figure, but in addition I would like to see a figure including other OHC data sets (similar to Fig. 9 for SST).

Re: A new figure has been created (Fig.12) in the revised manuscript to compare the IAPv4 time series with other data based on the plot shown in the Cheng et al. 2022 review paper. Some data have been updated to 2023. It is evident that despite the differences in datasets, IAPv4 shows a stronger long-term trend since the 1960s than almost all datasets, showing the impacts of XBT/MBT/Bottle bias corrections. Also, stronger warming occurs for IAPv4 than IAPv3 and many other data, mainly because of the QC.

[Figure]

**Figure 12: A comparison of annual mean OHC 0-2000 m time series from different data products.** Solid and dashed lines represent direct and indirect estimates, respectively, and shading indicates the IAPv4 90% confidence interval (pink shading). OHC anomalies are relative to a 2005–2019 baseline. The plot is updated from Cheng et al. (2022a).

Table 2: The annual results suggest a variance ratio between IAPv4 and CERES of >2, while Lyman and Johnson (2023) get a ratio of 1.3 for their data. This should be mentioned. A reason for this might be the fact that LJ (2023) seem to actually apply a stronger than annual smoothing (their annual OHC variation is obtained by differencing subsequent annual means) to their data, but reading your lines 792-793 it appears you are doing the same for this comparison? Needs to be clarified.

Re: We have mentioned this issue in the revised manuscript "In addition, Lyman and Johnson's (2013) data suggest a yearly variance ratio of 1.3 between annual RFROM and CERES data from 2008 to 2021. Using the yearly mean OHCT indicates a ratio of 1.4 at the same period between IAPv4 and CERES, which is similar to that of RFROM.".

We mark here that in Table 1, we have applied a 12-month running smoother to all-time series, while Lyman and Johnson (2023) used an annual time series (for their 1.3 ratio result). That's why we have made additional calculations using annual time series of IAPv4 and CERES data to derive the ratio, which is 1.4, similar to Lyman and Johnson 2023 data.

Table 2: Why is no CERES trend provided?

Re: OHC trend in CRESE record is adjusted by Johnson and Lyman dataset (0.71 Wm$^{-2}$ within 2005-2015) as described in Loeb et al., (2018). Here, we calculate the mean CERES net flux within

2005-2022, consistent with the period of other data, and assuming 90% of heat stored in the ocean. This number is added to the table as 0.77 Wm$^{-2}$. A sentence has been added in figure caption of the revised manuscript "The OHC trend for CERES is calculated as the mean of net TOA radiation flux within 2005–2022 multiplied by 0.9, assuming 90% of the EEI stored in the ocean."

Fig. 13: Please comment why the eastern Pacific cooling signal in upper 300m OHC is so much more prominent than in SST?
Re: Thanks. That is because Figure 13 is for the 1991-2023 period, but the SST figure in Fig.10 is for a longer period of 1955-2022. The previous plot, Fig. X2, shows the big inter-annual and decadal-scale variability in SST, so the trends for different time periods are different. For OHC, there is also substantial inter-annual to decadal scale variability (Fig. X3, below), so if one calculates the 1991-2023 trend, the eastern Pacific shows a negative trend for OHC0-300m.

[Figure]

Fig.X3. OHC0-300m time series in the Equatorial Eastern Pacific (EEP) and Southeast Pacific (SEP) regions in (a) and (b) respectively. (c) shows the spatial trend map for OHC0-300m from 1991 to 2022.

L695: This is only true for the tropics, not for higher latitudes.
Re: Thanks, to avoid potential issue of the quantification, this sentence is modified to "The ocean meridional heat transport (MHT) is fundamental to maintaining the earth's energy balance."

L730ff: Please add "Pacific" everywhere (also when stating correlations) to be clear you are not discussing full zonal averages
Re: Done.

L759: add "based on our data"
Re: Done.

L768: to me "EEI" is a rate of change, not an accumulated value. So maybe better to add "accumulated" before "EEI"
Re: Yes, modified.

L772-773: validation method (1) does not appear very meaningful to me, as the integrated CERES value depends on the one-time global adjustment for the EBAF product. Changing this adjustment to match the IAPv4 average OHC increase (as apparently done by the authors) does enforce the agreement seen in Fig. 16. I am unconvinced this is a meaningful approach and recommend to only keep method (2).
Re: Yes, the trend of the accumulated EEI depends on the adjustment, but the variation of the accumulated EEI does not, that's why the RMSE between accumulated EEI and IAPv4 time series is meaningful, which could be a useful metric for the EEI/OHC consistency in unit of ZJ (heat, not rate of change). This is the reason why we prefer to maintain both approaches. We believe this explanation makes things clearer.

L795: "consistency" in which sense? E.g., is the correlation significant?
Re: The sentence has been modified to "The IAPv4 and CERES estimates show inter–annual variability with a correlation of 0.44. The higher correlation of IAPv4 versus CERES than IAPv3 increases confidence for the new data (correlation of only ~0.15 for IAPv3)."

Fig. 17: The CERES series does not look de-trended.
Re: It is not de-trended. We stated in the figure caption that "The long–term mean is removed for all-time series.", so the baseline is removed, but the trend remains.

L815ff a: The methods are not clear to me. In my understanding, only steric sea level can be derived from IAPv4 data. It would be useful to note how the conversion from T/S profiles to sea level is performed? A reference would be useful.

Re: This has been made clearer in the revised manuscript: "The updated IAPv4 data is used to assess the sea level budget for 1960-2023 in combination with other data, including IAP salinity data, glaciers, Greenland, Antarctic ice sheets mass loss from Frederikse et al. (2020) and altimetry sea level record (see Methods section for details).".

And the description of the conversion from T/S to steric sea level is introduced in the Methods section: "To derive steric sea level, IAP salinity data is used (Cheng et al. 2020). The temperature and salinity data are converted to steric sea level based on the Thermodynamic Equation Of Seawater – 2010 (TEOS-10) standard (McDougall and Barkerv, 2011). "

L815ff b: It is confusing in table 3 that for all terms there is an IAPv4 entry, although everything but steric stems from other sources. Specifically, I understand that GMSL as well as "sum of contribution" (which itself should be explained better) are taken from Frederikse et al., but the values still differ. What is the reason for this? Is it only because of the different approach for trend computation? Can the authors provide the sensitivity to the trend estimation method based on IAP data?

Re: We agree that Table 3 is confusing. The sea level section has been rewritten and the Table 3 has been removed. In the revised manuscript, we have added two clean tests on the impact of IAPv4 on sea level budget closure: 1) we replace the steric sea level component in Frederikse et al. (2020) by IAPv4 and then test the residual error and RMSD; 2) we replace the thermosteric sea level component in IPCC-AR6 (Gulev et al. 2021, they do not have a halosteric sea level estimate) with IAPv4 and check the residual error and RMSD.

Because the other components are the same and the only difference is the steric sea level data, these two tests can isolate the impact of the new T/OHC data on sea level budget closure.

L815: 1991 or 1993?

Re: It is 1993, corrected.

L852-854: This is an important result, and it seems to suggest that the stronger warming in recent years as indicated by IAPv4 is more realistic. This should be stated more clearly. Also, here it would be useful to make a link to Fig. 11 or a potential new OHC figure including other OHC products.

Re: Great point, we added some discussion at the end of this paragraph "This suggests that the stronger warming in recent years, as indicated by IAPv4, is more realistic. As discussed in Section 3.4, the QC is mainly responsible for the increased warming of IAPv4 compared with IAPv3 since ~2015 (Fig. 11).

Many traditional QC procedures use a static climatological range check to filter out outliers, which does not account for the increase of extreme events with climate change and removes too many extreme (positive) values during the recent period. Thus, we strongly recommend that data product generation groups revisit the QC procedure. Furthermore, as the stronger long-term OHC trends since ~1960 in IAPv4 than in IAPv3 are mainly attributed to the bias corrections

for Nansen Bottle, MBT, and XBT data, it is also recommended that international groups to revisit the biases in ocean observations."

L916: Is there a reference for the sampling bias of CERES on monthly time scales?

Re: A reference "Loeb et al., 2009" is added. The issue is the number of samples per day to get the diurnal cycle and hourly variations in clouds. It was clearly inadequate in the early days, with 2 samples per day. After 2002 or so, there were 4 per day, but that was also inadequate for clouds. They have a major imbalance that is not physical. Part of that may be a diurnal cycle, but part of it is likely errors in angular corrections, etc. Loeb et al. (2009) stated the issue, and later, the CERES group just used OHC for adjustment, but the bias is real.

Loeb, N. G., B. A. Wielicki, D. R. Doelling, G. L. Smith, D. F. Keyes, S. Kato, N. Manalo-Smith, and T. Wong, 2009: Toward Optimal Closure of the Earth's Top-of-Atmosphere Radiation Budget. J. Climate, 22, 748–766, https://doi.org/10.1175/2008JCLI2637.1.

**Typos/edits:**
L41: suggest replacing "first" with "uppermost"

Re: Modified

L107: "support the follow-on studies on climate assessments" – This does not read very smooth. Please rewrite.

Re: Modified to "Upgrading the product with new developments is important to better support the ocean/climate research and climate assessments.".

L137: "this paper" à perhaps better say "the presented product"

Re: Modified

L222: is it a warm or cold bias?

Re: The XBT bias is generally positive but with time variation. The statement is modified to: "The XBT bias was found to be generally positive, as large as ~0.1 °C before 1980 on the global 0–700 m average, diminishing to less than 0.05 °C after 1990".

L232: "systematic biases" is redundant: either say "biases" or "systematic errors"

Re: Revised to "biases".

L243: here and in several other instances the references have the parentheses wrongly placed. Please revise.

Re: Corrected.

L270: I am not sure that "adjustive" exists

Re: Modified to "adjusted".

L282: It is unclear what you mean by "such a choice"

Re: Sentence modified to "Recent developments from other groups, such as Li et al., (2022), include the choice of a short-period climatology.".

L303-304: This alone is not necessarily a problem. But I assume salinity data are not always available?

Re: This sentence has been removed in the revised manuscript.

L435: I assume "monthly" climatology?

Re: Yes, "monthly" added here.

L462: please spell out "CERES-EBAF" once.

Re: Yes, "CERES-EBAF" spelled out here "Clouds and Earth's Radiant Energy Systems (CERES) Energy Balanced and Filled (EBAF)".

L663: Please add a reference to the subsection where this is explained

Re: Added.

L937: What is meant by "T/OC"? Do you mean T/OHC?

Re: Yes, corrected to "T/OHC"

---

## Author Comment (AC3)

The authors thank reviewer#2 for your constructive comments and evaluation on our study. Here, the point-to-point replies are provided in blue, the comments are in black, and the modified texts for the manuscript are shown in orange.

**Referee #2's comments and replies:**

The manuscript presents the description of the technical methods employed for the creation of the temperature and ocean heat content estimate IAPv4 and a basic assessment of the product in comparison to some other products. Additionally independent data such as sea level change or meridional ocean heat transport are employed to verify the product.

IAPv4 is an update with respect to its predecessor IAPv3 and a great deal of the manuscript is dedicated to the changes and impact between these different products as the reader would expect to see.

The manuscript is well written and lacks only few information. Detailed comments and suggestions are as follows:

Re: Thank you for the evaluation. We have addressed all your comments, which greatly improves the quality of this study.

L 55 I assume "based on gridded products" is more appropriate

Re: Revised.

L 78 Maybe a newer citation to point at the current product.

Re: For their gridded time series dataset, the NCEI/NOAA group did not have an updated reference, unfortunately.

L 124-125 Would it be possible to have for Fig.1a something like observed number of grid cells/months in addition to the casts? Fig.1a suggest the dominating importance of GLD while they provide very high resolution (in time and space) data which your product is not really be able to benefit from so much.

Re: Great point. We have added a new panel, Fig. 1b, to show the statistics of the observed number of grid cells by each instrument (see the figure below: Fig. R1). The new panel complements the current ones. Some texts are added in the revised manuscript "MBT, XBT, Nansen Bottle and CTD data are the major instruments before 2000 (Fig. 1a, b). The spatial coverage of these data increased to >30% in 1960 and >70% in the late 1960s for 1° × 1° × 1-year resolution. After 2005, there is a huge number of GLD and APB data, and they are mainly distributed in the polar regions (APB) and coastal regions (GLD) (Fig. 1a), their spatial coverage is usually less than 5% for 1° × 1° × 1 year resolution. By contrast, the Argo data cover most of the global open ocean since ~2005 (Fig. 1b)."

[Figure]

**Figure 1: (a) Yearly number of temperature casts for different instruments; (b) percentage coverage (%) of ocean data for each instrument, which is calculated by the ratio between the number of 1° × 1° × 1 year grid cells observed by each instrument and the total number of**

**ocean grids; (c) number of subsurface temperature casts in 1-degree grid box from 1940 to 2023 collected by different instruments:** CTD (Conductivity/Temperature/Depth), XBT (eXpendable BathyThermographs), MBT (Mechanical BathyThermograph), Bottle, APB (Animal mounted Pinniped Borne), PFL (Profiling Floats, i.e. Argo), GLD (Glider), MRB (Moored Buoy), and DRB (Drifting Buoy).

L 137 Which are the sources. That may be interesting to know for users that are looking for data.
Re: Thanks. We have to refer to our recent publication of the CODC-v1 dataset, where the sources of data are referred to and discussed. A dedicated paper will be published with respect to this dataset. The following texts are added in the revised manuscript "To complement the WOD with relatively less data in the Arctic and coastal regions of the Northwest Pacific, this presented product also uses data from other sources. The majority of these data are from the Chinese Academy of Sciences Ocean Science Data Center (Zhang et al., 2024), and some data are rescued from the old documents of marine survey. All these data will be publicly available. There are a total of 85,990 additional temperature profiles, about 0.50% of the data, which is expected to improve the reconstruction in these data-sparse regions (compared with IAPv3 and other products)."
Zhang, B. et al. CAS-Ocean Data Center, Global Ocean Science Database (CODCv1): temperature. Marine Science Data Center of the Chinese Academy of Science, doi:10.12157/IOCAS.20230525.001 (2024).

Fig.1 Define GLD
Re: Done: "GLD (Glider)" in the caption.

L 332-336 Often when too small influence radii are used, the anomalies may become zero and reconstructions fall back to climatology. This can be seen for instance in the earlier years of the EN3 objectively analysed fields. Do you have mechanisms to prevent this from happening, or are zero anomalies being accepted in case of lack of data. How frequent would that happen?
Re: Yes, this is often called "conservative bias" because many analyses (such as EN3, EN4) are not truly global analyses; in large data gaps, climatology (zero anomalies) is infilled, so the long-term warming trends have been under-estimated (Durack et al. 2014; Cheng et al. 2014, 2017, 2019). Durack et al. 2014 estimated that the underestimation could be 24–58% for global OHC, depending on gap-filling approaches. IAP analysis resolves this issue through several strategies (as fully described in Cheng&Zhu 2016; Cheng et al. 2017), some of them are mentioned in the texts, here just a brief introduction:
(1) A localization strategy is applied. The WOA/Ishii/EN4 method uses a radius of less than 900km. Instead, IAP uses 20 degrees for an influencing radius within 0-700m (25 degrees for 700-2000m). The large fractional coverage helps ensure that a near-complete global reconstruction can be reached, so the technique will not bias the reconstructed field toward the first-guess field in data-sparse regions.

(2) Previous products (WOA, EN4 and Ishii) have parameterized the background error correlation between two points as a function which decays in an exponential-like manner with the distance separating the points. This parameterized correlation is always isotropic, however, the covariance should be flow-dependent in the real ocean. The IAP product uses covariance from CMIP5 multi-model simulations. The models have the capability to simulate the general ocean circulation and could provide a better representation of the covariance.

(3) The use of time window to combine several months data together for a monthly estimate. Variable time windows (larger than one month) are used for monthly reconstructions to ensure a truly global analysis (Supplementary Table 1).

IAP data used the above-mentioned strategies to prevent the "conservative biases" and other errors in gap-filling process. Furthermore, a subsample test, in which subsets of data in the data-rich Argo era are co-located with locations of earlier ocean observations, is performed to quantify the sampling error. The subsample test is defined as the difference between the reconstructed and "truth fields". The truth field is taken to be a set of the gridded averaged temperature anomalies during the Argo era. Each truth field is subsampled according to the locations of historical observations and mapped to get the reconstructed fields. The IAP product is evaluated by this subsample test, showing an unbiased mean sampling error and with ocean temperature (or OHC) variability on decadal and multi-decadal timescales that can be reliably distinguished from sampling error.

L 345-350 Unclear what flow-dependent means and how the constraint with observations work, more information is needed here. How do you diagnose which type of flow is present when applying the flow dependent covariances or is this basically just done according to the location?

Re: The flow-dependency is ensured because CMIP5 model simulations are used, which can much better represent the ocean dynamics than traditional statistical Gaussian covariances (used in WOA, Ishii etc.). This is not explicitly parameterised because of the complexity of the covariances.

Optimization is achieved through an Ensemble Optimal Interpolation approach, where observations are combined with the CMIP5 model ensemble to estimate the minimum variance. The detailed formulation can be found in Cheng&Zhu 2016 and the Supplementary material of Cheng et al. 2017.

L360 What is E and i?

Re: It is our oversight. Ei is defined in the revised manuscript "*$E_i$* is the instrument's precision for each individual observation, assuming random error (the basic assumption is that after bias correction, the systematic errors can be eliminated)."

Fig 4 "Variance" probably should read standard deviation since the unit is deg C

Re: Yes, it is true; we change "The unit is degree Celsius" to "The unit is $°C^2$ ".

L 498 What is the relevance of the different land-sea distribution. Maybe you want to point to the amplitude?

Re: This is related to the ocean area/volume: with a similar surface heating rate, the larger the ocean area/volume, the more heat is input into the ocean. That is why the OHC amplitude is larger in the Southern Hemisphere than in the Northern Hemisphere (Fig. 6), and the annual cycle of the Southern Hemisphere dominates the global OHC.

L 518 Check the description: IAPv3 is black in the legend above
Re: Corrected.

L 525-527 Not clear why IAPv4 is considered less physical than IAPv3, there are clearly non-physical features in IAPv3 appearing as rays emerging from the pole
Re: Yes, it is a typo, we rewrite this sentence: "The spatial OHC anomaly distribution in the Arctic region of the IAPv4 is more spatially homogeneous than IAPv3, and IAPv3 appears as rays emerging from the pole which are not physical (Fig. 7)."

L 522 "Anomaly" maximum "change" is from Sep to Dec.
Re: This sentence has been modified to "In IAPv3, the maximum upper 2000 m OHC occurs in December, and the minimum OHC occurs in August. However, for IAPv4, the maximum amounts to 2.9 ZJ in October and decreases to a minimum of −3.4 ZJ in April."

L 536-537 Why January and July? Maximum MLD is expected to be later in the year: around March and August. Deep MLD in the Labrador Sea is surprising shallow.
Re: Yes, it should be better to present the spatial pattern of MLD in March and August. Figure 8 has been changed as follows, and the sentence in Line 536-537 has been modified to "Spatial distributions of the MLD in March and August are shown in Fig. 8 for IAPv4". The maximum MLD in the Labrador Sea in March can reach 581 m.

[Figure]

**Figure 8: Spatial pattern of the climatological mean MLD (left panels) and zonal mean MLD (right panels) in March (top) and August (bottom) estimated from the IAPv4.** Here, the MLD is calculated using the temperature difference criterion of ΔT = 0.02°C between the surface and 10-meter depth.

L 546 "Norwegian Sea", but the maximum appears to be southeast of Iceland which is in the Iceland Basin
Re: This is corrected to "Iceland Basin".

L 552-555 de Boyer Mont.gut et al., pointed out limitations of the delta T criteria. I think it is useful to acknowledged that these limitations also apply for the MLD estimate here.
Re: Yes, thanks. The sentence has been added after Line 555:
"However, as pointed out by de Boyer Montégut (2004), the MLD estimated from the average temperature profiles might lead to an underestimation of MLD by ~25% compared to the MLD computed from individual profiles based on the same 0.2°C criterion method. This potential issue needs further investigation."

L 582 to the south of
Re: Modified.

L 609 Interanual variations are also different

Re: Modified to "Data QC impacts the intra-seasonal and inter-annual variation of the OHC time series"

L 644 Which depth range is used?

Re: The depth range information is added "for the upper 2000 m"

L 679-680 Given the extend of that pattern I would rather call this a negative PDO phase related to the fact that a long warm phase ended in 1999 and since then it is mixed with somewhat more cold phases. Maybe bring this together with your following remarks about PDO

Re: Modified to "A trend pattern mimicking a negative Pacific Decadal Variability (PDV) phase appears in the Pacific for the 0–300 m, 0–700 m, and 0–2000 m OHCs."

L 687-688 They describe an intensification in the South but a spin-down in the North Pacific

Re: Great, thanks, this sentence is modified to "Broad warming in most regions, but subtropical oceans in the West Pacific and South Indian oceans show a cooling, which is likely related to the subtropical gyre intensification in the North but a spin-down in the North Pacific Ocean (Zhang et al., 2014).".

L699-701 It would be good to briefly outline how the OHC enters the estimate of the MHT, maybe also give an idea how important OHC is in comparison to Fs

Re: Yes, the information has been added here:

  "The ocean MHT can be derived from the OHC and air-sea heat flux data (Trenberth and Fasullo, 2017; Trenberth et al., 2019) as follows: we integrate the OHC and air-sea heat flux from the North Pole southward in the Atlantic Ocean, and solve the energy budget question, the residual at each latitude is the MHT, i.e.,

$$MHT(\varphi) = \int_{\varphi}^{90} \left[ Fs + \frac{dOHC}{dt} \right] a \, d\varphi$$

Where $a$ is the Earth's radius, $\varphi$ is latitude, $Fs$ is net surface heat flux. Both $Fs$ and OHC are important for the MHT derivation: the integrated air-sea heat flux dominates the magnitude of the MHT while the OHC dominates the variability of the MHT (Liu et al., 2020).

L 734-735 What does it mean released from 20S-5N to 5S-20S?  I assume "released" means to the atmosphere, otherwise it would be better to write redistributed., or do you argue is that the redistribution involves release and re-absorption?

Re: Yes it should be better to say "redistributed" because it is mainly the ocean processes.

L 771-774 Why is 90% EEI used in Fig.16? What does this discrepancy mean?

Re: 90% is used because 90% of the EEI is stored in the ocean (increasing OHC); the other 10% of net heat stored in the ocean is used to heat the atmosphere and land and melt the ice. An additional sentence is added here to explicitly state this issue ""

Table 3 What is the difference between GMSL and sum of components, how is the IAPv4 GMSL computed if not as a sum of components?

Re: We have rewritten this section and removed the Table 3. In the revised manuscript, we have added two clean tests on the impact of IAPv4 on sea level budget closure: 1) we replace the steric sea level component in Frederikse et al. (2020) by IAPv4 and then test the residual error and RMSD; 2) we replace the thermosteric sea level component in IPCC-AR6 (Gulev et al. 2021, they do not have a halosteric sea level estimate) with IAPv4 and check the residual error and RMSD.

Because the other components are the same and the only difference is the steric sea level data, these two tests can isolate the impact of the new T/OHC data on sea level budget closure.

GMSL can be observed directly by tide gauge or altimetry. The sum of components is the independent observation of the drivers of sea level rise, including steric sea level, glacier, Greenland ice sheet, Antarctica ice sheet and land water storage.

L 925 Why in particular warm eddies as opposed to cold eddies?

Re: This should include both warm and cold eddies. The sentence is rewritten to "first, are there still real temperature extremes being removed by CODC-QC, such as in small warm/cold eddies?".

Summary: Regarding methods to improve the estimate. Could you comment on the interpolation of the anomalies on isopycnal surfaces rather than depth levels, this could facilitate larger radii and better gap filling without the danger of making the solution overly smooth.

Re: This idea has been explored before by Palmer and Hains (2009). There might be some useful work to do in the future: different mapping approaches should be inter-compared to find whether one is superior. Theoretically, interpolation of anomalies on isopycnal surfaces should have a larger decorrelation length scale, but as introduced in this study (e.g. Fig.2 flow chat), reconstruction involves a lot of techniques and data processing procedures. The difficulty of this strategy would be the identification of isopycnal surfaces, which will add some uncertainty, especially in high latitudes. Nevertheless, a sentence is added in Summary section "Besides, other mapping techniques deserving further investigation include interpolation on isopycnal surfaces (Palmer and Haines, 2009)."

Palmer, M. D., and K. Haines, 2009: Estimating Oceanic Heat Content Change Using Isotherms. J. Climate, 22, 4953–4969, https://doi.org/10.1175/2009JCLI2823.1.

---

## Author Comment (AC4)

**The authors thank Dr. Catia Domingues for the extensive suggestions on the paper, here the point-to-point replies are provided in blue, the comments are in black, the modified texts for the manuscript are shown in orange.**

Major comments:

1) No dedicated section on caveats.

Why do caveats are not really discussed to inform users? For example, the gridding process (section 2.6) relies on CMIP model covariances for infilling, so this IAPv4 product (and earlier versions) are not purely based on observations. This observational-model mixed approach has circular implications for studies focused on comparison or evaluation of CMIP models, detection & attribution as well as in constraining CMIP model projections. In order words, the use of IAPv4 is not appropriate for these types of studies.

Re: We do have an extensive discussion on the remaining issues, see the last four paragraph of the final section, I copied here "**Despite several marked improvements, issues needing further investigation remain.** Although inter-annual and decadal-scale changes of satellite-based EEI and observational OHC are generally consistent, a mismatch remains between EEI and OHC for their month-to-month variation, as the monthly variation of OHC is still much larger than implied by EEI. There are several possibilities, in our opinion: first, there is substantial heat storage and release for land and ice monthly, which needs to be accurately quantified; second, the accuracy of OHC estimate on a monthly basis still needs to be improved for month-to-month variation because of the limited data coverage; third, the EEI observed by CERES also suffers from sampling biases on a monthly basis (Loeb et al., 2009). Thus, a better understanding of the monthly variation of OHC and EEI is still a research priority. Besides, the failure to close the 2015-2023 sea level budget indicates that the underlying data still has bias problems, which need to be explored and resolved.

Second, the application of CODC-QC in IAPv4 leads to a stronger ocean warming rate in the past decade than WOD-QC used in IAPv3 because WOD-QC removes more positive temperature anomalies than CODC-QC. This could imply that the rate of increase in OHC is still slightly underestimated and deserves an in-depth investigation. Several fundamental questions must be answered: first, are there still real temperature extremes being removed by CODC-QC, such as in small warm/cold eddies? Are the extremes well sampled by the current observation system? If not, what is the impact? Moreover, it is clear that the high latitudes where sea ice occurs are not well sampled and need more attention.

Third, during the development of the data product, we discovered that much metadata relating to the profiles in the World Ocean Database is missing and that much existing metadata is incorrect, also giving rise to duplicate profiles, putting a strain on the overall quality of a database of oceanic observations. More than ever, long-term concerted efforts are needed to eliminate duplicate profiles and identify and correct missing metadata using statistical methods, expert control, or machine learning techniques. For example, the International Quality-Controlled Database (IQuOD) group is coordinating some activities related to data processing

techniques, uncertainty quantification, and improving the overall quality of ocean data (Cowley et al., 2021).

Furthermore, the quantification of uncertainty for *in situ* measurements, gridded T/OHC values, and the global OHC estimates need to be improved. IAPv4 only accounts for the instrumental error and sampling/mapping error. In the future, comprehensive quantification of other uncertainty sources will be made, including the choice of climatology, vertical interpolation, XBT/MBT/APB/Bottle corrections, etc. It is also necessary to analyze the correlation between these error sources. This also helps to understand regions with larger uncertainty for OHC estimates, which supports the design of the global ocean observing system."

For the specific point of using CMIP5 models, we do not regard this technique as a caveat; instead, based on our previous studies (Cheng&Zhu, 2016, Cheng et al. 2017, 2020) and independent evaluations by many publications, this approach provides a good estimate of ocean temperature and OHC change and the estimate uncertainty is also provided. It is up to users to use this dataset or not, if the methods are transparently described, as in Section 2.6 "For each month, IAPv3 used 40 model simulations (historical runs) from the Coupled Model Intercomparison Project phase 5 (CMIP5) to provide a flow-dependent ensemble, which is then constrained by observations to provide optimized spatial covariance. IAP mapping uses model-based covariance because we argue that spatial covariance can never be satisfactorily parametrized by some simple basic functions (such as Gaussian) given its complexity. With model-based, flow-dependent, and dynamically-consistent covariance, the IAP mapping provides a more realistic reconstruction than other approaches based on Gaussian-based parameterized covariance, as evaluated by many studies (Cheng et al., 2017; Cheng et al., 2020; Dangendorf et al., 2021; Nerem et al., 2018)."

2) Although sea surface temperature was evaluated, no evaluation for the abyssal ocean was done (below 2000 m), relatively important given that this is a new aspect from previous versions and which also differ from published analyses. This could be done by subsampling the gridded data where profiles exist and compare differences.

Re: For the upper 2000 m estimate, we have subsampled the Argo-period observations which has near-global-ocean-coverage (Cheng et al. 2017) to evaluate the estimate. However, the deep ocean changes are different as we don't have data with satisfied global coverage to do this test. Therefore, this study simply applied the IAP mapping method to the deep ocean observations and gave an estimate, similar to previous studies (Purkey and Johnson, 2010, and EN4 data), just different mapping techniques are applied. There is no published method to evaluate the available deep ocean OHC estimates (i.e., subsampling the gridded data). We have mentioned this aspect in the Discussion section as a caveat to caution the users: "Fourth, the deep ocean changes below 2000 m are estimated based on the currently available data, including data from hydrological sections and Deep-Argo. IAP mapping technique is applied. Because of the lack of independent observations with global ocean coverage, evaluating the deep ocean change estimate is still an open issue. Thus, the below-2000 m estimate should be used with caution, as also indicated in previous estimates (Purkey and Johnson, 2010; Desbruyères et al. 2017; Good et

al. 2013). A community-agreed evaluation approach for the deep ocean changes is critically needed."

Other comments:

Line 51: Gridding methods are also the main source of spread among observational estimates, as found in Boyer et al. (2016) and Savita et al. (2022). Please inform the reader.
 Re: Gridded methods are to resolve the "irregular and incomplete data coverage" issue, so this aspect has been mentioned here.

Line 69: Missing Domingues et al. and Johnson et al. in the list.
Re: Good references, but as this is not a review paper, and these are just examples of a large set of literature. Please refer to the recent review paper (Cheng et al. 2022) for more comprehensive information. Domingues et al. are not referred to here because there is no T/OHC gridded fields available for this dataset, which is not comparable to other data products. The papers by the G.Johnson group are cited in some places and their data have been added in the revised manuscript for comparison; for example,

Johnson, G. C., Purkey, S. G., Zilberman, N. V., and Roemmich, D.: Deep Argo Quantifies Bottom Water Warming Rates in the Southwest Pacific Basin. Geophys. Res. Lett., 46, 2662-2669, https://doi.org/10.1098/rsta.2022.0188, 2019.

Lyman, J. M., and Johnson, G. C.: Estimating Global Ocean Heat Content Changes in the Upper 1800 m since 1950 and the Influence of Climatology Choice. J. Climate, 27, 1945-1957, https://doi.org/10.1175/JCLI-D-12-00752.1, 2014.

Lyman, J. M., Good, S. A., Gouretski, V. V., Ishii, M., Johnson, G. C., Palmer, M. D., Smith, D. M., and Willis, J. K.: Robust warming of the global upper ocean. Nature, 465, 334-337, https://doi.org/10.1038/nature09043, 2010.

Lyman, J. M., and G. C. Johnson, 2023: Global High-Resolution Random Forest Regression Maps of Ocean Heat Content Anomalies Using In Situ and Satellite Data. J. Atmos. Oceanic Technol., 40, 575–586, https://doi.org/10.1175/JTECH-D-22-0058.1.

Line135: My understanding is that the grey list is for operational centres. Profiles on that list should not be removed in your case. Please check with Argo data management team.
Re: As described by the Argo community, the "grey list" contains a list of active Argo floats that are suspected of malfunctioning (Wong et al. 2023). For example, in Argo webpage, it states "*In early 2007, it was discovered that Argo profiles from SOLO floats with FSI CTD (Argo Program WHOI) may have incorrect pressure values. The problem did not affect any other combination of instrument and sensor. In GTS TESAC messages, potentially affected instruments can be identified by instrument type 852 (SOLO FSI, see WMO Code Table 1770). Some profiles can be corrected automatically and some need additional study. The automatic fix for these profiles was instituted on 10 October, 2007. For profiles need additional attention, the float will stay on the greylist until the profiles have been fixed. While studying the pressure offset errors, a related problem was discovered in a group of WHOI/SBE profiles. For the affected WHOI/SBE instruments, all profiles*

*have been corrected and are available on the GDACS as of 14 September 2007.*" Described in (https://argo.ucsd.edu/data/data-faq/)."

Thus, although it is definitely upon the users to choose to use or not use the grey list, it is highly likely that floats on the grey lists are problematic data identified by the Argo group. Therefore, the IAP group decided to use the grey list to reduce the risk of including data from malfunctioning floats.

Wong, A. P. S., Gilson, J., and Cabanes, C.: Argo salinity: bias and uncertainty evaluation, Earth Syst. Sci. Data, 15, 383–393, https://doi.org/10.5194/essd-15-383-2023, 2023.

Line 136: Why those data are not directly available via WOD?
Re: They can be available via WOD (if the WOD group agrees) after the paper's publication (as the data will be publicly accessible, we have another publication on the way to describe these data).

Line 177: There are several definitions for extreme events. Which one are you using? Please include reference and rationale for selecting one of the various definitions.
Re: The sentence has been modified to "Local climatological ranges change with time to account for the long-term trends of ocean temperature accompanied by more frequent extreme events (e.g., Oliver et al., 2018). Previously, the use of the static local ranges tended to remove too many "extreme values" (at the tails of the temperature distributions) associated with climate change in recent years that were actually real, leading to a QC-procedure-related bias in the gridded dataset and OHC estimate (Tan et al., 2023);". The definition of the extremes is not relevant here. We did not define any extremes. It is a general description of high/low-temperature values at the tails of the temperature distribution. Please refer to Tan et al. (2023) for more information on how QC works.

Line 179: What is the reference for the real events?
Re: We added Oliver et al.2018 as an example for sea surface temperature and Sun et al. (2023) for subsurface changes.

Line 191: What is the reference for the manually QC-ed datasets?
Re: Tan et al. (2023) used Quata and WOCE datasets, and we added references here (Gouretski and Koltermann, 2004; Thresher et al., 2008).

Line 206: Has this been observed before in other publications, for example, Roquet et al?
Re: After double-checking the profiles before and after QC in 2008 below 4000 meters, and we find that this high rejection rate (~81.9%) is because of the gross errors in CTD data, not in Glider data (see Fig. R1 below). We also noted that this has not been reported before, to our knowledge, but we think it is easy to be identified by any QC system because they are apparent outliers.

Therefore, the sentence has been modified to "For example, the higher rejection rate within 2008-2009 below 4000 meters is because of the gross errors in the CTD data."

[Figure]

Fig. R1. (a) All CTD profiles in 2008. Here, the black color denotes profiles without performing any QC. The red color denotes the good profiles identified by CODC-QC. (b) is the zoom-in of (a) between 4,000 and 6,000 meters.

Line 215: It is true that Gouretski and Koltermann (2007) were the first to report on the XBT biases. However, Domingues et al. (2008) were the first to demonstrate the significant impact on the magnitude and variability of the global upper-ocean warming over multiple decades (see also AR5, ocean observations chapter).

Re: We respectfully disagree with this statement. First, Gouretski and Koltermann (2007) study is not the first to report on the XBT biases. The XBT bias has been found way back to old times around 1970 (see Cheng et al. 2016 BAMS for an overview). Second, Gouretski and Koltermann (2007) is the first to show that XBT biases have a global impact on OHC, and their abstract says "*We use a global hydrographic dataset to study the effect of instrument related biases on the estimates of long-term temperature changes in the global ocean since the 1950s. The largest discrepancies are found between the expendable bathythermographs (XBT) and bottle and CTD data, with XBT temperatures being positively biased by 0.2–0.4°C on average. Since the XBT data are the largest proportion of the dataset, this bias results in a significant World Ocean warming artefact when time periods before and after introduction of XBT are compared. Using bias-corrected XBT data we argue reduces the ocean heat content change since the 1950s by a factor of 0.62. Our estimate of the ocean heat content increase (0–3000 m) between 1957–66 and 1987–96 is 12.8·10^{22} J. Because of imperfect sampling this estimate has an uncertainty of at least 8·10^{22} J*". It is clear that it shows the global impact of XBT biases on OHC. Gouretski and Koltermann (2007) also demonstrated that un-corrected XBT data were responsible for an artificial decadal-scale variability of the global OHC time series.

As our paper is not a review paper and it is not a good place to discuss the history of XBT bias, we suggest to refer to the community papers Abraham et al. (2013), Cheng et al. (2016) and Goni et al. (2019) for more information.

Line 228: Please include comparison plots for the older and newer coefficients in the Suppl. Material, so readers can compare the differences arising from the update during the overlapping periods.

Re: The latest XBT bias correction scheme has been provided in www.ocean.iap.ac.cn/ under the label of Data Service -> New techniques. The code for implementing CH14 is also available (http://www.ocean.iap.ac.cn/ftp/images_files/XBT_cor_Matlab_code2023.zip).

Line 268: See also Boyer et al. (2016) for the impact of climatological choices.

Re: Boyer et al. (2016) added here.

Line 279: Refer to relevant figures in Rhein et al. 2013 (Suppl. Material).

Re: We are confused by this comment. This sentence in our paper stresses the inconsistency of different baselines at different locations could violate the spatial structure of the anomaly field. This aspect is not assessed in Rhein et al. (2013) (IPCC-AR5 reference). The most relevant plot in Rhein et al. (2013) is the Number of temperature profiles extending to 700 m depth in each 1° × 1° square, by decade, between 65°N and 65°S from the 1950s to 2000s. But this is only marginally relevant to our statement. Indeed, IPCC-AR5 (2013) only mentioned that the choice of climatology is one uncertainty source of OHC estimate, I quote: "*but other sources of uncertainty include the different assumptions regarding mapping and integrating UOHCs in sparsely sampled regions, differences in quality control of temperature data, and differences among baseline climatologies used for estimating changes in heat content (Lyman et al., 2010).*". On this basis, we'd like to avoid confusing readers and cite only the most relevant references here.

Line 309: How do these thresholds compare with the choices in Willis et al. 2007?

Re: I guess you are referring to Willis et al. 2008. This study used Argo data and did not provide thresholds for vertical intervals. Thus, if you can be more specific and point us to the resources, we would appreciate it.

Willis, J. K., D. P. Chambers, and R. S. Nerem, 2008: Assessing the globally averaged sea level budget on seasonal to interannual timescales. J. Geophys. Res. Oceans, 113, C06015.

Line 316: How does the distribution of depth levels in IAPv4 compare with WOA?

Re: Here are the standard levels in WOA (102 levels from 0 m to 5500 m) and IAPv4 (119 levels from 1 m to 6000m).

WOA_levels=[0,5,10,15,20,25,30,35,40,45,50,55,60,65,70,75,80,85,90,95,100,125,150,175,200,225,250,275,300,325,350,375,400,425,450,475,500,550,600,650,700,750,800,850,900,950,1000,1050,1100,1150,1200,1250,1300,1350,1400,1450,1500,1550,1600,1650,1700,1750,1800,1850,1900,1950,2000,2100,2200,2300,2400,2500,2600,2700,2800,2900,3000,3100,3200,3300,3400,3500,3600,3700,3800,3900,4000,4100,4200,4300,4400,4500,4600,4700,4800,4900,5000,5100,5200,5300,5400,5500]

IAPv4_levels=[1,5,10,15,20,25,30,35,40,45,50,55,60,65,70,75,80,85,90,95,100,110,120,130,140,150,160,170,180,190,200,220,240,260,280,300,320,340,360,380,400,425,450,475,500,525,550,575,600,625,650,675,700,750,800,850,900,950,1000,1050,1100,1150,1200,1250,1300,1350,1400,1

450,1500,1550,1600,1650,1700,1750,1800,1850,1900,1950,2000,2100,2200,2300,2400,2500,2600,2700,2800,2900,3000,3100,3200,3300,3400,3500,3600,3700,3800,3900,4000,4100,4200,4300,4400,4500,4600,4700,4800,4900,5000,5100,5200,5300,5400,5500,5600,5700,5800,5900,6000];

93 levels of the WOA and IAPv4 standard levels are the same. Most differences come from 100~400 m, where IAPv4 levels are denser than WOA, that's because we find the denser levels could help to improve the calculation of the mixed layer depth in the middle latitude regions. The set of standard levels is mostly practical, taking account of the vertical scales of variability. The standard levels are different for different groups.

Line 322: monthly "mean" climatology?
Re: Yes, "mean" added here.

Line 323: Why not median? (instead of mean)
Re: Before, the use of "median" usually aimed to minimize the impacts of (erroneous) outliers. However, after our QC, the data quality was satisfied in our case, so we used the arithmetic mean here to better define the gridded averages. And we did not find using a median significantly impacts our reconstruction.
A more fundamental question behind this choice is: given the skewed distribution of the temperatures in each 1deg grid and 1 month (i.e. not Gaussian in most places, see a statistic on skewness in Tan et al. 2023), what is the target estimate? Gridded median or gridded mean? They are different quantities. In our case, we want to estimate the gridded mean for IAPv4.

Line 327: Please include reference or evidence which shows that is physically grounded.
Re: This is a basic physics (more of a common sense) I think it is well-accepted by physical oceanographers: generally, upper ocean variability (month-to-month and inter-annual) is higher than in the deep ocean, and the deep ocean changes are generally slower than the upper ocean (because of the interactions between sea water and the "fast" atmosphere). To be more specific, this sentence is modified to "This process takes advantage of the larger persistence of anomalies (generally smaller monthly and inter-annual variability) in the deep ocean than in the upper ocean and thus is physically grounded."

Line 334: Is this procedure originally based on this reference? Should it be included?
Smith, D.M. & Murphy, J.M. (2007) An objective ocean temperature and salinity analysis using covariances from a global climate model. *Journal of Geophysical Research: Oceans*, **122**, C02022.
Re: The sentence in our paper on line 334 says "IAPv4 adopted a similar mapping approach (Ensemble Optimal Interpolation with dynamic ensemble: EnOI-DE) as in IAPv3 introduced in Cheng and Zhu (2016)". This is a description of EnOI-DE approach used for both IAPv4 and IAPv3. Thus, this sentence is not relevant to Smith&Murphy (2007), as they used an optimal

interpolation method with spatial covariance provided by a single model, so the approach is different.

Line 340: Does it account for narrow high-latitude fronts (e.g. across ACC)? Does the approach have awareness or does it mix water from two sides of fronts?

Re: The covariance is defined by the model ensemble, so if the model's simulation has fronts across ACC, it can represent them in error covariances. Here we provide an example, showing the temperature anomaly reconstruction in December 2023 at 500m (http://www.ocean.iap.ac.cn/), it is clear that the temperature anomalies along ACC can be well represented and the reconstruction is physically meaningful (I mean, describe the physics that we know).

For your second question "Does the approach have awareness or does it mix water from two sides of fronts?". I'm confused, how does any approach mix water from two sides of fronts?? I would appreciate it if you could provide specific comments on how to look into this.

[Figure]

Figure R2. Temperature anomaly reconstruction field (IAPv4) at 500m at December 2023.

Line 348: What are the implications for certain studies when the gridded estimates are not purely observational?

Re: We would appreciate your being specific on this question. The method is transparent and peer-reviewed before. We refer to our replies to your major comments for the point of combing models with observations.

Line 352: Can this approach be benchmarked via the IAPSO's ME4OH working group best practices?

Re: ME4OH is an ongoing project which is in its initial phase, and there is no general agreement on how to benchmark the mapping approach; in other words, there is no published paper on this

issue. That's why we chose not to mention this activity before it can provide a community-accepted benchmark approach.

Line 370: Also compare to Savita et al. 2022.

Re: This sentence is about a large variance of temperature in some eddy-rich regions, which is not relevant to Savita et al. 2022, which compares OHC estimates based on different approaches.

Table 1: Does the radius of influence cross ocean basins or does it have awareness? What about frontal structures, particularly in the Southern Ocean?

Re: The radius of influence does not cross the land; this is a good point, and we mentioned it in the manuscript: "The radius of influence does not cross the land."

Line 396: See also Savita et al. 2022 and Meyssignac et al. 2019 studies on the impact of ocean masks.

Re: Both references added.

Lines 403-423:   What is the difference it makes to the global values? 1%, 20%?

Re: Some quantification of the impacts on OHC is provided: " Since the open ocean accounts for the vast majority of the global ocean volume, the influence of the VC method on the global OHC trend is small. For example, the upper 2000 m OHC trend with VC is ~0.15% (~0.45%) smaller than without VC from 1958-2023 (2005-2023) for IAPv4. However, it can significantly affect regional OHC estimates, especially in regions with complex topography. For example, the Maritime Continent region's 0-2000 m OHC trend is reduced by 6.9% (4.2%) after applying VC from 1958-2023 (2005-2023) (Jin et al. 2024)."

Line 449: How can you say it is a superior dataset? Compared to what? What happens if other datasets have (compensating) issues? How do you know the other datasets used in the budget are perfect?

Re: The sentence in our manuscript is: "A superior dataset should be capable of closing the sea level and the Earth's energy budgets.". This is a general statement on one metric to show the performance of the dataset, not a comment on any specific dataset. Couldn't a good/reliable dataset close the sea level and energy budgets? This is a physical-based metric.

We also did not say that the other datasets used in the budget are perfect; all estimates have uncertainty attached to them, and the sea level budget closure can be assessed in the context of uncertainty (as we did, as all other published studies such as Church et al. 2011, Frederikse et al., 2020, IPCC). Even in your paper, Domingues et al. 2008, you used the budget closure to indicate the improvement of your reconstruction, as I quote: "***The improved closure of the sea-level budget over multi-decadal periods (Fig. 3b) and the better agreement in the magnitude of observed and simulated decadal variability (Fig. 2c, d) increase confidence in the present results and represent progress since the last two IPCC reports2,19.***". Thus, we are confused about your questions on this sentence.

Line 476: Please include figure in Suppl. Material to show this point.

Re: This sentence says: "This approach provides an effective method to quantify the local trend by minimizing the impact of year-to-year variability and start/end points.". We refer to the peer-reviewed study (Cheng et al. 2022b) for more information; this is a dedicated study to introduce and evaluate this LOWESS-based approach to quantifying the local trends. We did not repeat the analysis here.

Line 496: surface area?

Re: Yes, "surface" added.

Line 508: Which of the improvements is making the most difference?

Re: As said, the most important thing is the use of "degree distance" instead of "km distance". The sentence is "The improvement in IAPv4 is mainly because of the methodology improvements: IAPv3 used 1990–2005 data to construct climatology which suffered from errors related to sparse data coverage, use of "degree distance" instead of "km distance", and other error sources.".

Line 528: Please include figure in Suppl. Material to demonstrate this point. What about the added profiles?

Re: The difference is already shown in Fig.7; the upper panels are IAPv3, where the degree distance is used, and the lower panels are IAPv4, where the km distance is used. The features are very illustrating; for example, you can see in IAPv3 that the anomalies are "as rays emerging from the pole" (new texts added in the revised manuscript), which shows the impact of using "degree distance" because if the degree distance is used, the actual km distance goes smaller when going closer to the pole. Hence, the influencing distance of each observation is more and more limited in space.

Line 540: What is the definition of subtropics and midlatitudes? Please include the latitudinal range for each.

Re: There is no clear boundary for the MLD feature changes: they are just changing gradually with space. Nevertheless, in the revised manuscript, we roughly provided the latitudinal ranges: "The MLD shows a much stronger seasonal variation in the subtropics and midlatitudes (for example, 20°~70° in both hemispheres) than in other regions (including the tropics, for example, 20°S~20°N) ".

Line 566: What do you mean by "quantitatively consistent"?

Re: This sentence has been removed.

Line 574: Sparser observations in the ocean or satellite SST?

Re: Both. Sentence modified to "The largest difference between IAPv4 and other SST products comes mainly from the Pacific and the Southern Ocean before 1980, associated with sparser in situ observations for both SST and subsurface temperature data."

Line 608-610: Please include figure in Sup. Material to demonstrate these two points.

Re: The sentence you are referring to is "The updated MBT and XBT corrections are mainly responsible for the difference between 1980 and 2000. Data QC impacts the month-to-month variation of the OHC time series.". We kindly refer you to three previous studies for this statement. The references are now added in this sentence.

XBT: Cheng et al. (2014) Fig.17: Cheng, L., Zhu, J., Cowley, R., Boyer, T., and Wijffels, S.: Time, Probe Type, and Temperature Variable Bias Corrections to Historical Expendable Bathythermograph Observations. J. Atmos. Ocean. Technol., 31, 1793-1825, https://doi.org/10.1175/jtech-d-13-00197.1, 2014.

MBT: Gouretski and Cheng (2020), Fig.10: Gouretski, V., and Cheng, L.: Correction for Systematic Errors in the Global Dataset of Temperature Profiles from Mechanical Bathythermographs. J. Atmos. Ocean. Technol., 37, 841-855, https://doi.org/10.1175/jtech-d-19-0205.1, 2020.

QC: Tan et al. (2023), Fig.15: Tan, Z., Cheng, L., Gouretski, V., Zhang, B., Wang, Y., Li, F., Liu, Z., & Zhu, J.: A new automatic quality control system for ocean profile observations and impact on ocean warming estimate. Deep Sea Research Part I: Oceanographic Research Papers, 194, 103961, https://doi.org/10.1016/j.dsr.2022.103961, 2023.

Lines 620-629: How does the gridded data subsampled at the locations (x,y,z,t) of the actual profiles compare? Is there any significant difference between them? Does the gridded product also use Deep Argo floats? Do the wide known significant CMIP model drifts (particularly in the deep ocean) were removed before the calculation of the co-variances?

Re: Yes, the gridded product used Deep Argo floats in combined with other available data.

The drift is not removed before calculation because drift removal at a global scale only impacts the global T/OHC and does not impact spatial covariance. If one removes the drift at each grid box with local linear or quadratic regression, it generates problems for the model representation of covariance because the resultant fields are not dynamically consistent any more (as the errors in drift removal at local scales are not perfect).

For the comparison of gridded data with the in situ profiles, here we provide an example of the difference between the reconstruction and the observations at 3100m (Fig. R3, upper) compared with the temperature trend at 3100m from reconstructed data (Fig. R3, middle). It seems that the difference pattern is significantly different from the trend pattern. In general, there are no substantial systematical biases in the reconstruction (Fig. R3, lower). For instance, the major warming appears in the Southern Ocean with trends varying from 0.01~0.05 $^{\circ}$C dec$^{-1}$ from 1991 to 2023, so the total temperature changes are 0.33~1.65 $^{\circ}$C. The reconstruction and observations differences are around zero. Using other dept levels yields similar results.

[Figure]

**Figure R3.** (upper) difference between the reconstruction and the in situ temperature observations at 3100m (upper), the unit is degree Ceilsus. (middle) the temperature trend at 3100m from reconstructed data in IAPv4, the unit of ℃ dec$^{-1}$. The data are for 1991-2023 period.

(lower) the zonal mean difference between the reconstruction and the in situ temperature observations at 3100m from 1991-2023.

Figure 11, panel a: There is a large interannual variability around 2000-2005, just before the Argo array achieve its global float target. Could this unusual step change (compared to the other variability observed for the entire record) arise from the radical change in the observing system?   Should a cautionary note be included in the text?

Re: There is no evidence that the variation between 2000-2005 is "unusual". Based on the subsample test by Cheng et al. (2017), the current mapping approach does not lead to significant biases associated with sampling changes, including the 2000-2005 period.

Line 640: Was this reported in a paper before Trenberth et al. 2016? Please cite reference if exists. Please consult with Loeb/Sato.

Re: We don't know a dedicated paper reporting this before Trenberth et al. 2016, please let us know if you know any studies.

Line 649: Did you apply the same smoother to the other timeseries? Is the comparison fair?

Re: Yes, all time series are smoothed in the same way, as stated in the table caption "12-month running mean (13-points are used, with start-point and endpoint 656 weighted by 0.5)". The same approach is also used in Trenberth et al. 2016. One can do a cross-check.

Line 666: How does your SST data compare with satellite SST along boundary current regions? Does it look realistic? Or is it too warm because the QC is not flagging data errors? Missing proper evaluation.

Re: The SST trend pattern has been compared with other data in Fig.10; IAPv4 is consistent with other data, and the pattern is definitely realistic, please check section 3.3 texts.

This section is for OHC, and we compare the OHC trend map between IAPv4 and IAPv3; this sentence is a state of fact shown in Figs. 12, 13 that "The IAPv4 shows stronger warming near the boundary currents regions". For the explanatory sentence of "mainly because of the improved QC that does not flag high-temperature anomalies.", this has been tested and published in Tan et al. (2023). But to better show you this issue, here we provide an example to show the QC's impact (Fig. R1, below). You can see that the impact of QC on OHC is mainly in the eddy-rich regions (Fig. R1 upper panel), and the WOD-QC that is used in IAPv3 is removing more positive anomalies than CODC-QC (Fig. R1 lower panel). A careful check of profiles indicates that most of the removed positive anomalies look realistic. Moreover, altimetry data show that these positive anomalies are located within warm eddies and are associated with thermocline changes (so the temperature anomalies are big at the sea subsurface, around the thermocline).

[Figure]

Fig. R1. (upper) Mean Upper 2000m OHC difference between 2005-2020 between the reconstruction results based on WOD-QC (used in IAPv3) and CODC-QC (used in IAPv4). In the two reconstructions, the other data processing procedures are the same. (lower) The distribution of 500 m temperature anomalies in each 0.01°C bin in the Northwest Pacific Ocean for observations without any QC (red), with WOD-QC (blue) and CODC-QC (black).

Figure 12: Where is there statistical difference between IAPv3 and v4?

Re: We decided not to provide any statistical check on the differences for IAPv4 minus IAPv3, because of the following reasons: 1) it is nearly impossible to have a correct statistical check, because the statistical significance check of the different field relies on the covariance of the two products, because they are not independent. This estimate is nearly impossible; 2) even if we can identify regions with insignificant differences between IAPv4 and IAPv3, they might be important because OHC is an integral quantity, summing up the small numbers might yield a big/non-negligible difference for OHC. In this regard, providing significance for regional differences is misleading and gives readers the wrong impression that the insignificant regions are not important.

For these reasons, we feel that giving the trend maps of IAPv3 and IAPv4 and then showing their differences are sufficient just to illustrate the impact on the trend and trend patterns. The users can then assess if the differences are important or not based on their needs.

Line 686: Why does deep ocean warming occur after 1990 and not before? What is the physical explanation and evidence? Do we have enough deep ocean observations before 1990s?

Re: We here refer to the peer-reviewed paper Cheng et al. 2017 again that using a subsample test indicates that the deep ocean changes within 700-2000 m can be reliably reconstructed since 1960. The uncertainty is relatively large though, see Fig.11 error range.

It is beyond the scope of this study to explain why deep ocean warming occurred after 1990 and not before. This study aims to describe the data/methodologies/results.

Line 712-713: Is the interannual variability statistically significant? Where are the error envelope for the other datasets?

Re: There is substantial/significant inter-annual variability in the derived MHT time series, which are also physically meaningful. We refer to Trenberth et al. (2016, 2019a, 2019b) for more details on the analyses of MHT variability. Also, similar approaches are used to derive MHT in other groups, such as Mayer et al. (2022) and Liu et al. (2020); there are dedicated studies on the inter-annual variability of MHT with similar approach.

For the other datasets, the error range cannot be given here because the uncertainty estimates require the complete estimates of local OHC uncertainty and the spatial error covariances, which are not available for other datasets. This does not impact the conclusion of this study because we are here to compare how better different estimates based on different OHC data fall into the RAPID envelope and their correlation with the RAPID time series.

Trenberth, K. E., Y. Zhang, J. T. Fasullo, and L. Cheng, 2019a: Observation-Based Estimates of Global and Basin Ocean Meridional Heat Transport Time Series. J. Climate, 32, 4567–4583, https://doi.org/10.1175/JCLI-D-18-0872.1.

Trenberth, K. E., and J. Fasullo, 2017: Atlantic meridional heat transports computed from balancing Earth's energy locally. Geophys. Res. Lett., 44, 1919–1927, https://doi.org/10.1002/2016GL072475.

Trenberth, Kevin E. and Zhang, Yongxin, 2019b, Observed Interhemispheric Meridional Heat Transports and the Role of the Indonesian Throughflow in the Pacific Ocean, Journal of Climate Vol. 32, No. 24, pp 8523, 1520-0442

Mayer, Johannes, Mayer, Michael, Haimberger, Leopold, and Liu, Chunlei, 2022, Comparison of Surface Energy Fluxes from Global to Local Scale. Journal of Climate Vol. 35, No. 14, pp 4551, 1520-0442

Liu, Chunlei, Allan, Richard P., Mayer, Michael, Hyder, Patrick, Desbruyères, Damien, Cheng, Lijing, Xu, Jianjun, Xu, Feng, and Zhang, Yu, 2020, "Variability in the global energy budget and transports 1985–2017" Climate Dynamics Vol. 55, No. 11-12, pp 3381, 1432-0894

Figure 16: Please add uncertainty timeseries to demonstrate how an improved ocean observing system is making a difference in reducing uncertainty.

Re: Thanks. Good point. We added the uncertainty range (95% CI) in the shading of our energy budget estimate.

Figure 17: Please include other estimates (IAPv3, EN4, ISH, Johnson et al) for comparison as, for instance, done in Figure 14.

Re: A comparison of different products is currently under development in the other papers in a more comprehensive way (GEWEX GDAP group study led by Dr. Maria Hakuba) with a standardised approach to derive dOHC/dt, common mask and other issues, so we don't want to replicate the efforts add more details here. This paper mainly aims to describe IAPv4 compared with IAPv3.

Line 837 and Table 3: Incorrect AR6-related statements and values. AR6 trends were not based on least-squares fit nor on Frederikse et al. 2020. "Based on the ensemble approach of Palmer et al. (2021) and an updated WCRP Global Sea Level Budget Group (2018) assessment (Figure 2.28) GMSL rose at a rate of 1.32 [0.58 to 2.06] mm yr$^{-1}$ for the period 1901–1971, increasing to 1.87 [0.82 to 2.92] mm yr$^{-1}$ between 1971 and 2006, and further increasing to 3.69 [3.21 to 4.17] mm yr$^{-1}$ for 2006–2018 (*high confidence*). The average rate for 1901–2018 was 1.73 [1.28 to 2.17] mm yr$^{-1}$ with a total rise of 0.20 [0.15 to 0.25] m (Table 9.5)."

Table 9.5 | Observed contributions to global mean sea level (GMSL) change for five different periods. Values are expressed as the total change (Δ) in the annual mean or year mid-point value over each period (mm) along with the equivalent rate (mm yr–1). The very likely ranges appear in brackets based on the various section assessments as indicated. Uncertainties for the sum of contributions are added in quadrature, assuming independence. Percentages are based on central estimate contributions compared to the central estimate of the sum of contributions.

Re: Thanks for this information. Table 3 has been removed.

In the revised manuscript, what we were doing is simpler: download AR6 data and replace the thermosteric sea level time series with IAPv4 thermosteric sea level to check the impact.

Please check the report and/or with authors. Chapters 2 and 9.

https://iopscience.iop.org/article/10.1088/1748-9326/abdaec/meta

Re: Thanks for this information. Please check the supplementary material of Cheng et al. 2022 NREE review paper for our comments on the caveats of this uncertainty estimate approach.

Table 3: Does IAPv4 have GMSL estimates as this table implies?

Re: Table 3 has been removed.

Line 850: If salinity change is irrelevant for global mean sea level, should it be wise to compute thermosteric rather than steric sea level for budget purposes? As salinity data tend to be less reliable than temperature data?

Re: In the revised manuscript, to compare with IPCC-AR6 results, only thermosteric sea level is considered.

Lines 901-903:   What is the relative importance of each factor? Which factor(s) is(are) making the most difference?

Re: This is a good topic for a separate paper (actually we are already working on that) to understand the relative importance of each factor. It is not a simple answer because the different factors are not independent of each other: changing one technique can impact the contribution of the other factors. Thus, this deserves a dedicated study.

Lines 908-919: Please include this in a caveat section, along with other caveats (e.g. use of model covariance and applications not recommended).
Re: See our reply to your first major comment.

Line 922: Does any of the products have enough spatio-temporal resolution to resolve mesoscale variability? Is this being aliased or properly accounted for as error?
Re: None of the 1-degree resolution data products can resolve meso-scale variability (with typical spatial scales of 20~300 km). The key point here is to properly represent the statistical feature/characteristics of the mesoscale eddies (i.e. the averaged temperature changes). The errors related to this issue are "representative error" as quantified and represented in IAPv4 mapping approach (see Method section for details).

Line 928-936: IQuOD is doing much more than just uncertainty. Please represent the comprehensive IQuOD activities properly, and how that might support other activities, such as yours, reanalyses, etc.
Re: Yes, we agree that IQuOD plans to do more, here we just want to describe the most relevant activities, which are already quite broad "For example, the International Quality-Controlled Database (IQuOD) group is coordinating some activities related to data processing techniques, uncertainty quantification, and improving the overall quality of ocean data (Cowley et al., 2021).". We believe it is not a place to completely introduce IQuOD, even though I'm the current co-chair of IQuOD. Please note, similar to your comments on ME4OH, a very extensive discussion on the "wish list" of these projects is dangerous, instead, what has actually been done and the suggestions grounded on the present study is more scientifically meaningful.

Line 937-944: Please cite Boyer et al. 2016, Savita et al. 2022, and IAPSO's ME4OH best practice working group:
https://iapso-ocean.org/images/stories/_working_groups/Best_practice_study_groups/mapeval4oceanheat__2021-proposal.pdf
Re: I'm not sure it is proper to attribute these recommendations only to Boyer et al. 2016, Savita et al. 2022, and IAPSO's ME4OH. First, these comments in the present study are more specific than those presented in Boyer et al. 2016, Savita et al. 2022, for example, we suggest "In the future, comprehensive quantification of other uncertainty sources will be made, including the choice of climatology, vertical interpolation, XBT/MBT/APB/Bottle corrections, etc. It is also necessary to analyze the correlation between these error sources.". This is very specific and has not been fully discussed before. Second, there are more studies recommending various aspects of issues related to the OHC estimates; only attributing these points to your mentioned

references is not proper. Third, our recommendations are grounded on the results of the present study.

Boyer et al. 2016 and Savita et al. 2022 are cited in our manuscript in the proper places. We did not cite the ME4OH document because it is not peer-reviewed literature, and no publication from the ME4OH group has been available until now (to our best knowledge).

---

## Author Response (AR2)

**The authors thank reviewer#1/2 for the additional comments on our study. Here, the point-to-point replies are provided in blue, the comments are in black, and the modified texts for the manuscript are shown in orange.**

**Referee #1's comments and replies:**

I would like to thank the authors for preparing the revised and improved manuscript as well as their extensive responses. I am mostly happy with how my comments have been addressed and only have minor comments where more major changes have been made.

Re: Thank you for the additional comments.

Fig. 6: It is evidently difficult to assess the realism of the seasonal amplitude of Arctic OHC (panel d). Additional evidence could be provided by referring to the data provided in Table A3 of Mayer et al. 2019 (https://doi.org/10.1175/JCLI-D-19-0233.1), who additionally constrained Arctic OHC with atmospheric data by enforcing energy budget closure: conversion of their monthly d/dt OHC to the units used here suggests a peak-to-peak amplitude of Arctic OHC of ~4 ZJ, which is in good agreement with the IAPv4 annual cycle presented in Fig. 6d.

Re: This is a good point, we have added this information in the revised manuscript: "This estimate of the Arctic annual cycle is consistent with a constrained Arctic OHC estimate with atmospheric data by enforcing energy budget closure (Mayer et al., 2019).".

Mayer, M., S. Tietsche, L. Haimberger, T. Tsubouchi, J. Mayer, and H. Zuo, 2019: An Improved Estimate of the Coupled Arctic Energy Budget. J. Climate, 32, 7915–7934, https://doi.org/10.1175/JCLI-D-19-0233.1.

Section 3.6: Please note that sea ice is neglected in the MHT equation but that the introduced error is small on at least annual time scales

Re: This is our oversight. We did use sea ice data (PIOMAS), so we have revised the approach description in the revised manuscript."we integrate the OHCT, air-sea heat flux and heat gain/loss by sea ice changes from the North Pole southward in the Atlantic Ocean, and solve the energy budget equation, the residual at each latitude is the MHT, i.e.,

$$MHT(\varphi) = \int_{\varphi}^{90} \left[ Fs + \frac{dOHC}{dt} + Q_{ice} \right] a \, d\varphi$$

Where $a$ is the Earth's radius, $\varphi$ is latitude, $Fs$ is net surface heat flux, and $Q_{ice}$ is the heat inferred from the changes of sea ice mass. Consistent with Trenberth et al. (2019), this study uses the sea ice volume data from the Pan-Arctic Ice Ocean Modeling and Assimilation System (PIOMAS; Schweiger et al. 2011), and assumes a constant latent heat of fusion of $3.34 \times 10^5$ J kg$^{-1}$ and a density of ice of 900 kg m$^{-3}$."

And, yes, the sea ice contribution is negligible for the derived time series (it is important for the annual cycle).

L882: question -> equation

Re: Corrected.

L1019: please add statement on statistical significance of the correlation coefficient.
Re: Done: "(the correlation is statistically significant at 90% confidence level, where autocorrelation reduction is taken into account)"

Section 3.9: The section reads better now, although an improved table compared to that in the original manuscript might have been useful (but not essential) for an overview. Other minor comments are:
L1054: Long-term GMSL increase is based on which data?
Re: Here we used the time series provided in Frederikse et al. (2020), where the GMSL is derived by combining tide-gauge observations with estimates of local vertical land motion from permanent Global Positioning System stations and the difference between tide- gauge and satellite altimetry observations (Frederikse et al. 2018; 2020).
The sentence is modified to "From 1960 to 2023, the observed GMSL rise is 2.07 ± 0.55 mm yr$^{-1}$ (Frederikse et al., 2020), which is derived by combining tide-gauge observations with estimates of local vertical land motion from permanent Global Positioning System stations and the difference between tide- gauge and satellite altimetry observations (Frederikse et al., 2018)."

Frederikse, T., Jevrejeva, S., Riva, R. E. M. & Dangendorf, S. A Consistent Sea- Level Reconstruction and Its Budget on Basin and Global Scales over 1958–2014. Journal of Climate 31, 1267–1280. doi:10 . 1175 / JCLI - D - 17 - 0502 . 1 (Feb. 2018).

L1055: "sum of contributions": suggest to state "contributors" in parantheses at first occurrence of this expression
Re: Modified to "the sum of contributors (Glaciers, Greenland and Antarctic ice sheets, land water storage, and steric sea level)"

L1073: I assume this is a "temporal" RMSD?
Re: Yes, "temporal" added in the revised manuscript.

L1087: I suggest to move the word "again" down to the end of the next paragraph, where you re-iterate the conclusion regarding the increased warming of IAPv4.
Re: Yes, moved.

L1088: stronger warming compared to what? Please link to the relevant figure/table/study.
Re: This is compared with the steric sea level estimate in IPCC-AR6 (Gulev et al., 2021), which is added in the revised manuscript: "This suggests that the stronger warming since the 1993 revealed by IAPv4 than assessed in IPCC-AR6 (Gulve et al., 2021), seems more realistic."

**Referee #2's comments and replies:**

Fig. 16/18 Captions "Meridional" and "Atmosphere" are capital

Re: Corrected.

Fig.20 Captions: Not clear what "A 6–month running smooth " is. Should probably be replaced with "6–month running mean"

Re: Modified to "6–month running means are shown here to reduce the noise."